# Folding correctors can restore CFTR posttranslational folding landscape by allosteric domain–domain coupling

Naoto Soya [1], Haijin Xu[1], Ariel Roldan[1], Zhengrong Yang[2], Haoxin Ye[1], Fan Jiang[2], Aiswarya Premchandar[1], Guido Veit [1], Susan P. C. Cole [3], John Kappes[2], Tamás Hegedüs [4,5] & Gergely L. Lukacs [1] ✉

The folding/misfolding and pharmacological rescue of multidomain ATP-binding cassette (ABC) C-subfamily transporters, essential for organismal health, remain incompletely understood. The ABCC transporters core consists of two nucleotide binding domains (NBD1,2) and transmembrane domains (TMD1,2). Using molecular dynamic simulations, biochemical and hydrogen deuterium exchange approaches, we show that the mutational uncoupling or stabilization of NBD1-TMD1/2 interfaces can compromise or facilitate the CFTR(ABCC7)-, MRP1(ABCC1)-, and ABCC6-transporters posttranslational coupled domain-folding in the endoplasmic reticulum. Allosteric or orthosteric binding of VX-809 and/or VX-445 folding correctors to TMD1/2 can rescue kinetically trapped CFTR posttranslational folding intermediates of cystic fibrosis (CF) mutants of NBD1 or TMD1 by global rewiring inter-domain allosteric-networks. We propose that dynamic allosteric domain-domain communications not only regulate ABCC-transporters function but are indispensable to tune the folding landscape of their posttranslational intermediates. These allosteric networks can be compromised by CF-mutations, and reinstated by correctors, offering a framework for mechanistic understanding of ABCC-transporters (mis)folding.

ATP-binding cassette (ABC)-transporters are multidomain membrane proteins that are indispensable for transporting a plethora of substrates to maintain organismal homeostasis in all phyla of life. Many eukaryotic ABC-transporters contain two cytosolic nucleotide binding domains (NBD1,2) and two transmembrane domains (TMD1,2), each comprising six TM helices (TMH) on a single polypeptide[1]. The TMDs interface the NBDs through four cytosolic loops (CLs) (Fig. 1a). The CLs drive TMD1/2 conformational transitions upon NBD association-dissociation cycles. The NBD/TMD coupling is a requirement for substrate translocation and is facilitated by domain-swapped structural elements in Pgp-like (Type IV) exporters, which enable communication of each domain with all the others[1–3].

Numerous point mutations cause misfolding and premature degradation of ABC-transporters by the endoplasmic reticulum (ER) associated protein quality control (PQC) machinery, accounting for the development of several human proteopathies[4,5]. The loss-of-functions of ABCC transporters, including ABCC1 (MRP1) and ABCC6, are associated with enhanced susceptibility to drug toxicity and ectopic mineralization, respectively, while the Cystic Fibrosis Transmembrane conductance Regulator (CFTR/ABCC7) anion channel

[1]Department of Physiology and Biochemistry, McGill University, Montréal, QC, Canada. [2]Heersink School of Medicine, University of Alabama School of Medicine, Birmingham, AL, USA. [3]Division of Cancer Biology and Genetics, Department of Pathology and Molecular Medicine, Queen's University Cancer Research Institute, Kingston, ON, Canada. [4]Department of Biophysics and Radiation Biology, Semmelweis University, 1085 Budapest, Hungary. [5]ELKH-SE Biophysical Virology Research Group, Eötvös Loránd Research Network, Budapest, Hungary. ✉e-mail: gergely.lukacs@mcgill.ca

**Fig. 1 | Posttranslational rescue of ΔF508- and L206W-CFTR misfolding by VX-445 and VX-809 pharmacological chaperones. a** Left panel: The inward-facing human CFTR cryo-EM structure (PDB:5UAK). The same color-coding of CFTR domains (NBD1, NBD2, TMD1, and TMD2) is used for all relevant illustrations of MRP1 and ABCC6. P67, L206, and F508 residues were highlighted as spheres. Right panel: Interdomain contacts of CFTR. CL1,4 and CL2,3 form domain interfaces with NBD1 and NBD2, respectively. TMH1-6 and 7–12 are colored with blue-green and red-yellow gradients, respectively. **b** Conformational maturation efficiency of radioactively pulse-labeled L206W-CFTR in the absence or presence of VX-809 in BHK-21 cells. Cells were exposed to 3 μM VX-809 during depletion, pulse, and chase periods as indicated. B-band and C-band are the core- and complex-glycosylated CFTR, respectively. Phosphorimage visualization (left panel) and quantification of the maturation efficiency (right panel). Means ± S.E.M, $n = 6$ and 8. **c** Conformational maturation efficiency of ΔF508- and WT-CFTR was measured by radioactive pulse-chase technique and phosphorimage analysis in BHK-21 cells. Exposure to VX-445 (2 μM) and VX-809 (3 μM) during the Met/Cys depletion, pulse-labeling and/or chase is indicated by "+" in the corresponding rows. Pulse-labeling was also included without the chase (indicated by a gray box in "chase" row) to determine the total radioactivity incorporated into the core-glycosylated CFTR in the absence or presence of correctors. Means ± S.E.M, $n = 7,3,6,3,5$ from left to right. **d** Conformational maturation efficiency of the core-glycosylated L206W-CFTR into complex-glycosylated was measured as described in **c**. Cells were exposed to 3 μM VX-809 during depletion, pulse-labeling and/or chase as indicated by +. The gray box indicates that only pulse-labeling was included without the chase. Means ± S.E.M, $n = 9, 4, 3, 3, 3$. In all figures $P < 0.05$ (*); $< 0.01$ (**); $< 0.005$ (***); $< 0.0001$ (****). Unpaired two-tailed $t$-test. Source data, including specific $P$ values are provided as a Source Data file.

deficiency leads to cystic fibrosis (CF), the most prevalent lethal genetic disease in Caucasian populations[5,6].

The discovery of the synergistic conformational rescue of the most common ΔF508 (F508del) and rare CF-mutations with combinations of suppressor mutations and/or small molecules that target distinct structural defects[7–9] facilitated the development of the FDA-approved Trikafta. Trikafta consists of two folding correctors VX-661 (Tezacaftor, an analog of VX-809) and VX-445 (Elexacaftor), as well as the gating potentiator VX-770 (Ivacaftor)[10–14]. Presently, ~90% of CF-patients are eligible for Trikafta treatment. The binding sites of these drugs have been identified using a combination of techniques, including cryo-EM structural determinations of drugs in complex with WT-, G551D-, and temperature-rescued ΔF508-CFTR in detergent micelles[15–18]. It was proposed that VX-809/VX-661 cotranslationally binds to and stabilizes the TMD1 as it attains its native fold[15,19], a paradigm consistent with CFTR domain-wise folding model[20,21]. This model, however, is challenging to reconcile with (i) the co-translational instability of the CFTR transmembrane helix 6 (TMH6)[22], (ii) the high and comparable protease susceptibilities of the WT and mutant CFTR nascent chains[23,24] and (iii) the recognition of N-terminal half ABC transporters including CFTR as non-native conformers by the ER PQC[25–27].

The CFTR cooperative domain-folding model dictates that following the formation of secondary and some tertiary structural elements co-translationally, the native domain-swapped conformational ensembles develop posttranslationally[15], which requires dynamic and/or energetic interdomain coupling and the expression of the full-length CFTR or its minimal folding unit (TMD1-NBD1-RD-TMD2 or CFTR-ΔNBD2)[8,27]. The cooperative folding model aligns with the observation that the correctors binding sites overlap with the WT-CFTR conformational poses in the cryo-EM structures[15,17], suggesting that conformers can accumulate synchronously with the posttranslational development of permissive inter-domain interactions. The intrinsically unstructured regulatory (R) domain (Supplementary Fig. 1a) was found to be dispensable for CFTR expression[28].

In contrast to the rapid cotranslational two-state folding of small single-domain proteins, the folding kinetics of large multi-domain soluble proteins with complex topology is impeded by the accumulation of co-translational folding intermediates both in vitro and in vivo. Single-molecule and population studies suggest that inter-domain interactions are required for the posttranslational conformational biogenesis of some soluble multi-domain proteins[29–32]. Tethering isolated domains to soluble multi-domain proteins can also influence the structural, dynamic, and energetic characteristics of the constituent domains[33]. Several studies have investigated the ΔF508

effects on the NBD1 structure, but few have attempted to relate these effects in the context of the multi-domain CFTR[34,35].

We posit that the folding landscape biogenesis of ABC-transporters requires an array of inter-domain (TMD1/NBD1/TMD2/NBD2) dynamics- and enthalpic-driven interface interactions. This process is conceivably facilitated by the NBD1 predominant co-translational folding[36] in CFTR and the more delayed folding of other domains. Our model is in line with the observation that the de novo folding efficiency of ABCC-transporters is under kinetic control, which is reflected by the 5–7-fold longer posttranslational conformational maturation than their estimated translational time (~5 min), the impact of NBD1 translational initiation rate, as well as molecular chaperone activities and ribosome interactions[24,37–44].

Here, we employ strategies to investigate the molecular/cellular mechanism of complex domain-folding and -misfolding of ABCC-transporters, as well as the rescue mechanism of the VX-809 and VX-445 CFTR folding correctors that may provide life-changing improvements for CF patients[45]. Using targeted mutational- and pharmacophore-induced perturbations, our working model integrates results that are obtained on three ABCC-transporters at multiple different levels. First, at the cellular level, the transporters' biosynthetic processing, expression, and metabolic stability, as well as their domains' conformational stability, are monitored. Second, at the isolated NBD1 level, we determine the domains' thermal stability and backbone NHs conformational dynamics. Finally, changes in CFTR conformational dynamics are assessed both at the backbone NHs dynamics and fast atomic motions level, using hydrogen-deuterium exchange and mass spectrometry (HDX-MS), and molecular dynamic (MD) simulations, respectively.

## Results

### Posttranslational rescue of CFTR misfolding by VX-809 and VX-445

The pulse-chase technique enables the monitoring of the metabolic fate and folding efficiency of pulse-labeled CFTR molecules, as they develop from immature core-glycosylated (B-band, ~150 kDa) nascent chains to mature core-glycosylated forms that can rapidly exit the ER and are subjected to complex-glycosylation (C-band, ~170 kDa) in the Golgi-complex[23,46,47]. We have demonstrated that the conformational maturation of pulse-labeled WT-CFTR and CFTR-ΔNBD2 folding intermediates takes approximately an hour, regardless whether their exit from ER and N-linked complex-glycosylation are permitted or not[24,27]. To determine whether the VX-809 and VX-445 folding correctors act on co-translational and/or posttranslational folding intermediates, we examined the folding efficiency of the L206W- and ΔF508-CFTR, exposed to correctors during the $^{35}$S-Met/$^{35}$S-Cys pulse-labeling period (10 min) or posttranslationally during the chase period (3 h).

The L206W and ΔF508 CF folding-mutations are confined to the TMD1 and NBD1, respectively (Fig. 1a and Supplementary Fig. 1a), and they reduce the WT-CFTR folding efficiency from ~31% to ~0.7% in stably transfected BHK-21 cells (Fig. 1b, c, Supplementary Table 1). While the VX-809 orthosteric binding to the TMD1 restores the L206W-CFTR posttranslational folding efficiency to that of the WT-CFTR level (Fig. 1b), the ΔF508-CFTR maturation efficiency was marginally increased from ~0.7% to ~2% (Fig. 1c and Supplementary Table 1)[8,47]. Combination of VX-809 with the VX-445 corrector, which targets the TMD2(TMH10-11) and the N-terminal Lasso segment[15,17], restored WT-like folding efficiency (~36%) of ΔF508-CFTR (Fig. 1c), consistent with the mutant steady-state cellular accumulation based on functional studies[14]. In these experiments, BHK-21 cells were exposed to the drug(s) during the Met/Cys-depletion (30 min), the $^{35}$S-Met/$^{35}$S-Cys pulse-labeling, and the chase (3 h) periods to maximize drug-accumulation and -interaction with folding intermediates.

To restrict corrector binding exclusively to the co-translational phase CFTR, we exposed cells to VX-445 and/or VX-809 only during the Met/Cys-depletion and pulse-labeling periods. Under these conditions, the ΔF508- and L206W-CFTR folding efficiency was reduced to ~8–9% (Fig. 1c, d). The residual amount of ΔF508- and L206W-CFTR maturation can be attributed to the rescue of those radioactively labeled nascent chains that are completed before and shortly after the termination of pulse-labeling. In contrast, if drug(s) were present only during the chase periods, when most posttranslational folding occurs, the folding efficiency of the ΔF508- and L206W-CFTR was restored to ~32% and ~35%, respectively (Fig. 1c, d and Supplementary Table 1). These results suggest that the posttranslational conformational ensembles, including the folding intermediates of all core domains are required to support the drug-induced native fold selection of CFTR.

### Uncoupling NBD1-TMD1/2 interface misfolds CFTR

Although the significance of NBD1 interface uncoupling from the CL4 in TMD2 and the NBD1 energetics destabilization in ΔF508-CFTR misfolding has been recognized, their relative contribution(s) to the channel misfolding remains enigmatic[9,24,27,48–51]. To assess the contribution of the NBD1-CL4(TMD2) interface coupling to CFTR folding/stability, we disrupted the F508 side chain cation-π electrostatic interaction with R1070 (in CL4) and its hydrophobic interactions with the CL4 coupling helix (L1065/F1068/Y1073/F1074 (Fig. 2a). While the F508 side-chain truncation preserved the WT-like energetic and kinetic stability of the NBD1 as demonstrated by domain values of melting temperature ($T_m$), folding free energy ($\Delta G^o$), and unfolding activation energy ($\Delta G^{\#}_u$) comparable to those of WT-NBD1 (Fig. 2b and Supplementary Fig. 1b), the F508G-CFTR complex-glycosylated form (C-band) expression and plasma (PM)-density were reduced to ~15% of the WT-CFTR (Fig. 2c, d). CFTR PM-density was measured by anti-HA antibody ELISA, as all our CFTR constructs incorporate a 3HA-tag in the 4$^{th}$ extracellular loop. The F508G mutation also compromised CFTR cooperative domain-folding, evident from the augmented in situ limited proteolysis of all four core domains (TMD1/TMD2 and NBD1/NBD2) upon trypsin or chymotrypsin digestion. The digests were probed by CFTR domain-specific immunoblotting of F508G- and WT-CFTR expressing isolated microsomes (Fig. 2e, f and Supplementary Fig. 1c). The NBD1-, NBD2-, TMD1-, and TMD2-containing fragments, represented by 29–33, 29–31, 35–37, and 53–76 kDa immunoreactive bands, respectively, have been previously validated[24,52].

The trypsin sensitivity of the mature full-length F508G-CFTR and the tryptic F508G-NBD1 fragment was increased by ~20-fold and ~3-fold, respectively, consistent with the compromised posttranslational domain-domain coupling (Fig. 2f). The F508G disrupted domain-domain coupling reduced the maturation efficiency of the core-glycosylated CFTR from 31.0 ± 2.2% to 2.3 ± 0.7% (mean ± SEM, $n = 5$, Fig. 2g and Supplementary Table 1). Accelerated co-translational degradation cannot explain the F508G-CFTR phenotype, as the abundance of the pulse-labeled (10 min) and mRNA normalized core-glycosylated F508G- and WT-CFTR nascent-chain was similar (Fig. 2h).

To assess the possible role of altered NBD1 dynamic interface coupling in F508G-CFTR folding, we hyperstabilized the F508G-NBD1 by introducing the 2PT- or 6SS-suppressor mutations (Supplementary Table 2). The 2PT-suppressor mutations have been established to stabilize NBD1 variants and are mainly located outside of the domain interfaces (S492P, however, is positioned at the NBD1/CL4 interface). The 2PT increased the $T_m$ of the F508G- and WT-NBD1 by ~18 °C (Fig. 2b), as well as improved the F508G-CFTR cellular/PM expression level and cooperative domain folding (Fig. 2c–f). The 2PT-suppressor augmented the F508G-CFTR ER-folding efficiency from ~2.3% to ~20.8% (Fig. 2g and Supplementary Table 1).

Metabolic turnover of complex-glycosylated F508G-CFTR-2PT was reduced but not restored to the WT-CFTR levels, measured by cycloheximide (CHX) chase studies (Supplementary Fig. 1d, e). This

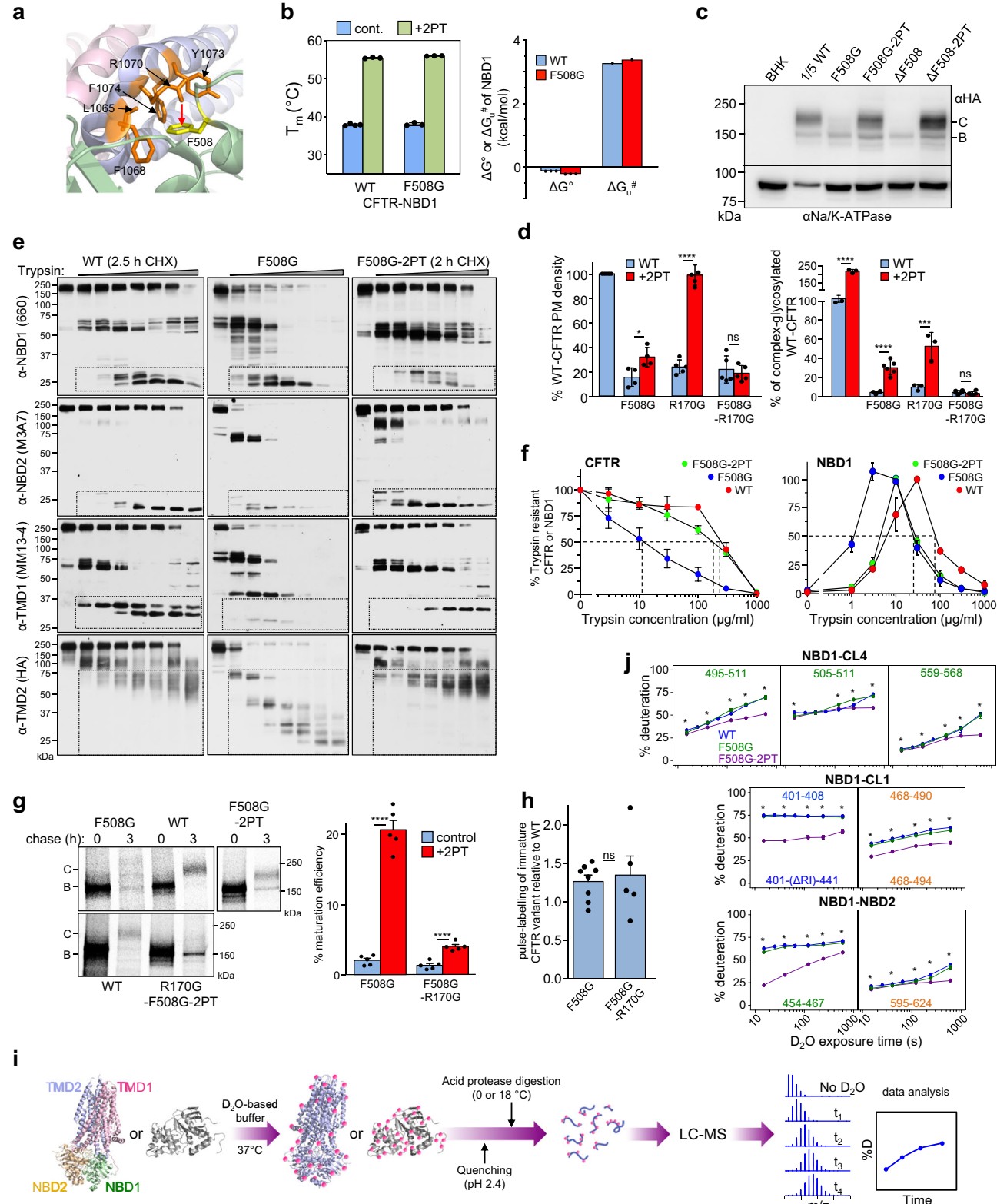

metric is an indirect indicator of the conformational stability of integral membrane proteins in post-Golgi compartments[53–55]. The F508G-CFTR-2PT folding correction was accompanied by TMD1, TMD2, and NBD2 allosteric stabilization (Fig. 2e) that could be explained by the rigidification of the F508G-NBD1-2PT interfaces and its propagated effect on CFTR domain folding. This hypothesis was examined by comparing the isolated WT-, F508G- and F508G-NBD1-2PT backbone conformational dynamic by determining the

domains' amide hydrogens (NHs) solvent accessibility with HDX-MS at peptide level (Fig. 2i). The WT-like HDX kinetics of the F508G-NBD1 was globally attenuated by the 2PT-suppressor (except residues 579–594 and 631–650), including peptides in contact with all three domain interfaces (Fig. 2j and Supplementary Fig. 2a), suggesting that structural and dynamic stabilization of F508G-NBD1-2PT interfaces may participate in the partial folding rescue of the F508G-CFTR-2PT.

**Fig. 2 | CL4- and CL1-NBD1 interactions are required for CFTR allosteric domain folding. a** The F508(NBD1)-CL4(TMD2) interface and interacting residues in the cryo-EM structure of the inward-facing human CFTR (PDB:5UAK). The domain color-coding is as in Fig. 1a (left). **b** The isolated WT- and F508G-NBD1s $T_m$ were determined by differential scanning fluorimetry (DSF). Means ± S.E.M, $n = 3$ (left panel). The F508G- and WT-NBD1 have similar thermodynamic and kinetic stabilities at 37 °C. The $\Delta G^o$ and the $\Delta G^{\#}_u$ were determined as described in Methods. $\Delta G^o$: means ± S.E.M, $n = 3$ and $\Delta G^{\ddagger}_u$ extrapolated values based on $n = 2$ experiments at 4 different temperatures. **c** Immunoblotting of CFTR-3HA variants in equal amounts of BHK-21 lysates, except for the WT-CFTR (20%). Na$^+$/K$^+$-ATPase: loading control. B- and C-bands: core- and complex-glycosylated CFTR, respectively. $n = 4$. **d** The NBD1-CL4 and/or -CL1 uncoupling effect on the PM density (left) and complex-glycosylated CFTR (band-C, right) expression and modulation by the 2PT-suppressor in BHK-21 cells, measured by PM ELISA and immunoblotting, respectively (see Supplementary Fig. 1g). Data are expressed as the percentage of the WT-CFTR and normalized for cell lysate amount and mRNA expression. Means ± S.E.M., panel left: $n = 5,4,4,5,5,5,5$ and right: $n = 5,6,6,6,3,6,6$. Biological replicates.

**e, f** Conformational stability of the full-length CFTR variants and individual domains, probed by limited trypsinolysis and immunoblotting of isolated microsomes. Dotted squares depict proteolytic fragments containing the respective CFTR domain, domain fragments and its antibody epitope. CFTR and NBD1 variants were quantified using densitometry (**f**). Means ± S.E.M, $n = 3$ (WT and F508G-CFTR) and $n = 4$ (F508G-2PT). NBD1 digestion; $n = 3$. Biological replicates. WT:red, F508G:blue, F508G-2PT:green. **g** Folding efficiency of CFTR variants was measured by the pulse-chase technique as described in Fig. 1b. Means ± S.E.M, $n = 5$. Unpaired $t$-test. **h** Pulse-labeling (10 min) efficiency to determine the co-translational degradation of core-glycosylated CFTRs. Phosphorimage quantification was expressed as WT-CFTR percentage, normalized for CFTR mRNA and protein. Means ± S.E.M., $n = 8$ and 5. **i** Flowchart of HDX-MS technique. **j** The interface backbone NHs dynamics of NBD1 variants was determined by HDX-MS. Means ± S.D., $n = 3$, technical replicates. The a.a. number color coding indicates their subdomain localization (green: α-subdomain, orange ATP-binding core, blue: β-subdomain). WT:blue, F508G:green, F508G-2PT: purple. Unpaired $t$-test. Source data are provided in Source Data file.

The contribution of the NBD1-CL1(TMD1) interface coupling to WT-CFTR and F508G-CFTR-2PT cooperative domain-folding was assessed by disrupting the E403(NBD1)-R170(CL1) electrostatic interaction with the R170G CF-mutation (Supplementary Fig. 1f). Although the R170G severely diminished the CFTR cellular/PM expression and caused TMD1/2 and NBD2 allosteric misfolding, R170G has a limited effect on fast dynamical motions of TMD1 (Fig. 2d, Supplementary Figs. 1g, h and 3a). This is consistent with the permissive effect of R170-E403 interaction on CFTR conformational maturation by strengthening the NBD1-CL1(TMD1) interface coupling. Both the expression and folding defects of the R170G-CFTR were rescued by the 2PT-suppressor (Fig. 2d and Supplementary Fig. 1g, h). This rescue, however, was severely compromised by the NBD1-CL4(TMD2) interface uncoupling with the F508G mutation (Fig. 2d, g and Supplementary Fig. 1e, g). Collectively, these results suggest that modulation of NBD1 domain-domain coupling efficiency at at least two domain interfaces can allosterically modulate the posttranslational conformational landscape of CFTR domains.

## MD simulations of CFTR inter-domain allostery and its perturbations

Next, we assessed the dynamical consequences of CFTR inter-domain perturbations at the fast atomic motion level of amino acids by molecular dynamics (MD) simulations. Although the F508G mutation only marginally increased the deuteration of the WT F508-loop peptides locally (Fig. 2j and Supplementary Fig. 3b), it amplified the Cα atom thermal motions of the CL4 (TMD2, a.a.1064–1074) hydrophobic pocket and reduced native contacts at all NBD1 inter-domain interfaces (Fig. 3a, b). These motions were also relayed to the CL1, and NBD1/NBD2 interfaces, manifesting in the NBD2 α-subdomain destabilization (Supplementary Fig. 3c–f).

To reveal mutagenic perturbations in the network dynamics, we assessed the allosteric communication between distant protein parts by determining dynamic communities and their couplings by calculating betweenness centrality in a residue-network(s) based on pairwise residue motion-correlation built from MD simulation trajectories[56]. Physical and allosteric coupling between dynamic communities (e.g.: CH3 and CH4, TMD1+TMD2, TM2ic, and TM3ic) was attenuated or rearranged relative to the WT (Fig. 3b, c and Supplementary Fig. 4), likely contributing to the F508G-CFTR impeded posttranslational conformational maturation and the destabilization of its complex-glycosylated fold (Supplementary Fig. 1c–e).

The R170G mutation decreased the importance (betweenness centrality) of residues in allosteric pathways in its neighborhood, except for the S168 residue (Supplementary Fig. 4a), reflecting a decrease in the redundancy of allosteric signals propagation. Although the increased centrality of the CL4 ensured an augmented dynamic coupling via the F508 residue, this was not sufficient for the complete rescue of the R170G-CFTR global misfolding (Fig. 2d). The R170G-CFTR defect was marked by the allosteric conformational destabilization of CFTR domains, as well as ~80% loss of the complex-glycosylated and PM-resident R170G-CFTR pools, both reversed by the 2PT-suppressor (Fig. 2d and Supplementary Fig. 1g, h). These observations in conjunction with observations on the F508G-CFTR-2PT phenotype (Fig. 2c–g and Supplementary Fig. 1e) support the notion that 2PT-mediated allosteric conformational rescue of the F508G-CFTR requires stabilization of fast atomic motions at both CL1- and CL4-NBD1 interfaces, because simultaneous disruptions of both interfaces thwarted the 2PT-dependent folding rescue (Fig. 2d and Supplementary Fig. 1e, g, h).

The 6SS-suppressor (Supplementary Table 2) that impart a greater thermal[51,57] and backbone dynamic stabilization of the NBD1 than the 2PT-suppressor (Figs. 4a and 2j), restored the CL4 dynamics (Fig. 3a), as well as the native contacts of NBD1-NBD2 and NBD1-CL1, but not of the NBD1-CL4 interface in the F508G-CFTR, as expected (Fig. 3b and Supplementary Fig. 3d). While changes in domain contacts and global dynamics (Supplementary Figs. 3e–f and 5) caused by mutations are not fully congruent, the 6SS-suppressor consistently decreased fluctuations at the NBD1-NBD2 interface and augmented the dynamic coupling between network communities of NBD1-NBD2 that allosterically stabilized the intracellular loops, as indicated by a large central community consisting mainly of CL residues (cyan spheres in Fig. 3c and Supplementary Fig. 4b). The 6SS-elicited rescue uncovered with MD, in conjunction with the 2PT-induced F508G-CFTR processing and stability rescue, support the notion that modulation of the NBD1 inter-domain interface dynamics can rewire the CFTR posttranslational folding energy landscape and partially restore F508G-CFTR cooperative domain folding and processing.

## Coupled domain folding of ABCC1 and ABCC6

To determine if NBD1-TMD2 domain coupling is critical for the biogenesis of other ABCC transporters, the human multidrug resistance protein 1 (MRP1/ABCC1), an organic anion exporter[58], and the ABCC6 ATP-exporter[59], were subsequently investigated. The conserved F713(ABCC6) and F728(MRP1) of NBD1 H3-H4 loops (Supplementary Fig. 6a) are engaged in a hydrophobic patch formation with F1140/F1146 (CL7/TMD2) and a cation-π interaction with the R1173 (CL7/TMD2) (Supplementary Fig. 6b), respectively.

As the isolated NBD1s of MRP1 and ABCC6 displayed 7–10 °C higher $T_m$ (46–52 °C) than the CFTR-NBD1, their thermal resistance to unfolding upon interface uncoupling was predicted to be higher than CFTR NBD1 at 37 °C (Fig. 3d). To destabilize the NBD1-CL7(TMD2) interfaces, the F713G or F728G mutations were introduced that did not alter the thermodynamic and kinetic stabilities of NBD1s, based on their $T_m$, $\Delta G^O$, and $\Delta G^{\#}_u$ determination (Fig. 3d, e and Supplementary Fig. 6c, d). Despite the preserved backbone conformational dynamics

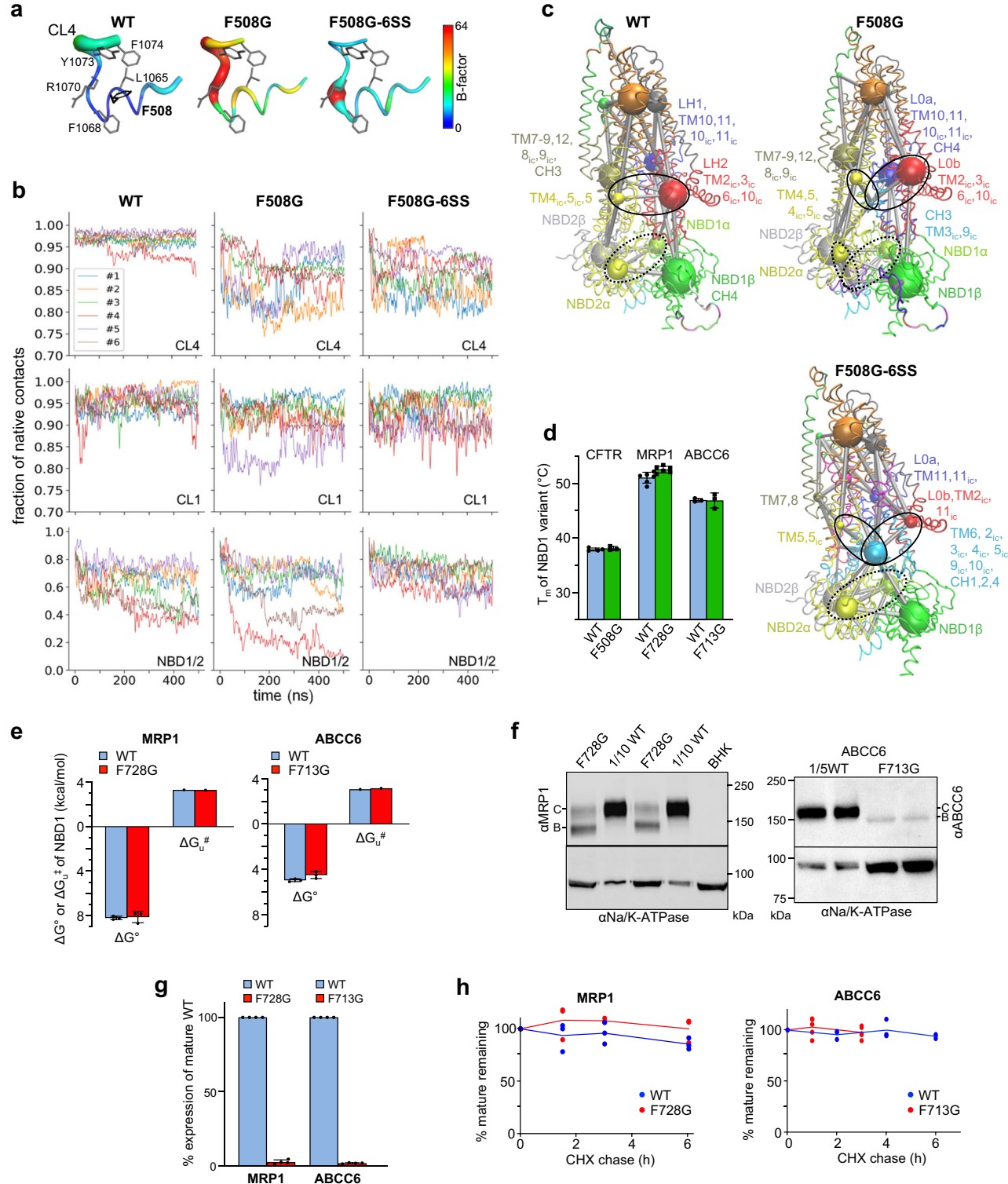

of the mutant NBD1s, suggested by their unaltered deuteration kinetics (Supplementary Fig. 2b, c), the expression of complex-glycosylated F713G-ABCC6 and F728G-MRP1 was diminished to 5−10% of their WT counterparts (Fig. 3f, g). This cannot be attributed to the complex-glycosylated transporters destabilization, as their turnover was preserved in vivo, measured by CHX chase and immunoblotting experiments (Fig. 3h and Supplementary Fig. 7a, b). Thus, the NBD1-TMD2 interface disruption similar to that of CFTR reduces the conformational maturation of core-glycosylated folding intermediates, indicated by domain-wise conformational destabilization of the mature

F728G-MRP1 relative to that of WT-MRP1. This was evident in the proteolytic digestion pattern of microsomes, probed with domain-specific antibodies against MRP1, as well as by the delayed ER-degradation of the F713G-ABCC6 following the proteasome inhibition with bortezomib (Supplementary Fig. 7c, d).

The F728G-MRP1 NBD1-TMD2 interface uncoupling by the F728G mutation was documented by MD simulations (Supplementary Fig. 8a−c). In addition, a reduction in NBD1/NBD2 native contacts occurred in two out of three trajectories when compared to that of the WT-MRP1 (Supplementary Fig. 8c). The introduction of the F728G into

**Fig. 3 | CFTR allosteric dynamics perturbation by deletion of the F508 side-chain. a** Root mean square fluctuations (RMSF) are indicated on the CFTR CL4 structure using color and thickness coding. Higher RMSF values are indicated by warmer colors and thicker tube and calculated from $n = 6$ MD trajectories (frames = 30,000). **b** Fraction of native contacts of CL4 and CL1 with the rest of the protein, and between NBD1 and NBD2 were calculated over all trajectories and plotted for WT- and F508G-CFTR, as well as F508G-CFTR-6SS. **c** Allosteric coupling was investigated by community analysis of residue networks based on motion correlation. Amino acids in a community are color-coded. The spheres indicate the community size. The thickness (1-weight) of edges between spheres represents the dynamic coupling between communities. Black dotted and solid ovals highlight the alterations in coupling between the α-subdomain of NBD1 and NBD2 in WT (weight between communities: 0.6576), F508G (0.6922), and F508G-6SS (0.9274), and between the distant TMH4/5(CL2) and Lasso in WT (0.6576) or F508G (0.6918), and

F508G-6SS (0.9285), respectively. **d** The $T_m$ of the isolated NBD1 variants of CFTR, MRP1, and ABCC6. Means ± S.E.M., $n = 4,3,6,6,3,3$ from right to left. Biological replicates. **e** The folding free energy ($\Delta G^o$) and unfolding activation energy ($\Delta G^{\#}_u$) of WT and mutant NBD1 variants of ABCC6 and MRP1 were estimated by CD-spectroscopy with urea- and temperature-dependent unfolding as described in Methods. $\Delta G^o$: means ± S.D.M., $n = 3$, $\Delta G^{\#}_u$, extrapolated values based on $n = 2$ experiments at 4 different temperatures. **f, g** Effect of NBD1-CL7 interface uncoupling on the complex- (C-band) and core-glycosylated (B-band) ABCC6 and MRP1 expression in BHK-21 cells (**f**), determined by quantitative immunoblotting (**g**). Only 10 or 20% of WT variants were analyzed. Means ± S.E.M., $n = 4$. Biological replicates. **h** The metabolic stabilities of complex-glycosylated MRP1 and ABCC6 were monitored by CHX chase and quantitative immunoblotting. Means ± S.E.M., $n = 3$ except WT-ABCC6; $n = 5$. Biological replicates. WT:red, mutants:green. Source data are provided in Source Data file.

MRP1 increased the number of dynamic communities in relation to that of the WT transporter (Supplementary Fig. 8d). For example, the β-subdomain of NBD1 was split into two communities, indicating highly perturbed motions of NBD1 residues. We also observed a considerably reduced coupling between distant regions (e.g.,: the NBD2 α-subdomain and TMD1) and an overall alteration of the dynamic network within TMDs (Supplementary Fig. 8c), consistent with the MRP1 allosteric domain misfolding (Supplementary Fig. 7c).

Thus, uncoupling of NBD1-CL7(TMD2) interfaces results in severe defects in post-translational coupled domain-folding for MRP1, and by inference for ABCC6, without energetic destabilization of the corresponding isolated NBD1 subunits.

### HDX probes CFTR interdomain allostery
Intramolecular allostery has been established to regulate the CFTR and MRP1 function[7,24,27,50,51,60–62]. The observation that the ΔF508-NBD1 primary folding defect and its global consequences in the core domains were rescued by both suppressor mutations in cis or correctors binding to the TMDs or NBD2 in trans[12,15,51,57,63,64] suggest that dynamic allosteric pathways may be involved in CFTR global folding and stability co- and posttranslationally. Here, we posit that an aspect of these long-range domain coupling can be investigated by determining their propagated changes in backbone amide dynamics provoked by the 6SS-induced NBD1 CFTR hyperstabilization.

At the cellular level, the 6SS-suppressor improved the conformational maturation and steady-state expression of the complex-glycosylated CFTR (Supplementary Fig. 9a, b), whereas it did not alter the relative activities of the resting and cAMP-dependent protein kinase (PKA)-dependent chloride transport in bronchial epithelia, based on short circuit current measurements ($I_{sc}$) in CFBE14o- cells (Supplementary Fig. 9c). The comparable $I_{sc}$ magnitude of the WT- and 6SS-CFTR expressing CFBE14o- cells remains to be addressed.

To determine the backbone NHs dynamics, we isolated the WT- and 6SS-CFTR in synthetic glyco-diosgenin (GDN)-micelles (Supplementary Fig. 9d). The 6SS-suppressor increased the purified CFTR $T_m$ by ~5 °C (Supplementary Fig. 9e). The isolated CFTRs displayed comparable PKA-stimulated ATPase activity (Supplementary Fig. 9f), determined by using NADH-coupled ATPase assay as reported[65,66].

Continuous deuteration kinetics of isolated CFTRs was monitored at 37 °C with an overall sequence coverage of ~84% (Supplementary Fig. 10a). Intuitively, the 6SS-suppressor not only stabilized the conformational dynamics of peptides positioned at interfaces of the isolated NBD1 (Fig. 4a and Supplementary Fig. 2d), but also of peptides positioned in other domains, beyond NBD1 interfaces in the full-length CFTR. This was visualized by projecting the differential deuteration (%ΔD$_{(6SS-WT)}$) of individual peptides on the hCFTR cryo-EM structure (Fig. 4b and Supplementary Fig. 10b) and plotting the deuteration time course (Fig. 4c, Supplementary Figs. 10c and 11). The dynamic stabilization of TMD1/TMD2 and NBD2 regions is shown after 20 min (Fig.4b, c), as well as 10 s and 4 min deuteration

(Supplementary Fig. 10b). The pervasive stabilization of the TMD1/2 coupling helix 1 (CH1) and CH4 interfacing the 6SS-NBD1 progressively decayed along the neighboring peptides (Fig. 4c). The deuteration of the CL4 (e.g.,: a.a.1033–1043, 1063–1067) and the TMH11 (e.g., 1088–1092), as well as the CL1 (e.g., 164–170), including its CH1, and the adjacent TMH3 (e.g.,: 184–194), was inhibited by ~10–40% (Fig.4c, Supplementary Figs. 10c and 11a–e). The backbone dynamics were also attenuated at the TMH4 (e.g.,: 247–254 and 251–258), CL2 (e.g., 266–270), and CL3 (954–964), as well as propagated to the N-terminal Lasso (or L0) region (e.g., 15–21 and 50–56), and the TMD1-NBD1 linker (e.g., 379–383) but not the R domain (Fig. 4b and Supplementary Fig. 10b, c, 11a–e). The TMH1, TMH2, TMH5, and TMH8 segments reduced deuteration became apparent after 4 min incubation, while that of the TMH11 was suppressed from 10 s onwards (Supplementary Fig. 11a–e). The delayed deuteration of NBD2 peptides outside the NBD1 and TMD1/2 interfaces (e.g., 1228–1232, 1233–1239, 1315–1319, 1320–1324, and 1325–1336) was also observed (Supplementary Fig. 11a–e).

The thermodynamic and kinetic stabilization of TMDs peptides was calculated in the 6SS-CFTR relative to that of the WT-CFTR. The 6SS-suppressor reduced the opening free energy difference of the TMD1 (L0, CL1, CL2), TMD2 peptides ($\Delta\Delta G^o_{(WT-6SS)}$ ~0.1–1.2 kcal/mol), while increased the unfolding activation energy of CL4 peptides ($\Delta\Delta G^{\#}_{u(6SS-WT)}$ ~1.0–1.9 kcal/mol) in the final fold (Fig. 4d–e). Thus, allosteric dynamic rigidification contributes to the stabilization of multiple domains in the 6SS-CFTR (e.g., Fig. 4b, c), albeit the channel global static and functional conformations remains similar to that of the WT-CFTR, suggested by their comparable sensitivity to PKA-mediated activation (Supplementary Fig. 9c) and the overlapping cryo-EM structures of the E1371Q-CFTR and G551D-E1371Q-CFTR-6SS[17,67]. As a corollary, we infer that the energetic and dynamic perturbations of domain interface by CF-mutations may allosterically impair other interfaces and CFTR biogenesis, including its final fold.

### CFTR domain packing stabilizes NBD1
The impact of inter-domain packing on the NBD1 conformational stability was investigated by comparing the HDX kinetics of NBD1 in isolation versus in the context of full-length WT-CFTR. The deuteration kinetics of several NBD1 peptides was inhibited in the β-subdomain (S1–S2 and S4), in the ATP-binding α/β-core subdomain (S3, S6, and H1), as well as in the α-subdomain (H3) in purified CFTR (Fig. 4f, g). The estimated cumulative $\Delta\Delta G^o_{(NBD1-CFTR)}$ stabilization of NBD1 peptides was ~5 kcal/mol based on their HDX kinetics in the context of WT-CFTR (Fig. 4h), which was attenuated for the 6SS-NBD1. These results revealed that multidirectional dynamic/energetic interdomain allosteric networks improve the thermal stability of the newly translated NBD1 in the context of full-length CFTR and can, in principle, also relay the pharmacological stabilization of TMD1/2 to rescue the primary or coupled folding defects of NBD1, which was assessed next.

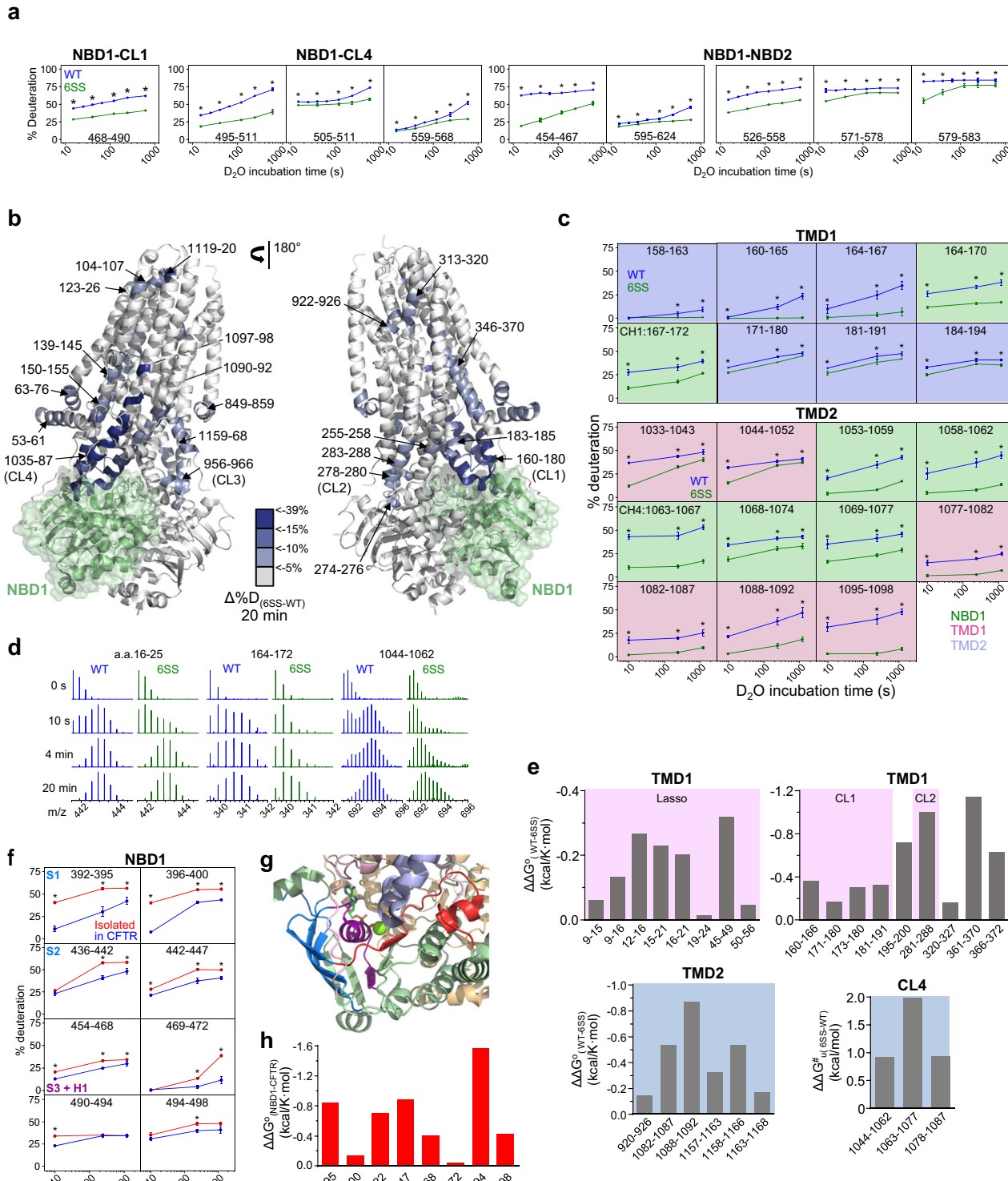

### P67L- and L206W-CFTR allosteric misfolding are rescued by VX-809 and VX-445

Although VX-809/VX-661 correctors were proposed to stabilize the TMD1 native fold by binding to the hydrophobic pocket formed by the TMH1, 2, 3, and 6 of the WT-CFTR[15,18,19] (Fig. 5a), the mechanism of action of VX-809/VX-661 or VX-445 on variants carrying CF mutations has yet to be elucidated. To interrogate the molecular mechanism of correctors, we employed the P67L and L206W CF-mutations (Fig. 5a) that are in the L0 and the TMH3(TMD1) regions in CFTR, respectively, and are susceptible to pharmacological correction[15,18,19,68,69].

The P67L and L206W mutations elicit cooperative CFTR domain misfolding (Fig. 5b) similar to that of other NBD1, TMD2, and NBD2 CF point mutations, probed by limited trypsinolysis and domain-specific immunoblotting[27]. The P67L, as well as the L206W, causes defects in conformational maturation and cellular/PM expression of CFTR (Fig. 5c–f). VX-809 exposure, however, could restore coupled domain-folding of both P67L- and L206W-CFTR, as indicated by their native-like conformational stability of individual domains (Fig. 5b), WT-like posttranslational folding efficiency and steady-state cellular/PM expression of complex-glycosylated forms (Figs. 5c–f and 1b), as well

**Fig. 4 | Global stabilization of 6SS-CFTR by dynamic inter-domain allosteric networks. a** The deuteration kinetics of isolated WT(blue)- and 6SS-NBD1 (green) was depicted at the inter-domain interface by representative peptides. Means ± S.D., $n = 3$, technical replicates. Unpaired $t$-test. The entire domains HDX kinetics are shown in Supplementary Fig. 2d. **b** Differential HDX kinetics of selected peptides obtained in purified WT- and 6SS-CFTR ($\Delta\%D_{(6SS-WT)}$). The $\Delta\%D_{(6SS-WT)}$ of TMD1/2 and NBD2 after 20 min deuteration is projected on the hCFTR cryo-EM structure (PDB:6MSM) by pseudo-color representation. **c** Deuteration kinetics of WT- and 6SS-CFTR peptides in TMD1 and TMD2. Peptides representing the coupling helices (CH1, CH4) of CL1 and CL4 are indicated. The individual peptide domain-specific interactions are color-coded as in Fig.1a. WT(blue)- and 6SS-NBD1 (green). The HDX kinetics of the entire CFTR is shown in Supplementary Fig. 11.

Means ± S.D., $n = 3$ technical replicates. $P < 0.05$ (*). **d** Representative mass-spectra of WT- and 6SS-CFTR peptides showing the EX1 bimodal distribution. **e** The differential opening free energy ($\Delta\Delta G^o_{(WT-6SS)}$) and unfolding activation energy ($\Delta\Delta G^{\#}_{(6SS-WT)}$) of selected peptides were determined from their HDX time-course. Means, $n = 2$ biological replicates. **f** Deuteration time-course of NBD1 in isolation (red) and in CFTR (blue). Only the NBD1 stabilized regions in full-length CFTR assembly are shown. NBD1 in isolation (red). Means ± S.D., $n = 3$, technical replicates. **g** Domain-domain interactions stabilize the β-subdomain (blue), Walker A region (purple) and Q-loop region (red) of NBD1 in CFTR as compared to the isolated NBD1. **h** The $\Delta\Delta G^o_{(NBD1-CFTR)}$ stabilization of the isolated NBD1 in CFTR was calculated based on the time-course HDX. Means, $n = 2$ biological replicates. Source data are provided as a Source Data file.

as the post-ER metabolic stabilization of the mature P67L-CFTR (Fig. 5g). The higher P67L-CFTR rescue by VX-809 than VX-445 (Fig. 5d, e) may be attributed to their orthosteric versus allosteric stabilization of TMD1 (Fig. 5a), based on their binding site localization in the TMD1 and TMD2, respectively, in CFTR-E1371Q[15], temperature rescued ΔF508-CFTR-E1371Q[64] and G551D-CFTR-6SS[17]. In line, the combination of VX-445 with VX-661/VX-809 exert a prominent rescue effect on the ΔF508-CFTR[12] (Fig. 1c) and F508G-CFTR (Fig. 5h) posttranslational misfolding by their global allosteric stabilization, in analogy to that observed for allosterically acting investigational correctors[63].

The VX-809 elicited inter-domain allosteric communication for NBD1/2 folding is likely relayed via multiple CLs in P67L- and L206W-CFTR (Fig. 5b), a notion supported by the attenuated cellular/PM expression of these mutants upon uncoupling their NBD1-CL4(TMD2) interfaces with the F508G substitution (Fig. 5d–f). In line, the F508G-CFTR rescue is highly and additively susceptible to VX-809 and/or VX-445 correction (Fig. 5h), consistent with the function of multiple inter-domain allosteric networks in CFTR (Fig. 3c)[70]. The significance of the reciprocity of inter-domain coupling is also underscored by the P67L-CFTR partial rescue by introducing the 6SS-suppressor in the NBD1 (Fig. 5i).

To directly demonstrate the P67L-induced inter-domain perturbations at the level of backbone dynamics and their reversal by folding correctors, we used the HDX-MS technique on isolated CFTR variants. As the P67L mutation reduced the WT-CFTR expression by >90% (Fig. 5d, e), we selected the P67L-CFTR-6SS variant, which partially mended the P67L-CFTR misprocessing, while ensuring acceptable protein yield (Supplementary Fig. 9d). The P67L mutation reduced the thermal stability of the 6SS-CFTR (Supplementary Fig. 9e) and preserved the susceptibility to VX-809 and VX-445 rescue (Fig. 5d right panel and 5i).

The effects of P67L mutation on fast atomic motions were assessed in MD simulations at the F508-loop, CL4, and NBD2. Similar motion acceleration was observed in the F508-loop and NBD2 compared to F508G, while the CL4 showed reduced dynamics in P67L (Supplementary Figs. 3c, and 5a–c). Conversely, in the 6SS constructs the CL4 dynamics decreased in F508G and restored in P67L. However, in both the F508G and P67L backgrounds the 6SS increased dynamic coupling of the intracellular parts of central helices (cyan community in Supplementary Fig. 4b).

The differential HDX ($\Delta HDX_{P67L-WT}$) also revealed propagated destabilization of 6SS-CFTR peptides in all domains by the P67L mutation. These include the TMD1 (L0 region, TMH5, and CL1), the TMD2 (TMH9, CL4, and TMH12), as well as the NBD1/NBD2 (Fig. 6a and Supplementary Fig. 12a). In the presence of VX-809, however, the P67L-induced backbone destabilization was partially reversed in the L0 region (aa. 50–60 and 73–78 in the VX-809 binding site proximity) and distantly in the TMD2 (CL3, CL4 and TMH12), reflected by the delayed deuteration kinetics (Fig. 6b and Supplementary Fig 12b). These events partly led to the P67L-CFTR-6SS cellular rescue and its resculpted folding landscape in the presence of VX-809 (Fig. 5b–d, i). The results underscore the prediction that destabilization of a mutant

domain can also compromise interdomain allosteric networks and lead to cooperative domains misfolding of P67L-CFTR and P67L-CFTR-6SS (Supplementary Fig. 4a–c and 5b, c).

The VX-445-induced backbone stabilization of the P67L-CFTR-6SS was initiated at the N-terminal Lasso region and propagated to the CL3, CL4, and NBD2-linker region in the TMD2 (Fig. 6c and Supplementary Fig. 12c). Although peptides from the known VX-445 binding site at TMH10-11[64] were not recovered in our HDX-MS analysis, we observed variable reduction in peptides deuteration in the Lasso helix1 (Lh1), TMH10, and TMH12. These results are in line with VX-445 direct interaction with F16, S18, R21, and R25 residues of the Lh1 in the G551D-CFTR-6SS[17] and temperature-rescued ΔF508-CFTR-E1371Q[64] cryo-EM structure, as well as the attenuated P67L-CFTR-6SS rescue by VX-445 relative to VX-809 (Fig. 5i). They also suggest that the binding of CFTR folding corrector to the TMDs can shift the mutant conformational ensembles towards native TMDs conformers, thereby eliciting long-range dynamic and steady-state stabilization of interfacing domains.

## Discussion

Here, we demonstrated that strengthening or attenuating NBD1-TMD1/2 interdomain coupling can significantly augment or diminish, respectively, the posttranslational cooperative domain-folding of CFTR, documented by changes at fast atomic motions, backbone, and domain conformational dynamics, as well as the natively folded channel expression level. Selective disruption of NBD1-TMD2 interface by mutations also compromised the MRP1 and ABCC6 transporters posttranslational folding without destabilizing their isolated NBD1s. Collectively, our results suggest that allosteric inter-domain networks of folding intermediates can modulate the ABCC-transporters folding landscape and are susceptible to pharmacophore-dependent enhancement at the ER[24,37–43].

We propose a simplified two-step folding model for the in vivo multi-layered conformational biogenesis of ABCC transporters (Fig. 7). This model incorporates critical inter-domain conformational coupling events that, by diminishing the folding intermediates' propensities to fall into kinetic traps, ensure the development of native conformers at the right time scale (Fig. 7a). While the first folding phase consists of the obligatory co-translational formation of secondary structural elements of domains, the native tertiary fold completion is likely delayed by the combination of (i) ribosome interactions with CFTR nascent chains, (ii) the large size and complex subdomains' topology of NBDs, and (iii) the slow posttranslational development of inter-domain coupling due to the proteins' domain-swapped structure (Figs. 1a and 7a)[36,42,71,72]. Allosteric communication is required to minimize kinetic traps along the folding pathway, as well as to achieve stabilization of individual native-like domain, as suggested by the domain packing-induced NBD1 energetic stabilization in the full-length CFTR (Fig. 4f–h) and the global consequence of the localized NBD1-CL1/CL4 interface perturbations.

As is the case for other ABC-transporters[37,39,40], correct folding of individual domains of CFTR requires the completion of translation, followed by the posttranslational folding phase. The paradigm that the

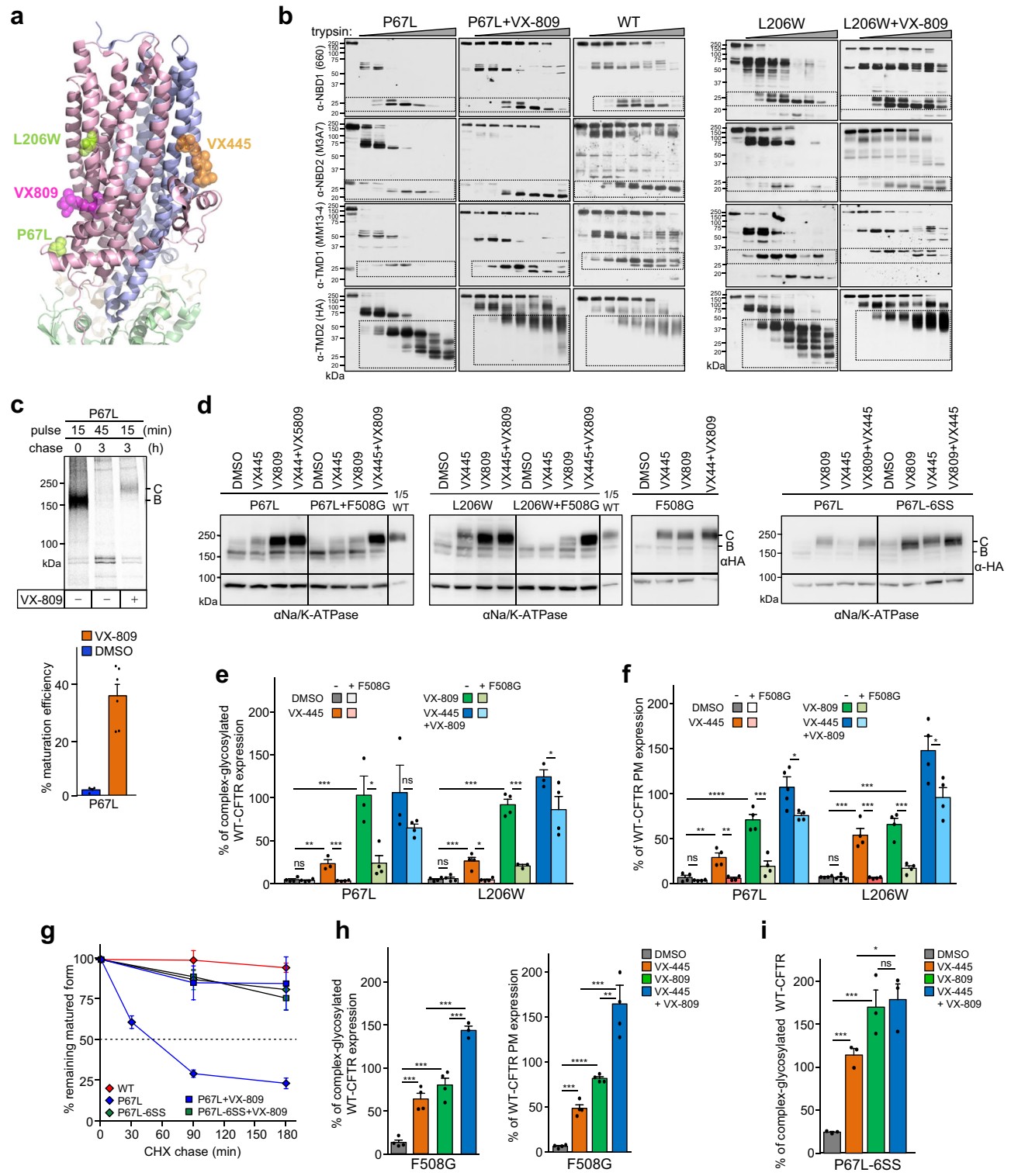

native tertiary structure of CFTR is largely acquired post-translationally is consistent with the comparable protease susceptibilities and ER degradation kinetics of newly synthesized (core-glycosylated) WT- and ΔF508-CFTR nascent chains[23,24,46,73] together with the indispensable role of molecular chaperones during the post-translational folding of ABCC-transporters[39,41,74,75]. This paradigm is also supported by the efficient post-translational rescue of L206W- and ΔF508-CFTR misfolding by correctors (Fig. 1c, d), when all translated domains can participate in cooperative folding. Localized development of TMHs tertiary structure is likely facilitated when the NBD1s of

CFTR[36,37] and of the more stable ABCC6 and MRP1 attain their compact core with loosely packed interfaces. These molten-globule-like NBD1s may serve as intramolecular chaperones in a process that resembles the action of molecular chaperones on their clients' local folding landscape[76]. Consistent with this model, the "premature" co-translational inter-domain cooperativity of TMD1-NBD1 has been recently suggested[21].

While the posttranslational conformational maturation of a handful of soluble multi-domain proteins is aided by coupled domain-folding, its mechanism remains largely enigmatic[29–32,45,77]. The results

**Fig. 5 | CF-causing mutations disrupt inter-domain allostery and its pharmacological reversal by VX-809 and VX-445 correctors, which require the NBD1-CL4 coupling. a** Location of P67L and L206W CF-mutations and the corrector (VX-809, VX-445) binding sites in CFTR (PDB:7SVR). **b** Limited trypsinolysis of P67L-, L206W-, and WT-CFTR microsomes, probed with domain-specific anti-CFTR immunoblotting. BHK-21 cells were exposed to 3 μM VX-809 for 24 h before microsome isolation. Dotted squares depict proteolytic fragments containing the respective CFTR domain epitope. Representative of $n = 3$. **c** The conformational maturation efficiency of P67L-CFTR in the absence or presence of VX-809 (VX) by the radioactive pulse-chase technique in BHK-21 cells. Cells were exposed to 3 μM VX-809 during depletion + pulse + chase periods. B- and C-band: core- and complex-glycosylated CFTR, respectively. Phosphorimage visualization (top panel) and quantification of the maturation efficiency (bottom panel). Means ± S.E.M, $n = 4,6,6$. **d** The susceptibility of the P67L-, L206W-, P67L-6SS-, and F508G-CFTR or the CFTR-F508G to VX-809 (3 μM), VX-445 (2 μM), or VX-809 + VX-445 rescue after

overnight exposure in BHK-21 cells, determined by immunoblotting. The combination of CF-mutations with the NBD1-CL4 interface uncoupling (F508G) diminishes their rescue efficacy. $n = 4$ Abbreviations as indicated in Fig. 2c. **e** Densitometric analysis of the complex-glycosylated (C-band) abundance of panel d. Means ± S.E.M, $n =$ P67L (4,4,3,4,3,4,4) and L206W (4,4,4,4,3,3,4). **f** The VX-809 and VX-445 rescue susceptibility of P67L-, L206W-, and F508G-CFTRs, determined by plasma membrane (PM) ELISA. Means ± S.E.M, $n = 4$ (P67L), 4 (L206W), biological replicates. **g** Metabolic turnover of the complex-glycosylated WT- and P67L-CFTR variants in the absence and presence of VX-809, determined by CHX chase and quantitative immunoblotting. Means ± S.E.M, $n = 4$ (WT), 5 (P67L), 6 (P67L + VX809), 6 (P67L-6SS), 6 P67L-6SS + VX809), biological replicates. **h, i** Rescue of the complex-glycosylated (C-band) and PM expression of F508G-CFTR and P67L-6SS-CFTR by the indicated correctors. Means ± S.E.M, F508G band C: $n = 4$, PM-density: $n = 4$. **i** F508G band C $n = 3$, biological replicates. Source data are provided in Source Data file.

presented here support the inference that mutual dynamic/energetic stabilization of NBD1 and TMD1/TMD2/NBD2 via multiple domain-domain interactions is required for completion of the extended posttranslational domain-folding phase of CFTR and, conceivably, of other ABCC (e.g., MRP1 and ABCC6) transporters, a process that is assisted by molecular chaperones[39,74]. The native conformer formation may be promoted by the enhanced intrinsic NBD1 interface stability of ABCC6 and MRP, facilitating the TMD1/TMD2/NBD2 posttranslational folding (Fig. 4b–e). However, we also uncovered that the less stable CFTR NBD1 is susceptible to reciprocal stabilization by CFTR domain packing, as indicated by the NBD1 reduced solvent accessibility in the context of CFTR (Fig. 4f–h). Conversely, the NBD1 6SS-suppressor were able to elicit a long-range dynamic stabilization of multiple CFTR domains, according to HDX-MS measurements (Fig. 4b–e).

Based on our results, we infer that the 6SS-induced long-range dynamic and conformational stabilization at the domain, backbone NHs, and fast atomic motion levels can reshape the conformational folding landscape of the F508G-, R170G-, P67L-, and L206W-CFTR variants. These observations with the cooperative domain misfolding and severely impeded posttranslational folding efficiency of the F508G-, R170G-, P67L-, and L206W-CFTR, as well as the results of HDX-MS and MD simulation demonstrate that the propagated effect of 6SS (or 2PT) on domain-domain coupling can promote cooperative domain stabilization and folding.

The cooperative domain coupling appears to be a general phenomenon, as the P67L- and L206W-CFTR allosteric domain misfolding was rescued by the TMD1 corrector (VX-809), an effect that was propagated by backbone dynamic stabilization to other domains, probed with limited proteolysis, MD simulations, and HDX-MS. These results can explain the long-range effects of disease-causing point mutations that provoke allosteric misfolding of all CFTR domains[27] and led the misprocessing of other ABCC transporters (Fig. 7a)[78,79]. Selective uncoupling of the NBD1-TMD2 interface was sufficient to elicit global folding and processing defects of MRP1 and ABCC6 transporters (Fig.7).

Intriguingly, the small amounts of complex-glycosylated ABCC-transporters that escaped the ER QC displayed either unaltered (ABCC6 and MRP1) or increased (CFTR) folding free energy, deduced by their metabolic stabilities, which likely influence their residual function. The biological relevance of ABCC-transporters allosteric domain folding is consistent with several other observations: (i) the marginal expression of CFTR individual domains or their limited combinations in bacteria and eukaryotes[9,51,80], (ii) the rescue of ΔF508-CFTR biochemical and functional phenotype with trans-complementation[26], (iii) the NBD1 bidirectional allosteric domain coupling in the context of CFTR (Figs. 2d and 4h), (iv) the post-translational allosteric mechanism of action of CFTR folding correctors (Figs. 1c, d and 6), and v) the improved activity of the BtuCD ABC

transporter upon its assembly from partially refolded domains as compared to assembly of native domains[81].

We also addressed the previously unexplored basis of long-range dynamic inter-domain stabilization of TMD1/2 and NBD1/2 with the 6SS-suppressor (Fig. 4)[51] and correctors (VX-661/VX-809 and/or VX445)[12]. Although 6SS mutations lack a universal stabilizing effect on the fast atomic motions (see Supplementary Figs. 3d and 5a), the NBD1-6SS stabilized allosterically the fast atomic motions and backbone NHs dynamics in CFTR, which translated into favorably changes in folding energy and unfolding activation energy of TMD1/2 peptides in CFTR-6SS (Fig. 4b, c). Conversely, CFTR was globally destabilized upon uncoupling the CL1- or CL4-NBD1 interfaces. Thus, uncoupling of CL4-NBD1 by the F508G mutation not only misfolded CFTR, but also attenuated the TMD1-dependent VX-809 rescue of P67L-CFTR misfolding without destabilizing the isolated NBD1 (Fig. 5b–d). Concordantly, VX-445, the TMD2 stabilizer, more effectively rescued the F508G-induced NBD1-TMD2 uncoupling than VX-809, while they additively restored the F508G-CFTR expression defect (Fig. 5h).

Likewise, the accumulation of native fold ensembles upon VX-809[15] or VX-445 binding to TMD1[15] or TMD1/TMD2[17,64], respectively, was able to partially suppress the P67L-CFTR cooperative domain misfolding by allosterically reversing the backbone NHs dynamic destabilization in multiple domains (Figs. 6b, c and 7b). A similar mechanism may be at work in the L206W-CFTR rescue[68]. Thus VX-809/VX-661 and VX-445 binding to the TMD1/2[15,17,64] can rewire the interdomain allosteric networks and tune the posttranslational folding landscape of CFTR domains (Figs. 4c and 6a), a mechanism that likely contributes to the basis for the considerable rescue of CFTR with ΔF508 and several rare misfolding mutations[11,12].

The HDX-MS studies on purified full-length CFTR variants have the limitation that the channel was reconstituted in GDN micelles. This may alter the channel microenvironment and influence the inter-domain coupling dynamics/energetics in response to mutations and pharmaco-chaperone binding. The second limitation of our work stems from relying on the 6SS-stabilized variant of the P67L-CFTR folding mutant. Although P67L-CFTR-6SS preserved its corrector susceptibility, we cannot rule out that the 6SS-suppressor partially masked the P67L-induced CFTR conformational defect, leading to the underestimation of VX-445 and VX-809 rescue effects.

Collectively, our results provide a plausible explanation for the prevalence and severity (penetrance) of point mutations throughout CFTR and other ABCC-transporters, since both primary domain interface uncoupling or structural defects can interfere with the long-range interdomain communications of both folding intermediates and the final fold[4–6,79,82]. The reported functional redundancy of CFTR inter-domain allosteric network is in line with the paradigm that misfolding by mutational disruption may be reversed by mono- or combinatorial-pharmacochaperones, relying on either orthosteric or allosteric stabilization of the primarily afflicted domain. This paradigm requires

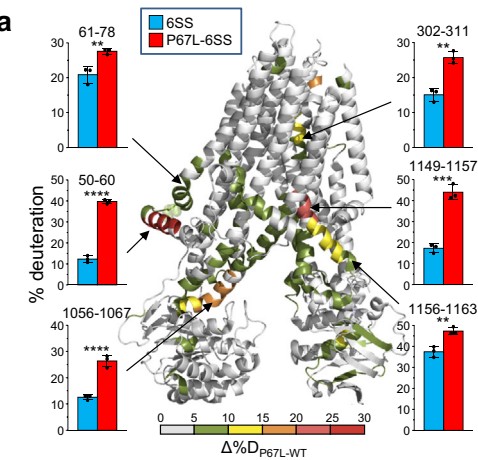

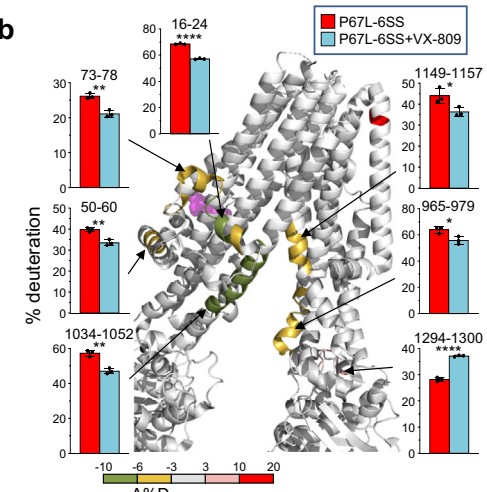

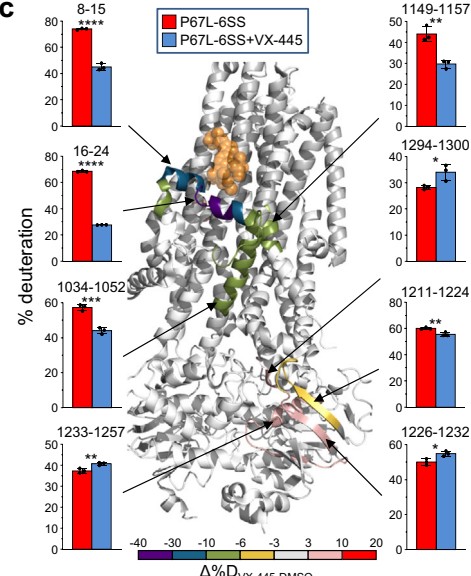

**Fig. 6 | VX-809 and VX-445 allosterically revert the global conformational dynamic perturbation of CFTR-6SS by the P67L CF-mutation. a** The P67L CF-mutation effect on the deuteration kinetics of purified WT-CFTR-6SS was determined by continuous deuterium labeling with the HDX-MS technique. The $\Delta\%$ $D_{(P67L-WT)}$ was depicted after 4 min deuterium (37 °C) with pseudo-colors projected on CFTR cryo-EM structure (PDB:5UAK). The bar plots illustrate the % of maximum deuteration. Means ± S.D., $n = 3$ technical replicates. The time course of deuteration uptake is included in Supplementary Fig. 12a. The effect of 30 μM VX-809 (**b**) or VX-445 (**c**) on the deuteration kinetics of purified P67L-CFTR-6SS was determined and visualized as in **a** after 4 min incubation. **b** $\Delta\%D_{P67L-6SS\ (VX-809\ -\ DMSO)}$ (PDB:7SVR). The VX-809 is shown as magenta spheres. **c** $\Delta\%D = \%D_{P67L-6SS(VX-445-DMSO)}$ (PDB:8EIQ). VX-445 is shown as yellow spheres Means ± S.D., $n = 3$ technical replicates. The time course of deuteration uptake is included in Supplementary Fig. 12b, c. Source data are provided as a Source Data file.

vector cloning and isolation *E. coli* StBL3 and all other plasmids preparation *E. coli* DH5α strains were utilized. Sequence information for primers used during cloning is available in Supplementary Data 1.

## Cell models
BHK-21 (ATCC #CCL-10) and CFBE41o- human CF bronchial epithelial cell lines were grown in DMEM/F-12 (5% fetal bovine serum; Invitrogen) and MEM (Invitrogen) supplemented with 10% FBS, 2 mM l-glutamine and 10 mM 4-(2-hydroxyethyl)-1-piperazineethanesulfonic acid (HEPES), respectively, at 37 °C under 5% $CO_2$. CFTR, MRP1 and ABCC6 variants were stably expressed (Supplementary Data 1) in BHK-21 as described[27]. Generation of CFBE41o- cell lines (a gift from D. Gruenert, University of California, San Francisco)[83] expressing the inducible WT- and 6SS-CFTR was done as described[84]. CFTR variants were induced for ≥3 days with 250 ng/ml doxycycline.

## Antibodies and reagents
Antibodies (Abs) were obtained from the following sources. Monoclonal mouse ani-HA Ab from BioLegend (1:2000, #901515), anti-CFTR Ab L12B4 (1:1000, recognizing residues 386–412 of NBD1, #MAB3484) and M3A7 (1:500, recognizing residues 1365–1395 at the C terminus of the NBD2, #05-583) were purchased from Millipore Bioscience Research Reagents (Temecula, CA) or provided by J. Riordan/M. Gentzsch laboratory (University of North Carolina, Chapel Hill, NC). 660 Ab (1:2000, recognizing NBD1) was provided by J. Riordan and the CF Foundation via the CFTR Antibodies Distribution Program. Mouse monoclonal anti-CFTR Ab MM13-4 (1:500, specific to the N-terminal a.a. 25-36, #05-581) was purchased from Millipore (Billerica, MA). Anti-human MRP1 monoclonal Abs: QCRL-1 (1:1000, specific for a.a. 918-924, #SC-18835) was purchased from Santa Cruz Biotechnology. MRPr1 (1:1000, specific for a.a. 238-247, #Ab3368, Abcam), 897.2 (1:1000, specific for a.a. 1316-1388[78]) and 643.4 (1:1000, specific for MRP1NBD2) were provided by X.B. Chang, Mayo Clinic College of Medicine (Scottsdale, AZ)[62]. Anti-ABCC6 Ab (1:1000, #D9D1F) was obtained from Cell Signaling Technology (Danvers, MA). Anti-Na/K-ATPase Ab (1:5000, #SC-48345) was purchased from Santa Cruz Biotechnology. Horseradish-peroxidase (HRP)-conjugated secondary Abs: sheep anti-mouse IgG (1:2000, GE Healthcare, #NXA931), goat anti-Rat IgG (1:2000, Jackson ImmunoResearch, #112-035-003), F(ab')₂ Fragment Goat anti-Mouse IgG (1:1000, Jackson ImmunoResearch, #115-036-003).

## Determination of CFTR plasma membrane (PM) expression by using ELISA
The PM expression of CFTR-3HA variants was measured by quantifying the specific binding of the mouse anti-HA primary Ab on live cells as described[53]. Briefly, BHK-21 cells grown in 24- or 48-well plates were chilled on ice, incubated with the primary antibody in phosphate-buffered saline (PBS) binding solution (PBS + +: PBS buffer

further validation to select possible pharmacotherapy for conformational diseases associated with other ABC-transporters.

## Methods
### Bacterial strains
For recombinant NBD1s expression *E. coli* BL21 (DE3) RIL was used. NelpII and Ulp1 were produced in *E. coli* BL21 (DE3). For lentiviral

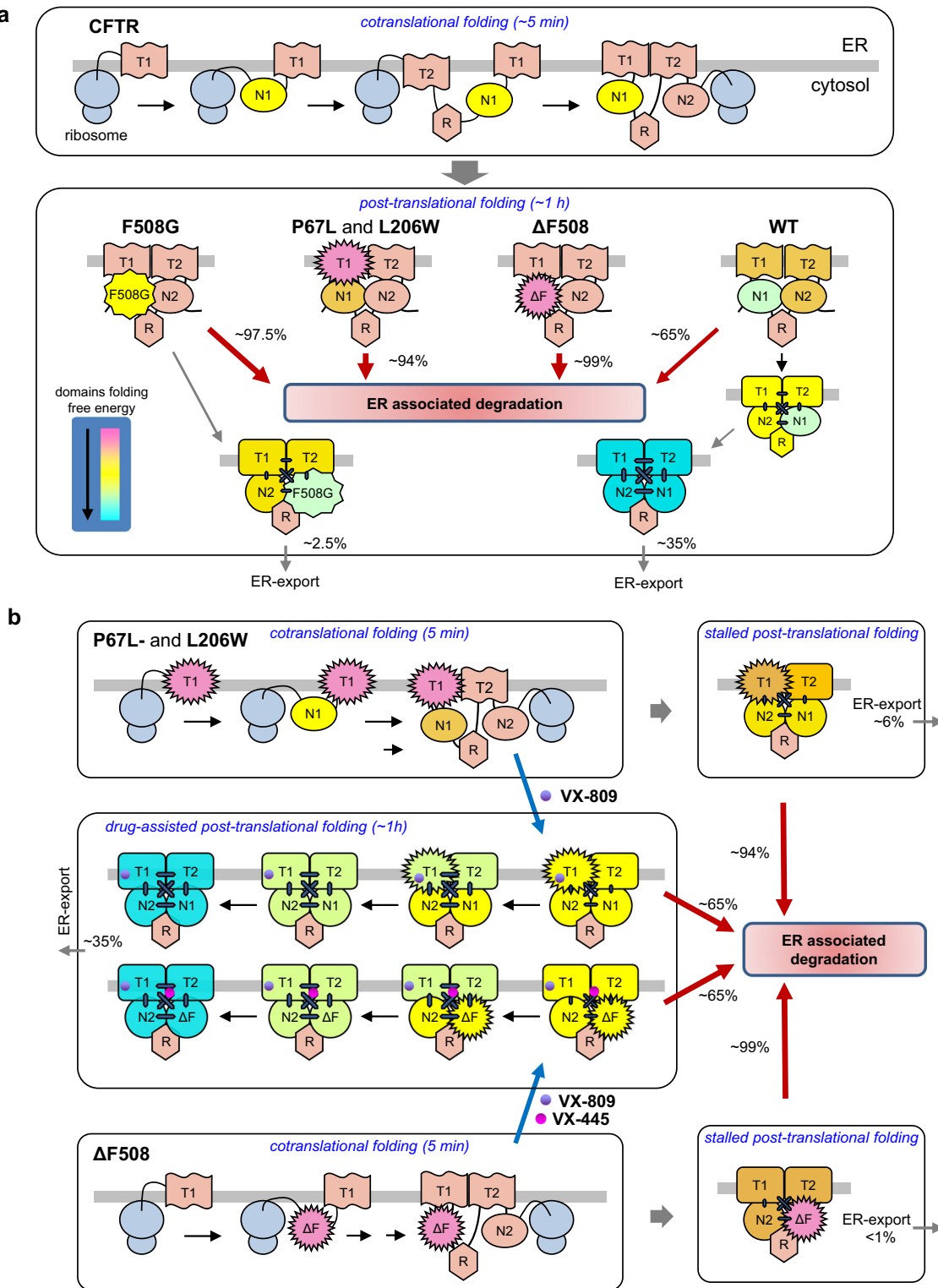

supplemented with 1 mM MgCl$_2$, 0.1 mM CaCl$_2$, and 0.5% bovine serum albumin) for 1 h at 0 °C. After washing thrice with PBS + +, the cells were incubated with HRP-conjugated goat anti-mouse AffiniPure F(ab')$_2$ secondary Ab (Jackson ImmunoResearch Laboratories, West Grove, PA) for 1 h at 0 °C. Specific anti-HA Ab binding was determined after correcting for background fluorescence signal, measured in the presence of non-specific primary IgG with Amplex Red substrate (Invitrogen) at 4 °C. The fluorescence signal was detected with a TECAN Infinite M1000 plate-reader[53] and normalized for cellular protein content using a BCA protein assay kit (Pierce). Results were expressed as percentage of control and/or relative to the WT-CFTR-3HA expression.

### Radioactive pulse-chase experiment

The radioactive pulse-chase technique was used to determine (i) the amount of newly synthesized nascent chain of CFTR variants during short pulse-labeling to minimize the post-translational maturation, and (ii) the posttranslational conformational maturation efficiency of newly synthesized core-glycosylated CFTR variants into the complex-glycosylated mature CFTR in the absence or presence of CFTR folding

**Fig. 7 | Working model for the CFTR co- and post-translational coupled-domain folding, its perturbation by point mutations, and pharmacological correction.**
**a** Biphasic conformational maturation of CFTR variants. The co-translational domain folding entails the formation of secondary structure elements in NBDs and TMDs, while completion of the channel tertiary structure, except that of the NBD1, is largely delayed by ribosome-nascent chain and chaperone interactions (not shown), the slow development of inter-domain couplings, and the CFTR domain-swapped structure during the post-translational phase. The relative folding free energy of CFTR, including its four conserved domains can reach their minima (color coded) and tertiary fold upon completing their slow post-translational cooperative domain folding. This may be facilitated by the faster development of the NBD1 near-native, molten globule-like conformation as a scaffold soon after its translation is completed[21,36], although domain packing further stabilizes the folding free energy of NBD1 in the context of native full-length CFTR ensembles (Fig. 4h). Besides the domains structural integrity, molecular chaperones are indispensable in CFTR posttranslational domain folding[41,74,75]. This chaperone-dependent post-translation domain folding (CD-PDF) phase can be disrupted by the NBD1-CL4 interface uncoupling with F508G or in combination with the NBD1 energetic destabilization by the ΔF508 mutations, as well as NBD1-CL1 destabilization by compromising long-range inter-domain allosteric communication at both the NHs dynamics and fast atomic motion level. **b** VX-809 or VX-445 correctors rescue the P67L-CFTR misfolding. The P67L (similarly to the L206W) CF mutation primarily destabilizes the TMD1 and secondarily compromises CFTR coupled domain folding. The mutants' cooperative domain misfolding (Fig. 5b) leads to ER-degradation and profound loss-of-expression. VX-809 or VX-809 + VX-445 pharmacochaperones rescue the fully translated folding-intermediates of L206W- and ΔF508-CFTR, conceivably via binding to the hydrophobic pockets formed by the TMH1, 2, 3, 6 and the Lasso region[15], or the Lasso helix1 and segments of TMH10/11 (TMD1 + TMD2)[17], respectively, present in the subpopulation of conformation ensembles. This interaction can shift the TMDs conformational equilibria towards their native conformers, which further facilitates the restoration of inter-domain coupling, cooperative domain-folding, and the accumulation of native-like conformational ensembles of P67L- and L206W-CFTR. The overall conformational maturation efficiency of pulse-labeled full-length folding intermediates is indicated in percentage based on the radioactive pulse-chase results.

corrector(s) during 3 h chase period. Pulse-chase experiments were performed as described previously[9,24]. Briefly, CFTR variants were pulse-labeled with 0.1–0.2 mCi/ml $^{35}$S-methionine and $^{35}$S-cysteine (Easy Tag Express, Perkin Elmer) in Cys- and Met-free DMEM (Gibco™ 21013024) medium for 10, 15, or 45 min as required and chased in full-medium complemented with 2 mM cysteine and methionine for 3 h. The folding efficiency of CFTR variants displaying <5% maturation efficiency was also measured after three-fold longer pulse-labeling period (45 min) and 2.5 h long chase. The corresponding pulse-labeled pool size was determined in parallel samples after 15 min radioactive pulse to allow comparison, while minimize the loss of radioactivity during the long-pulse.

CFTR was routinely immuno-precipitated with M3A7 and L12B4 anti-CFTR antibodies as described[24]. To improve the immunoprecipitation efficiency of CFTR-6SS variants that lack the NBD1 Ab epitope, C-terminal HA epitope-tagged variants were used as described[85]. These constructs were immunoisolated by anti-HA Ab precipitation. The $^{35}$S-methionine and $^{35}$S-cysteine incorporation into the core- and complex-glycosylated CFTR variants were visualized by fluorography and quantified using phosphorimage analysis on a Typhoon imaging platform (GE Healthcare). CFTR radioactivity was quantified after background subtraction, using the ImageJ software (National Institutes of Health and the Laboratory for Optical and Computational Instrumentation, University of Wisconsin).

## Q-PCR
Total RNA was extracted from BHK-21 cells using the miRNeasy kit (Qiagen) and reverse transcription of 1 µg total RNA was performed with the QuantiTect reverse transcription kit (Qiagen) following the manufacturer's recommendations. The abundance of transcripts was determined using the RT2 SYBR Green Rox Fast Mastermix (Qiagen) with an QuantStudio™ 7 Flex Real-Time PCR System (ThermoFisher Scientific). Data analysis was performed as described previously[84]. Sequence information for QPCR primers is available in Supplementary Data 1.

## Determination of CFTR and MRP1 in situ conformational stability using limited proteolysis and domain-specific immunoblotting
The conformational stability of full-length CFTR and MRP1, as well as their domains were probed by limited trypsinolysis or chymotrypsinolysis, using isolated microsomes from BHK-21 cells, stably expressing the respective ABCC transporters. Proteolytic fragments were visualized with immunoblotting, using domain-specific monoclonal antibodies as described[24,27,52,78,86].

Microsomes, containing the indicated CFTR or MRP1 variants in their ER, Golgi, endosome, and plasma membrane-enriched vesicles, were isolated by nitrogen cavitation and differential centrifugation from BHK-21 cells. To facilitate the selective detection of mature, complex-glycosylated form digestion pattern, cells were exposed to CHX (2.5 h) to degrade the core-glycosylated form by the ER PQC in some experiments. The P67L-, L206W- and ΔF508-CFTR were pharmacologically rescued by treating the cells with VX-809 (3 µM) for 24 h.

Isolated microsomes were resuspended in 10 mM HEPES, 250 mM sucrose, 1 mM EDTA, pH 7.6 and used either immediately or after snap-freezing in liquid nitrogen. Microsomes (1–2 mg/ml proteins) were digested at progressively increasing enzyme concentrations of TPCK-treated trypsin (Worthington Biochemical Corporation, Lakewood, NJ) or chymotrypsin (Sigma) for 15 min on ice. Proteolysis was terminated by the addition of 2 mg/ml soybean trypsin inhibitor (Sigma), trypsin-chymotrypsin inhibitor (Sigma) or 0.4 mM MgCl2, 0.4 mM PMSF, 2 µg/ml leupeptin and pepstatin. After 2x Laemmli sample buffer addition, digested microsomes were subjected to immunoblotting with CFTR domain-specific antibodies.

Immunoblots were visualized by enhanced chemiluminescence (ECL-Pico, ThermoFisher) and detected by the BioRad Chemidoc MP Imaging System. Signal intensities were quantified using Image Lab 6.1 (BioRad) or NIH Image 2.0 software (http://rsb.info.nih.gov/nih-image/) and corrected with background substruction.

## Expression and purification of NBD1
Recombinant NBD1s of CFTR, MRP1 and ABCC6 were isolated as described[9]. NBD1s were expressed in BL21-CodonPlus-RIL E. coli strain (Stratagene). The LB medium was supplemented with 50 µg/ml kanamycin, 40 µg/ml chloramphenicol, and 50 mM Tris-MOPS buffer with 0.5% glucose, pH 7.6, and inoculated with 1% (v/v) of an overnight E. coli culture and incubated at 30 °C with shaking until $A_{600}$ was ~0.3. The cultures were then cooled down to 12 °C (2 h) and induced overnight with 0.1 mM IPTG. Cells were homogenized and further lysed by sonication. SUMO-NBD1 was purified using Ni-affinity chromatography essentially as described[80], followed by the proteolytic cleavage of the $His_6$−SUMO-tag by using recombinant Ulp1 (construct provided by C. Lima) and gel filtration chromatography on a HiLoad 16/60 Superdex 200 column controlled by an AKTA pure FPLC system (GE Life Sciences). The protein was concentrated to -3 mg/ml in buffer containing 150 mM NaCl, 1 mM ATP, 2 mM MgCl2, 1 mM TCEP, 10% glycerol and 50 mM sodium phosphate buffer, pH 7.5. Protein concentrations were determined by the Bradford assay[87].

## Differential scanning fluorimetry (DSF)
The melting temperature ($T_m$) of purified NBD1s and CFTR variants was determined by DSF. DSF of NBD1 (7–12 µM) was measured in 150 mM NaCl, 5 mM MgCl2, 10 mM HEPES, and 2.5 mM ATP at pH 7.5 in the

presence of 4x Sypro Orange concentration, using a Stratagene Mx3005p (Agilent Technologies) or QuantStudio7 Flex (Life Technologies, Carlsbad, CA) qPCR instrument as described[8]. The $T_m$ of purified CFTRs (0.7 μM) was monitored in the presence of 1 μM Bodipy-FL-cysteine (Life Technologies) at 505 nm excitation and 513 nm emission to determine the accessibility of 19 Cys of CFTR[88]. The temperature ramp rate was 1 °C/min. Data were fitted to a Boltzmann sigmoid function by Prism (GraphPad Software, San Diego, CA) or Protein Thermal Shift™ Software (ThermoFisher Scientific). All measurements were performed in technical triplicates and repeated at least two or three times.

## Circular dichroism

Circular dichroism (CD) spectroscopy was performed using a Chirascan CD spectrometer (Applied Photophysics, Leatherhead, UK). CD scans were collected at 14 μM NBD1 in 150 mM NaCl, 1 mM ATP, 2 mM MgCl₂, and 50 mM sodium phosphate buffer at pH 7.5 between 260 and 190 nm using a 0.2 mm path-length cuvette at 20 °C. Data were collected in 0.5 nm increments at 0.5 s integration time and repeated three times with baseline correction as described[9].

## Folding free energy and the unfolding activation energy of isolated NBD1s

The folding free energies (ΔG°) of isolated WT and mutant NBD1s of CFTR, MRP1, and ABCC6 as a function of urea concentration were determined by using CD spectroscopy. The fractional unfolding of NBD1 was measured at increasing urea concentrations (0–6 M) at 20 °C, 22 °C, 23 °C, 28 °C, or 32 °C and calculated according to Eq. (1)

$$\Delta G^o = -RT\ln(K) \tag{1}$$

where K is the ratio of unfolded and folded WT and mutant NBD1 at different urea concentrations and temperatures. The ΔG° of NBD1 variants at 37 °C in water was determined based on linear extrapolation of the trendline, generated using the ΔG° of NBD1 in zero urea between 20 to 32 °C.

The unfolding activation energy (ΔG‡ᵤ) of NBD1 variants and their corresponding 95% CI at 37 °C were predicted using Minitab (https://www.minitab.com/en-us/products/minitab/). ΔG‡ᵤ was measured by determining the NBD1s unfolding rate constant in water ($k_u^{H2O}$) and $m_{ku}$ with CD spectroscopy as described[9,89] by linear extrapolation for water of unfolding rates at increasing urea concentration according to the equation of

$$\log(k_u) = \log\left(k_u^{H_2O}\right) + m_{k_u} \tag{2}$$

To obtain the unfolding rates of NBD1s at 37 °C, urea-induced unfolding rates ($k_u^{H_2O}$) measured at 16–30 °C were extrapolated for 37 °C. The ΔG‡ᵤ at 37 °C was calculated based on the relationship

$$\Delta G_u^{\#} = -RT\ln(k_u) \tag{3}$$

where $\Delta G_u^{\#}$ is the unfolding activation energy, R (1.986 cal/K mol) is the universal gas constant and T is the temperature in Kelvin (K).

## TNP-ATP binding assay

To estimate the ATP binding affinity of isolated NBDs, ATP was depleted from NBD1 samples by centrifugation and buffer exchange using Amicon Ultra-0.5 ml centrifugal filters with a M.W. cut-off of 5 kDa (Millipore, Billerica, MA). After ATP removal, a solution of 10 μM NBD1 in 20 mM Tris-Cl (pH 7.4), 2 mM TCEP, 2 mM MgCl₂ was prepared. Aliquots of this solution were equilibrated with 0–20 μM TNP-ATP (Sigma) in the absence or presence 0.5 mM ATP to record the specific and non-specific fluorescence signal intensities. Fluorescence

was measured at λ_ex = 420 nm and λ_em = 535 nm using a TECAN Infinite M1000 microplate reader in technical duplicates as described[9].

## HDX of isolated NBD1 variants

Continuous HDX experiments of isolated NBD1 variants were carried out as described[90]. Briefly, HDX was initiated by mixing NBD1 stock solution (stored on ice) into D₂O-based buffer (same components as H₂O-based NBD1 stock solution dissolved in D₂O) at 37 °C at a 1:9 dilution. The mixtures were incubated for the indicated time at 37 °C and quenched by adding the 3–5 μL aliquot of the reaction mixture into 10–12 μl of chilled quenching buffer (300 mM glycine-HCl including 8 M urea, pH 2.4). Quenched solutions were flash-frozen in methanol containing dry ice and stored at −80 °C. The samples solutions were thawed and immediately used for UHPLC-MS analysis. Deuterated NBD1 was digested using on-line immobilized pepsin column[91] at a flow rate of 30 μl/min for 1.5 min, and the resulting peptides were trapped on a C₁₈ column (Optimized Technologies, Oregon city, OR). Following desalting for 1.5 min at a flow rate of 200 μl/min, peptides were loaded onto a C₁₈ analytical column (1 mm i.d. × 50 mm, ThermoFisher Scientific) connected to an Agilent 1290 Infinity II UHPLC system. Peptides were separated using a 5–40% linear gradient of acetonitrile (ACN) containing 0.1% formic acid (FA) for 6 min at a flow rate of 65 μl/min. To minimize back-exchange, all columns (analytical, trapping, and immobilized pepsin columns), solvent delivery lines, sample loop and other accessories were placed in an ice bath. The C₁₈ column was directly connected to the electrospray ionization source of LTQ Orbitrap XL or Eclipse (ThermoFisher Scientific). Mass spectra of peptides were acquired in positive-ion mode for m/z 200–2000.

Identification of peptides was carried out by using tandem MS (MS/MS) analysis in data-dependent acquisition mode and collision-induced dissociation. All MS/MS spectra were analyzed using Proteome Discoverer 2.3 (ThermoFisher Scientific). The search results were manually inspected, and peptides that didn't yield quantifiable HDX data were removed from subsequent analysis. The percentage of deuteration level of the theoretical maximum (%D) as a function of labeling time was determined by using the HDExaminer 2.5 (Sierra Analytics, Modesto, CA) software without correction for back-exchange. Thus, %D reflects the relative exchange levels across the protein samples[92]. All HDX data collected are included in the Source Data file.

## Expression and purification of CFTR variants

Purification of full-length WT-, 6SS- and P67L-CFTR-6SS were performed as described[51]. Briefly, the HEK293 or CHO cells were resuspended in pre-chilled Buffer W (20 mM HEPES, pH 7.5, 150 mM NaCl, 10% glycerol, 3 mM MgCl₂, 2 mM ATP) containing 1 mM DTT and protease inhibitor cocktail (200 μM AEBSF, 5 μg/ml aprotinin, 300 μM benzamidine, 2.5 μg/ml each E64 and chymostatin, 10 mg/ml each leupeptin and pepstatin A). The mixture was homogenized using a Dounce homogenizer. Under gentle stirring, 0.5% n-dodecyl-β-D-maltopyranoside (DDM) and 0.05% cholesteryl hemisuccinate (CH) were added slowly from a 10% w/v stock. The mixture was stirred for 30 min at 4 °C. Insoluble materials were removed by centrifugation in a Ti-70 rotor (35,000 rpm, 30 min). The supernatant was mixed with pre-washed Strep-Tactin Sepharose resin (IBA Lifesciences) at -1 ml/billion cells concentration and placed on a rotating platform for 3 h, at 4 °C. Lysate and resin were packed into a 20-ml disposable column at 4 °C and unbound materials were removed. The resin was washed in 20-column volumes of Buffer W, containing 500 mM NaCl, 0.2 mM TCEP, and 0.06% digitonin, followed by 10 column volumes of Buffer W, containing 0.2 mM TCEP and 0.06% digitonin. The proteins were eluted by gravity using 5-column volumes of elution buffer (Buffer W, supplemented with 0.2 mM TCEP, 0.06% digitonin, and 4 mM d-des-thiobiotin). To improve reproducibility, 0.06% digitonin was replaced with 0.01% glyco-diosgenin (GDN101, Anatrace) in some experiments.

Pooled fractions were concentrated to 0.2 mg/ml, flash-frozen in small aliquots and stored at −80 °C until use.

To remove the purification tags, the purified proteins were thawed on ice and mixed with TEV protease at a ratio of 1:2 (TEV protease: CFTR). During tag removal, the proteins were dephosphorylated by λ-PPase, added at the beginning of tag removal at 2 U/µg CFTR concentration. After incubating on ice for 2–4 h with occasional mixing, CFTR was further purified on a Superose 6 Increase 10/300 GL column pre-equilibrated in Buffer SC (Buffer W containing 0.2 mM TCEP and 0.06% digitonin or 0.01% GDN101). The peak fractions that contained monomeric CFTR (elution volume -14.5 ml) were pooled and concentrated to -0.5–1 mg/ml. The purified CFTR was flash-frozen in small aliquots and stored at −80 °C until use.

On-resin CFTR tag-removal was performed following the solubilizing, binding, and washing procedures described above. After thorough washing, it was resuspended in the same column with 2 column volumes of Buffer SC. TEV protease and either PKA or λ-PPase were added in the same amounts as described above. The mixture was incubated overnight under gentle mixing to ensure complete tag removal. After sufficient incubation, the resin was allowed to settle in the column, and the flow-through was collected. The resin was washed in 2-column volumes of Buffer SC. The flow-through from the washing contained tag-cleaved CFTR was mixed with the first flow-through. The sample was concentrated 10- to 20-fold and purified on the Superose column as described above. CFTR variants were quantified by using SDS-PAGE followed by Coomassie staining.

## NADH-coupled ATPase assay

CFTR ATP-ase activity was monitored at 37 °C using an NADH-coupled ATPase assay[65]. The reaction solution contained 3 µg of dephosphorylated or phosphorylated CFTR variants in 20 mM HEPES/NaOH pH 7.5, 150 mM NaCl, 2 mM ATP, 10 mM Mg(acetate)$_2$, 1 mM DTT, 0.01% GDN101 (Anatrace), 16 units/ml pyruvate kinase, 22 U/ml lactate dehydrogenase (Sigma), 4 mM phosphoenolpyruvate, and 0.2 mM NADH. NADH consumption to regenerate ATP from ADP was monitored with technical duplicates in optically bottom clear 96 well plates (Corning, Corning, NY) at 340 nm absorption, using an TECAN Infinite M1000 microplate reader.

## HDX-MS of isolated CFTR variants

To optimize CFTR peptide sequence coverage, the following modifications were employed: (i) on-line tandem digestion with immobilized nepenthesin-II (nep2)[93] and pepsin, as well as (ii) on-line protease digestion at 15 or 18 °C controlled by a column cooler (CERA, Baldwin Park, CA). Nep-II was purified as previously described[93]. Briefly, inclusion bodies from overnight IPTG-induced 2xYT cultures at 30 °C were recovered by standard procedures. Inclusion bodies were solubilized in 50 mM Tris, 8 M urea, 1 mM EDTA, 500 mM NaCl and 300 mM β-mercaptoethanol (pH 11). Refolding was done by dialysis, first with 50 mM Tris pH 11 at 4 °C followed by 50 mM MOPS, 300 mM NaCl (pH 7) for 24 h (4 °C). pH was then corrected to 2.5 with 1 M glycine-HCl and a final dialysis step was carried out with 100 mM glycine-HCl, pH 2.5 at 37 °C for 2–3 days.

HDX was initiated by mixing CFTR stock solution (stored on ice) into D$_2$O-based buffer pre-incubated at 37 °C at a 3:7 dilution ratio. The mixtures were incubated for 10, 240, and 1200 s at 37 °C, and the HDX reaction was quenched by adding the 12 µl aliquot of the mixture into 3 µl of chilled quenching buffer (1 M glycine-HCl, 0.02% DDM, pH 2.4), resulting in 15 µl final volume. Quenched samples were stored at −80 °C until subsequent analysis. The on-line tandem Nep-II and pepsin digestions were carried out at 100 µl/min flow rate for 2 min at 18 °C, and desalting was performed at a 200 µl/min flow rate for 1 min. Digestion mixtures were separated by an Agilent 1290 Infinity II UHPLC system using a 5–40% liner gradient of ACN containing 0.1% FA for 11 min at 65 µl/min. MS measurements, peptide identification and data

analysis were performed as above. All HDX data collected are included in the Source Data file.

## HDX of purified P67L-CFTR-6SS in the absence and presence of VX-809 or VX-445 correctors

Before the HDX experiments, P67L-CFTR-6SS was pre-incubated at 25 °C for 5 min in the absence or presence of 30 µM VX-809 or VX-445 correctors. DMSO (0.1%) was included in all conditions. HDX was initiated by mixing pre-incubated P67-CFTR-6SS into D$_2$O-based buffer pre-incubated at 37 °C, at 1:9 dilution. The corrector concentration was kept at 30 µM during HDX incubation for 10 and 240 s at 37 °C. The HDX reaction was quenched by adding a 10 µl aliquot of the mixture into 5 µl of chilled quenching buffer (1 M glycine-HCl including 0.02% DDM, pH 2.4). Quenched samples were stored at −80 °C. The on-line pepsin digestion was carried at 60 µl/min flow rate for 1.5 min at 15 °C, and desalting was performed at a 200 µl/min flow rate for 1.5 min. Digestion mixtures were separated by an Agilent 1290 Infinity II UHPLC system using a 5–40% liner gradient of ACN containing 0.1% FA for 11 min at 65 µl/min. MS measurements were performed using an Orbitrap Eclipse tribrid mass spectrometer (ThermoFisher Scientific). Mass spectra of peptides were acquired in positive-ion mode for m/z 200–2000. Data analysis was carried out using HDExaminer 3.3 (Sierra Analytics). MS/MS spectra were analyzed using Proteome Discoverer 2.4 SP1 (ThermoFisher Scientific). All HDX data collected are included in Source Data file.

## Determination of peptides unfolding activation energy differences (ΔΔG$^{\#}_{u}$) of isolated WT- and 6SS-CFTR

Bimodal isotopic distributions (EX1 kinetics, Fig. 4d) were deconvoluted using the HX-Express 2[94] software to determine the fraction of folded ($F_f$) population. The natural logarithm of the $F_f$ was plotted as a function of D$_2$O exposure time ($t$). The rate constant of unfolding ($k_u$) was determined by fitting the data using Eq. (4):

$$\ln\left(F_f\right) = -k_u t \tag{4}$$

The Eyring Eq. (5) was used to calculate the unfolding activation energy difference (ΔΔG$^{\#}_{u(WT-6SS)}$, Fig. 4e) using the $k_u$ of WT- and ΔF508-NBD1 variants ($k_{u(WT)}$ and $k_{u(6SS)}$, respectively) according to Eq. (6):

$$k_u = \frac{k_B T}{h} e^{\frac{-\Delta G^{\#}_u}{RT}} \tag{5}$$

$$\Delta\Delta G^{\#}_{u(6SS-WT)} = -RT \times \ln(\frac{k_{u(6SS)}}{k_{u(WT)}}) \tag{6}$$

where $k_B$ is the Boltzmann's constant, h is the Planck's constant, R is the gas constant and T is the temperature in K.

## Determination of opening free energy difference (ΔΔG°) of peptides between WT and 6SS CFTR and between isolated CFTR and NBD1

To calculate the HDX rate constant ($k_{HX}$), the time-course deuteration was fitted to either mono- (8) or biexponential Eq. (9) by using the GraphPad Prism 6 software[95]:

$$D(t) = A_0 + A\left(1 - \exp[-k_{HX} t]\right) \tag{7}$$

$$D(t) = A_0 + A_1\left(1 - \exp[-k_{HX1} t]\right) + A_2\left(1 - \exp[-k_{HX2} t]\right) \tag{8}$$

where $D(t)$ is the extent of deuteration at D$_2$O incubation time $t$, and $A_0$ is the fraction of amide hydrogens that undergoes burst-phase labeling. In Eq. (8), $A$ is the number of hydrogens that undergoes

HDX with $k_{HX}$. In Eq. (9), $A_1$ and $A_2$ are the number of hydrogens that undergo HDX with $k_{HX1}$ (faster) and $k_{HX2}$ (slower), respectively. We assumed that the faster HDX with $k_{HX1}$ occurred at unstructured or loosely structured regions, and the slower HDX occurred at structured regions, such as α-helices or β-strands.

$k_{HX}$ in (7) or $k_{HX2}$ in (8) was used to determine the opening free energy difference between the WT- and 6SS-CFTR ($\Delta\Delta G^{o}_{(WT-6SS)}$) shown in the Fig. 4e, h by equation[96]:

$$\Delta\Delta G^{o}_{(WT-6SS)} = -RT\ln\left(\frac{k_{HX(WT)}}{k_{HX(6SS)}}\right) \qquad (9)$$

Given the thermal instability of the NBD1 variants during pre-incubation at 37 °C[9,49], the temperature-jump method was adapted for continuous HDX experiments. Therefore, the determined $\Delta\Delta G^{o}_{(WT-6SS)}$ are approximate values for peptides showing EX2 kinetics.

### Structural models
The ATP-bound, outwardly opened human CFTR structural model, determined by using cryo-EM[97] (PDB ID:6MSM, phosphorylated human CFTR), was subjected to loop modeling. We favored the ATP-bound CFTR structure, as the non-phosphorylated form may not reflect the CFTR physiological conformation, since it was determined in the absence of ATP. The composite ATP-binding site 1 in the presence of cytosolic ATP concentration remains predominantly dimerized[98]. Shorter missing regions (a.a. 410–434 and a.a. 890–899) were built using the standard loop modeling method of Modeler 9.23[99]. Mutations were generated using built-in tools of VMD[100]. The regulatory insertion (RI, a.a. 404–435) was deleted manually, and the break was sealed by loop modeling using Modeler.

The sequence alignments of MRP1_BOVINE with MRP1_HUMAN or MRP6_HUMAN (ABCC6) were generated using ClustalW[101]. The identity/similarity of these sequences were 91%/ 96% for hMRP1/bMRP1 and 46.6%/64.2% for hMRP1/hABCC6, respectively. The outward-facing, ATP-bound bovine MRP1 structure (PDBID: 6BHU)[102] was used as a template. Modeler was employed to generate the hMRP1 (a.a. 205–271, 311–867, and 943–1531; RMSD value to the OF template is 0.225 Å) and ABCC6 (a.a. 200–263, 297–851, and 929–1489; RMSD value to the OF template is 0.182 Å) homology models. The unresolved long loops were not modeled. However, we sealed the break between residues 271 and 311 of MRP1 for MD simulations using Modeler's loop modeling algorithm. One hundred models were prepared for each conformation, and the best model was selected using the DOPE score from Modeler.

### Molecular dynamics simulations
The system and input files for all steps (energy minimization, equilibration, and production run) were generated using the CHARMM-GUI web interface[103] by submitting the ATP-bound, outward-facing, full-length structures described in the previous paragraph. Full-length structures were oriented according to the OPM database[104], and 1-palmitoyl-2-oleoyl-sn-glycero-3-phosphocholine (POPC) was selected to build a membrane bilayer. The following additional options were adjusted: (i) terminal residues were patched by ACE (acetylated N-terminus) and CT3 (N-methylamide C-terminus), (ii) 150 mM KCl in TIP3 water was used, (iii) grid information for PME (Particle-Mesh Ewald) electrostatics was generated automatically, and (iv) a temperature of 310 K was set. The CHARMM36m force field was used, which also contains parameters for $Mg^{2+}$ and ATP. The structures were energy minimized using the steepest descent integrator (maximum number to integrate: 50,000 or converged when force is <1000 kJ/mol/nm). From the energy-minimized structures parallel equilibrium simulations were forked, followed by production runs for 0.5 μs. Nosé-Hoover thermostat and Parrinello-Rahman barostat were applied. Electrostatic interactions were calculated using the fast smooth PME algorithm[105] and the LINCS algorithm was used to constrain bonds[106].

Constant particle number, pressure, and temperature ensembles were used with a time step of 2 fs. Six and three simulations on full-length CFTR and MRP1 models, respectively, were performed by using GROMACS 2018.

### Network analysis of MD trajectories
A short time period (450–500 ns) from each trajectory was selected for network analysis based on ref. 107. The frames from these parts from every simulation a specific protein were merged and aligned. Each merged trajectory was used as input for calculating correlated motions using the generalized correlation measure by Lange and Grubmüller, based on mutual information[108] implemented in the *wordom* package (0.22-rc2)[109]. A network was built using amino acids as nodes and pairwise correlations ($C_{ij}$) as weighted edges. The weight was defined as -log($C_{ij}$) to have a small distance between highly coupled residues. Networks were analyzed and visualized using the Bio3D R package (2.4)[110] and VMD (1.9.3)[100]. All analyses regarding to dynamics were derived from trajectories of OF structures.

### Analysis of conformational flexibility
The fraction of native contacts and RMSF were calculated by custom Python script using the MDAnalysis package (2.4.2)[111]. Figures and plots were generated using VMD, PyMOL (2.6, Schrödinger, LLC) and the *Matplotib* (3.7.0) Python package[112].

### Statistical analysis
Results are presented as mean ± SEM or SD with the number of experiments indicated. Statistical analysis was performed using two-tailed Student's *t*-test with the means of at least three independent experiments and at least a 95% confidence interval was considered as significant. Abbreviations used for the *P*-values throughout the entire manuscript are denoted by asterisks as follows: <0.05 (*); < 0.01 (**); < 0.005 (***); < 0.0001 (****); and non-significant (ns).

### Reporting summary
Further information on research design is available in the Nature Portfolio Reporting Summary linked to this article.

## Data availability
All data presented in this study are available within the Figures and in the Supplementary Information, including the Source Data, are provided and linked to this article. The HDX-MS data for Figs. 2h, 4b, c, f, and 6a–c, as well as for Supplementary Fig. 2a–d, 10b–c, 1–1a-e, and 12a-c are available in the Source Data file. The HDX-MS data and raw MS spectra have been deposited to the ProteomeXchange Consortium via the PRIDE partner repository (dataset identifier PXD042481). The in silico data generated in this study have been deposited in the Zenodo database under accession code 8388593. Previously published structures used in this work are available from the PDB with the following accession codes: 6MSM (phosphorylated human CFTR), 5UAK (dephosphorylated, ATP-free human CFTR), 6BHU (ATP-bound, outward-facing bovine (MRP1), 7SVR (dephosphorylated human CFTR complexed with VX-809), 8EIQ (phosphorylated human ΔF508-CFTR complexed with Trikafta (VX-445, VX-661, VX-770)). Sequence information of sense and anti-sense oligonucleotides for CFTR, MRP1, ABCC6, and their NBD1s mutagenesis, as well as for CFTR QPCR is available in Supplementary Data 1. Source data are provided with this paper.

## Code availability
Python and Tcl scripts for molecular dynamics simulations and analysis along with raw trajectory data are deposited in a Zenodo repository [https://doi.org/10.5281/zenodo.8388593].

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

## Acknowledgements

We thank the computational resources to the Governmental Information-Technology Development Agency (Budapest, HU), Wigner Scientific Computing Laboratory (Budapest, HU), and Grubmüller laboratory at Max Planck Institute (Göttingen, Germany). We thank the late D Gruenert (The University of California San Francisco) for the parental CFBE41o- cell line, DC Schriemer (University of Calgary) for the expression construct of Nep-II (Nep2), JR Engen and L Konermann for their initial help in setting up the HDX-MS technique. We acknowledge the contribution of Ritu Radhawa in characterizing of the MRP1 interface mutation. We thank the following funding agencies for their support: Fonds de recherche Québec - Santé Postdoctoral Fellowship for (A.P.). Cystic Fibrosis Foundation grant HEGEDU20I0 (T.H.), National Research, Development and Innovation Office grant K127961, K137610 (T.H.), Cystic Fibrosis Foundation grant KAPPES21XX0 (J.K.), Cystic Fibrosis Foundation grant LUKACS20G0 (G.L.L.), Canadian Institute of Health Research grant 153095 (G.V., G.L.L.), Cystic Fibrosis Canada grant 609247 (G.L.L.), National Institute of Health grant R01DK075302 (G.L.L.), Canada Research Chair salary support grant (G.L.L.), Canada Foundation of Innovation John R. Evans Leaders Fund for infrastructure 39044 (G.L.L.).

## Author contributions

Conceptualization: N.S., H.X., A.R., Z.Y., H.Y., S.C., G.V., J.K., T.H., and G.L.L.; Methodology: N.S., H.X., A.R., Z.Y., H.Y., F.J., A.P., G.V., S.C., J.K., T.H., and G.L.L.; Investigation: N.S., H.X., A.R., Z.Y., H.Y., F.J., A.P., G.V., J.K., and T.H.; Visualization: N.S., H.X., A.R., H.Y., F.J., A.P., J.K., T.H., and G.L.L.; Funding acquisition: N.S., A.R., Z.Y., G.V., J.K., T.H., and G.L.L.; Project administration: Z.Y., A.R., J.K., and G.L.L.; Supervision: J.K., T.H., and G.L.L.; Writing—original draft: N.S., H.Y., A.R., T.H., and G.L.L.; Writing—review & editing: N.S., H.X., A.R., Z.Y., H.Y., F.J., A.P., G.V., S.C., J.K., T.H., and G.L.L.

## Competing interests

The authors declare no competing interests.
