## [Peer review file · Nature Communications]

REVIEWER COMMENTS

Reviewer #1 (Remarks to the Author):

In this study, Soya et al. investigated mainly the effects of the folding correctors VX-405 and VX-809 on CFTR and its variants by the pulse-chase analyses, molecular dynamics (MD) simulations, and hydrogen-deuterium-exchange mass spectrometries (HDX-MS), and so on. The results obtained here and their findings would provide important insights into the folding landscape of ABC C-subfamily transporters. However, some revisions should be required before the publication as follows.

Major points:

(1) The structure-function relationships (cotranslational or post-translational foldings) are the main point of this study. The first section of results (p.4, l.23-) describes the post-translational rescue of CFTR misfolding (L206W and Δ F508 mutants) by VX-809 and VX-445. Therefore, in Fig. 1A, not only the position of F508, but also at least the position of L206 should be depicted on the CFTR whole structure (the readers cannot know it until Fig. 5). As the result for the P67L mutant (not described in this section; a little confusing) is also shown in Fig. 1B, it might be helpful to depict the position of P67 as well. Minorly, in the pulse-chase result on Fig. 1C, the band figures are separated and the columns (-,-,gray) are duplicated, which should be explained in the legend or somewhere.

(2) Moreover, in p.5, l.5 or p.6, l.24, the efficiencies are described as “~0.4% to ~2% (Fig. 1B-C)” or “~1.8% to ~19% in the presence of the 2PT-suppressor (Fig. 2F)”, and so on. However, it is difficult to see the values only from the graphs. So, it is better to show the values of efficiencies (Figs. 1B, 1C, 1D, 2F, and so on) in the supporting material as tables. (It is difficult to judge whether the values such as efficiency are correct.) Similarly, several representative values should be labeled as the readers can understand the increase/decrease such as “by ~20-fold and ~3-fold, respectively (Fig. 2E)”.

(3) The second section of results (p.5, l.24-) focuses on the NBD1-CL4 interface (particularly F508-related). So, the CL1/CL4(TMD1/TMD2)-NBD1 structure and its close-up (insert) shown in Fig. 1A should be moved to Fig. 2 (where it is desirable to indicate “the cation- π interaction of the F508 side chain with the R1070(CL4)” (p.5, l.30-31)). Several residues in CL4 are discussed in this and the following sections.

(4) The final working model diagram (Fig. 7) will need to be significantly revised. The rescue by VX-809 in the L206W mutant as presented in Fig. 1 by the pulse-chase analysis (and Fig. 5) is ignored there (In Fig. 7B, only the P67L rescue is focused presently). Thus, it is recommended that Fig. 7A (top) focuses mainly on cotranslational and post-translational folding for WT, Δ F508, L206W, and P67L, and Fig. 7B (bottom or middle) focuses on their rescues by VX-445 and VX-809 (and the P67L details on the 7C/bottom?). In the revised Fig. 7B, it is desired to show the drug-rescue relationships such as the TMD1 mutations L206W and P67L are rescued by VX-809 (which binds to TMD1), and the NBD1 mutation Δ F508 (at the NBD1-CL4 interface) is rescued by both VX-809 and VX-445 (which binds between TMD1 and TMD2). The final figure is desired to summarize the research in general, and if P67L in the working model (Fig. 7B) is the main topic, the presentation of the results should be rearranged.

(5) In the structural modeling and simulations, the phosphorylated, ATP-bound (OF) CFTR structure (PDB ID:6MSM, p.27, l.1) was used as an initial conformation. From the description, the treatment of ATP molecules is unclarified. Why did the authors use the OF structure instead of IF structure? (despite showing the IF structure in most of the figures such as Figs. 1A and 6A/B/C). And, were the ATP molecules included or not? (There might be observed the relaxation from OF state (of the starting structure) in the ATP-free state, which would affect the results).

(6) Of course, it is difficult to simulate the folding of a large molecule like CFTR and their processes

after the translation, but it is possible to compare the apo and holo states of a drug. Here, it was a simulation in the apo state, and there are several previous studies on $\Delta F508$ such as McDonald et al., *Biomolecules* (2022) and Odera et al. *Biophys Physicobiol* (2018). It will be necessary to cite and discuss them.

(7) In the network analysis of MD trajectories, the final "short time period (450-500 ns)" (p.26, l.2) was used for each. Please add the root mean square deviation (RMSD) from the initial structure to the supporting material for each, and justify them to see if the analysis for this period is appropriate.

Minor points:

(8) There are some typos such as retore \rightarrow restore (in the title), Trikata \rightarrow Trikafta (p.3, l.17). It is necessary to check throughout the manuscript for those typos. Moreover, the "protein quality control" is abbreviated twice (as PQC) (p.3, l.10 and p.3, l.24). Please also check these points.

(9) Figs. 4E and 4H show some energetics, but the whole folding energy is not so clear. Therefore, it would be appropriate to correct the words "folding energy landscape" to "folding landscape" (p.2, l.14, p.4, l.9, p.8, l.12, and p.15, l.1).

(10) In the structural modeling of CFTR, two short missing regions (residues: 410-434 and 890-899) were once modeled by the loop modeling, then the "RI was deleted manually" (p.27, l.2-4). Please specify which parts were left missing and where they were modeled. Specific descriptions are also required for hMRP1 and hMRP6. (i.e., The modeled regions (residues) of CFTR, MRP1, (and/or MRP6/ABCC6) should be provided specifically in the Methods section.)

(11) In the "structural models" section (p.26, l.30-), MRP6 is used, but it would be better to unify them with ABCC6 after clarifying the relationship.

(12) In the homology modeling, both bottom-open and bottom-closed structures were used as templates for MRP1 and MRP6. In the MD simulations, which was the initial structure?

(13) Now, the human MRP1/MRP6 model structures are available from the AlphaFold database. Although the identity and similarity seem to be sufficiently/moderately high, please compare which one is more suitable for use as the initial structure of the simulations.

(14) It is recommended that the sequence information of CFTR (with TM1, TM2, ..., NBD1, NBD2, RI, 2PT/6SS sites, and mutational sites (L206, F508, R170)) are provided in Fig. 1 or the supporting materials, and that those (TMs, F713, F728, etc.) in MRP1 and/or ABCC6 are in the supporting (which is kind for the readers to understand the results).

Reviewer #3 (Remarks to the Author):

General Comments

In this complex paper the authors provide extensive evidence that CFTR folding, like that of other ABC transporters of the C subfamily, relies on many post-translational inter-domain interactions. The mRNA sequentially encodes transmembrane domain 1 (TMD1), nucleotide binding domain 1 (NBD1), regulatory domain, TMD2 and NBD2, and the ribosome translates the domains in sequence. Some co-translational folding and secondary structure formation occurs immediately. But following this initial step, each domain interacts with structural elements of other domains to attain a native conformation. This is certainly not surprising or novel. The CFTR structure includes complex interfaces with elements from multiple domains, one of which involves structural elements of cytosolic NBD1, of TMD1 (cytosolic loop 1, CL1) but also of TMD2 (the "domain-swapped" CL4). Thus it has been shown that NBD1 folding depends on the presence of natively folded TMD1 [1]. That NBD2 folding occurs post-

translationally has been shown by the Lukacs lab and others decades ago [2, 3]. That correctors can work post-translationally is also already known. For instance, heterologously-expressed, purified F508del-CFTR can be coaxed into a native WT-like structure by incubation with FDA-approved CFTR modulators, VX-809 and VX-445 [4].

However, this paper does provide strong evidence of the role played by inter-domain communication in native folding, in misfolding due to mutations, and in reversion of misfolding by pharmacological correctors. Noteworthy are elegant pulse-chase experiments with correctors added at different phases to different CFTR variants including F508del-CFTR, unambiguously demonstrating correction during post-translational folding. In addition, CFTR variants with the F508G, P67L, L206W, R170G mutations are used in several different assays to shed light on post-translational conformational maturation of WT, F508del-CFTR and other CF-causing CFTR variants. Limited proteolysis protocols, monitored by using domain-specific antibodies, demonstrate the effects of the NBD1 F508G mutation (and the homologous mutation in phylogenetic CFTR relative MRP1), on "allosteric" domain folding (i.e. folding of TMD1, TMD2 and NBD2). Similar experiments investigate how the TMD1 P67L and R170G mutations affect NBD1, TMD2 and NBD2 folding and how some of these effects can be reversed (i) by the 2PT and 6SS "suppressor" mutations (two sets of NBD1 deletions and mutations known to provide structural stability to NBD1 and to full-length CFTR), and (ii) by the FDA-approved small-molecule modulators, VX-809 and VX-445. Further investigations include extensive molecular dynamics simulations and analyses (on F508G-, R170G-, P67L-CFTR, with and without the 6SS suppressor mutations) as well as measurements of amide backbone dynamics, following the kinetics of hydrogen/deuterium exchange, in isolated NBD1 (of CFTR, MRP1 and ABCC6) and in full-length CFTR variants (WT-CFTR, 6SS-CFTR and P67L/6SS-CFTR).

Overall, this work presents a considerable body of evidence supporting the hypothesis that a network of domain interactions plays a crucial role in CFTR folding and biogenesis. I believe the results are informative and will have an impact on the field, solidifying interpretative frameworks on folding of polytopic proteins that are used in basic and translational science alike. Results largely support the conclusions, but the rationale behind some of the inferences made is not clearly presented (see below).

There is scope for manuscript improvement.

The main problem I see lies in the multitude of variants used, and in how they are used to answer important questions of basic biology or of translational importance. Throughout the paper, very general, sweeping conclusions are drawn, but how exactly these relate to the individual results presented and variants investigated is sometimes not clear. Results obtained with many different variants are presented in loosely related subsections. The argument for the use of each variant needs to be made more explicitly and clearly and clarified for each experiment. In particular:

1. Why are the F508G mutation effects reversed by the 2PT suppressor in some experiments (Fig. 2, Fig. S2) and by the 6SS suppressor in the others? What difference do the three extra mutations in 6SS make? It is stated that 2PT mutations are not located at NBD1 domain interfaces. However, S492P is at the interface with CL4 of TMD2.
2. The F508G mutation is introduced in the biochemical and MD investigations but not in the HDX experiments, In the HDX experiments focus is on the stabilization provided by the 6SS suppressor mutations (6 point mutations plus a 30-residue deletion). How is the detailed investigation of backbone amide dynamics and the changes due to the 6SS mutations informing us on the process of folding of WT-CFTR and F508G- (or Δ F508-) CFTR to a native conformation? The authors link the F508G- and 6SS-CFTR datasets during interpretation, but very often the logical steps of the arguments are not clearly made (see details below, Page10-12)
3. Is the P67L/6SS-CFTR a good model system to investigate mechanism of action of correctors on CF-causing variants in patients? The cellular/biochemical experiments (Fig.5) are performed on P67L-CFTR, but the HDX experiments are on P67L/6SS-CFTR (Fig. 6 – the figure legend indicating "WT" is a bit misleading, should be changed to 6SS-CFTR)

The second, related, aspect that needs improvement is clarity of language. The language is often very obscure, with multiple clauses in long, winding sentences. Specific details I found confusing/inaccurate are presented below. It is possible that my interpretation is incorrect. This strengthens the point that

more clarity is needed.

Finally, some figures appear to be (i) mislabelled (Figure S3-Panels A, B, C and D do not correspond to what is described in the figure legend; mention is made of $\Delta F508$ -CFTR in the legend but not in the panels), or (ii) not enough information is given to readers to understand what is presented in each panel (Fig. S4A), or (iii) text descriptions and figures do not appear to match (Fig. 4c, see Page 10, below; Figs S4C-E and S3A, see Page 12, below). This also contributes to making reading very arduous.

Minor comments and specific language suggestions – alterations from original shown in CAPITALS
Title: "Folding correctors can reStore CFTR post-translational folding landscape by allosteric domain-domain coupling"

Abstract: "CFTR post-translational folding intermediates, kinetically trapped by NBD1 or TMD1 cystic fibrosis (CF) mutations, are rescued by the allosteric or orthosteric binding of VX-809 or VX-445 folding correctorS, respectively."

"The pharmacophore-induced rigidification of destabilized backbone dynamics propagates across domain-interfaces As demonstrated by hydrogen-deuterium-exchange mass spectrometry in purified CFTR variants."

Page 3: "Eukaryotic ABC transporters contain at least two cytosolic nucleotide binding domains (NBD1,2) and TWO transmembrane domains (TMD1,2). THE LATTER interfacE the NBDs through four cytosolic loops (CLs) (Fig. 1A)."

"The loss-of-function of ABCC1 (MRP1) and ABCC6 are associated with enhanced susceptibility TO DRUG TOXICITY and ectopic mineralization, while the Cystic Fibrosis Transmembrane conductance Regulator (CFTR/ABCC7) ANION channel deficiency leads to cystic fibrosis (CF), the most prevalent lethal genetic disease in Caucasian populations."

"The DISCOVERY OF THE synergistic conformational rescue of the most common $\Delta F508$ (F508del) and rare CF mutations with combinations of suppressor mutations and/or small molecules targeting distinct structural defects..."

"It was proposed that VX-809 binds to and stabilizes newly translated TMD1 that cotranslationally attains its native fold, a working model that is consistent with the predominant cotranslational, domain-wise folding model of CFTR."

"IN AGREEMENT WITH the cooperative domain-folding model, the WT-like conformational poses of the drug binding sites in the cryo-EM structures SUGGEST THAT stabilization of CFTR native conformers can be accomplished post-translationally, following the development of inter-domain interactions, a prerequisite for the TMD1/2 fold."

Page 4: "Here, we posit THAT the folding energy landscape during ABC-transporter biogenesis requires multiple inter-domain (TMD1/NBD1/TMD2/NBD2) dynamics- and enthalpic-driven interactions, a process that may be facilitated, IN CFTR, by the predominant co-translational folding of NBD1 AND more delayed FOLDING OF other domains." Could it be explained here, what is meant by "dynamics-driven interactions"?

"By gaining insights ON the role of coupled domain-folding in ABCC-transporter conformational biogenesis and misfolding here we elucidated the mechanism of action of the landmark FDA-approved VX-445 and VX-809/VX-661 CFTR folding correctors."

"The pulse-chase technique, enables MONITORING OF the metabolic fate and folding efficiency of a small cohort of radioactively pulse-labelled MOLECULES, AS THEY DEVELOP FROM immature core-glycosylated (B-band, ~150kDa) CFTR nascent chains TO the mature complex-glycosylated (C-band, ~170 kDa) channels."

"We have shown that CONFORMATIONAL MATURATION OF pulse-labelled WT-CFTR and CFTR- Δ NBD2 folding intermediates takes approximately one hour, REGARDLESS of their N-linked complex-glycosylation state"

"we determined the folding efficiency OF L206W- and $\Delta F508$ -CFTR..."

Page 5: "Combination of VX-809 with the VX-445 corrector, that associates with TMD2 (TMH10-11) and the N-terminal Lasso segment of TMD1, restored WT-like folding efficiency (~36%) TO $\Delta F508$ -CFTR (Fig. 1B), confirming previous reports that monitored the mutant steady-state cellular accumulation"

"To restrict corrector binding EXCLUSIVELY TO THE co-translational PERIOD, cells were exposed to VX-445 and/or VX-809 ONLY during Met/Cys-depletion and pulse-labelling."

"In contrast, if drug(s) were present only during the chase, WHEN MOST post-translational folding OCCURS, the folding efficiency of Δ F508- and L206W-CFTR was restored to ~31% and ~40% (Fig. 1C-D)."

"These results suggest that post-translational conformational ensembles can effectively support the SELECTION OF drug-induced native conformers and THAT the TMDs are unable to attain their native fold co-translationally."

"Furthermore, TMD1/2 stabilization BY CORRECTORS allosterically suppresses Δ F508-CFTR cooperative domain-misfolding, initiated by Δ F508-NBD1 misfolding in trans, A MECHANISM THAT MIGHT UNDERLIE THE SYNERGISTIC EFFICACY of preclinical small-molecules that target distinct conformational defects of Δ F508-CFTR." This is unclear, and I am unsure I understood correctly.

Page 6: "...preserved the WT-like energetic and kinetic stability of F508G-NBD1, AS DEMONSTRATED BY domain values of melting temperature (T_m), folding free energy (ΔG_o), and unfolding activation energy (ΔG^\ddagger_u) COMPARABLE TO THOSE OF WT-CFTR (Fig. 2A, Fig. S1A)."

"The F508G MUTATION compromised CFTR cooperative domain-folding, evidenced by the augmented in situ trypsin and chymotrypsin susceptibilities of ALL FOUR CORE DOMAINS (TMD1/TMD2/NBD1/NBD2)"

"The trypsin sensitivity of the mature full-length F508G-CFTR and the tryptic F508G-NBD1 fragment was increased by ~20-fold and ~3-fold, respectively (Fig. 2E), CONSISTENT WITH disrupted post-translational domain-domain coupling."

"Importantly, the F508G-CFTR global misfolding was partially reversed by F508G-NBD1 hyperstabilization with the 2PT-suppressor mutations, LOCATED outside the domain interfaces (Δ RI+S492P+A534P+I539T, Table S1), which have been used to stabilize NBD1." But note that S492P is at the interface with ICL4.

"Metabolic turnover OF complex-glycosylated F508G-CFTR-2PT was REDUCED but not restored to WT-CFTR levels (Fig. S1C-D). THIS METRIC IS an indirect indicator of conformational stability of integral membrane proteins in post-Golgi compartments"

Page 7: "Although the R170G MUTATION has A limited effect on fast dynamical motions of TMD1 (data not shown), it SEVERELY AFFECTED cellular/PM expression and CAUSED TMD1/2 and NBD2 allosteric misfolding; BOTH EXPRESSION AND MISFOLDING were rescued by the 2PT-suppressor (Fig. 2C and Fig. S1F-G). This rescue, however, was compromised by the F508G-induced uncoupling of the NBD1-CL4(TMD2) interface (Fig. 2C, F, Fig. S1D, F)."

"Although the F508G MUTATION ONLY marginally increased the DEUTERATION of F508-loop peptides locally (Fig. 2H, Fig. S3D), it amplified the Ca atom thermal motions of the CL4 (TMD2, a.a.1064-1074) hydrophobic pocket (Fig. 3A)."

"These motions were relayed to the CL1, and NBD1/NBD2 interfaces, RESULTING IN a NBD2 α -subdomain destabilization"

"These perturbations likely contribute to the post-translational conformational maturation DEFECTS OF F508G-CFTR and TO the instability of its complex-glycosylated form" or "These perturbations likely contribute to IMPEDING the post-translational conformational maturation of F508G-CFTR and TO the instability of its complex-glycosylated form"

"The R170G mutation increased the centrality of a single residue, S168, while decreasing the centrality of residues in neighboring regions, REFLECTING?/DUE TO? the redundancy of allosteric signal propagation (Fig. S4A)."

"DESPITE the centrality of CL4, RESULTING IN AN INCREASED dynamic coupling via the F508 residue in the R170G-CFTR, this could only marginally suppress the channel global misfolding, marked by the allosteric conformational destabilization of CFTR domains and ~80% loss of the complex-glycosylated and PM-resident R170G-CFTR pools, BOTH reversed by the 2PT-suppressor (Fig. 2C and S1F-G)."

Page 8: "Introducing the 6SS-suppressor (2PT+M470V, S495P, R555K, Table S1) mutation in F508G-CFTR restored the CL4 dynamics, the native contacts at NBD1-NBD2, and NBD1-CL1, but not at the NBD1-CL4 interface, as expected (Fig. 3A-B, Fig. S3A)." – What is the significance of the very high RMSF at G1069 (Fig. S3A)?

"These results OBTAINED WITH 6SS-ELICITED RESCUE OF MOLECULAR DYNAMICS, IN CONJUNCTION

with 2PT-elicited biochemical rescue of F508GCFTR processing and stability, support the notion that modulation of the NBD1 inter-domain interface dynamics can rewire the posttranslational folding energy landscape of CFTR and partially restore F508G-CFTR cooperative domain folding."

"Despite the preserved backbone conformational dynamics of the mutant NBD1s, SUGGESTED BY their unaltered deuteration kinetics (Figs. S2B-C, Table S2), the EXPRESSION OF complex-glycosylated F713G-ABCC6 and F728G-MRP1 was diminished to 5-10% of their WT counterparts."

Page 9: "This was evidenced by the proteolytic digestion pattern of microsomes PROBES with domain specific ANTIBODIES of MRP1"

"Thus, uncoupling of NBD1-CL7(TMD2) interfaces RESULTS IN severe DEFECT(S) IN post-translational coupled domain-folding FOR MRP1, and by inference FOR ABCC6, without energetic destabilization of THE CORRESPONDING ISOLATED NBD1 SUBUNITS."

"Intramolecular allostery IS KNOWN to regulate the function of CFTR and MRP1. HOWEVER, the allosteric pathways that contribute to CFTR folding and stability, and are likely exploited by folding correctors, have YET TO BE MAPPED."

"we MEASURED the propagated changes IN BACKBONE AMIDE DYNAMICS CAUSED BY introducing the hyperstabilized 6SS-NBD1 INTO WT-CFTR."

"At the cellular level, the 6SS-suppressor significantly improved the conformational maturation and steady-STATE expression of the complex-glycosylated CFTR (Figs. S7A-B), while it did not alter the basal and cAMP-dependent protein kinase (PKA)-stimulated CFTR chloride transport activity in bronchial epithelia, based ON short circuit current measurements (Fig. S7C)." This result is puzzling to me: why was no increase in I_{sc} found, when presumably, the improved biogenesis would have delivered more channels to the membrane? 6SS mutations have been shown to slow down the channel gating kinetics, but without altering open probability or single-channel conductance [5]. Do the authors believe the improved biogenesis was not occurring in the bronchial epithelia? Why were changes in biogenesis and in ion channel function associated with the 6SS mutations measured in different systems (Fig. S7A,B vs. S7C)?

Page 10: Why are the peptides shown in Fig. 4C not the ones mentioned in the text? True, you provide data tables that would allow a reader to check, but text and main figures should provide an overview that most readers would be satisfied with.

"The attenuated side chain and backbone dynamics, imposed by the NBD1-6SS, at least partly accounts for the long-range thermodynamic and kinetic stabilization of the TMDs in the 6SS-CFTR (Figs. 3B and 4C)." I do not understand how the text relates to Figure 3B.

"The prominent inter-domain dynamic and energetic coupling between the NBD1 F508-loop and the CL4 interface by HDX provides a plausible explanation for the F508G deleterious effect on cooperative domain assembly during the post-translational conformational maturation of CFTR (Fig. 2D-F)." I do not see how the full-length HDX data is informing us on the F508 loop-CL4 interface in particular. Most of the data compares WT to the 6SS mutant. Only Fig. 4B, on the isolated NBD1 refers to peptides at the CL4 interface. But is that enough to support the sweeping statement? I would appreciate a reference to the specific supporting evidence.

Page 11: "TWO CF-CAUSING point mutations, P67L and L206W also elicit cooperative domain misfolding of CFTR, AS probed by limited trypsinolysis and domain specific immunoblotting in isolated microsomes (Fig.5B)"

"Notably, in contrast to weaker allosteric rescue of P67L and L206W variants by THE VX-445 pharmacophore relative to THAT OF VX-809, VX-445, by stabilizing TMH10/11 and the CL4, can exert A prominent rescue EFFECT on the Δ F508-CFTR by reinforcing the NBD1-CL4(TMD2) dynamic coupling and allosterically stabilizing the Δ F508-NBD1 primary conformational defect."

Page 12: "The P67L-INDUCED structural destabilization is indicated by the significantly reduced thermal stability of the CFTR-6SS as compared to the WT-CFTR, based on their T_m (Fig. S7F)." This sentence does not appear right. Do you mean "reduced thermal stability of the P67L-CFTR-6SS as compared to the CFTR-6SS, based on their T_m (Fig. S7F)."?

"The accelerated atomic motions at the F508-loop, CL4, and NBD2 elicited by the P67L MUTATION were also suppressed by the 6SS MUTATIONS based ON MD simulations (Fig. S4C-E), amplifying the dynamic coupling of intracellular parts of central TM helices AS WAS observed for the F508G-6SS mutant." This is all very unclear. If one looks at the CL4 panel of figure S4C, P67L appears to reduce

fluctuations at most residues compared to WT, especially at the C-terminal end, with the 6SS-suppressor mutations restoring the high amplitude fluctuations when added to the P67L background. The opposite happens for the F508 loop: P67L increases fluctuations, and the 6SS-suppressor mutations restore low-amplitude fluctuations. On the contrary, in Fig S3A, the F508G mutation increases the amplitude of fluctuations in the F508 loop, but adding the 6SS-suppressor mutations in the F508G background further increases fluctuation amplitude. The sweeping statement above seems to be overly simplistic and/or misleading.

"Jointly, our HDX data provide structural evidence for the dynamical basis of VX-809 and VX-445-dependent allosteric rescue mechanism of CF folding mutants in the TMD1, and by extrapolation in other CFTR domains" This sentence is quite vague. What exactly, in your data is evidence for a "dynamical basis" of the correctors' mechanism of action?

Page 13: "We propose a two-step folding model for the in vivo folding of ABCC transporters, incorporating previously unidentified critical inter-domain conformational coupling events. THE LATTER diminish the propensity of folding intermediates FALLING into kinetic traps and ensure the accumulation of native conformers at a biological relevant time scale (Fig. 7A). "

Very unclear – sentence rewritten: "As demonstrated by the NBD1 energetic stabilization (~5 kcal/mol) in the full-length CFTR (Fig. 4G-H), inter-domain allosteric communication is required to avoid kinetic traps and to achieve energetic stabilization of individual native-like domains. As is the case for other ABC transporters^{37,39,40}, correct folding of individual domains of CFTR requires the completion of translation, as well as the progression of post-translational folding."

Page 14: "Our working model suggests that CFTR native fold is largely acquired post-translationally. THIS INFERENCE is consistent with the newly synthesized (core-glycosylated) and pulse-labelled WT- and deltaF508-CFTR nascent chains HAVING comparable protease susceptibilities and degradation kinetics by the ER PQC"

"This model is in line with the indispensable role of INTERACTIONS WITH MOLECULAR CHAPERONES during the slow post-translational folding phase of ABCC-transporters."

"The post-translational domain-folding of CFTR is further reinforced by the observed dynamic/energetic reciprocal stabilization of NBD1 in the full-length CFTR (Fig. 4F-H), which is dependent on the post-translational conformational maturation of the TMD1/TMD2/NBD2 domains, WHICH IN TURN REQUIRES the reciprocal influence of NBD1 inter-domain coupling to generate the natively folded and ER-exit competent transporter ensemble (Fig. 7A)."

"The long-range dynamics inter-domain rearrangements were confirmed by the global stabilizing effect of NBD1 genetic suppressors and CFTR folding-correctors of the TMDs, respectively." What does "respectively" refer to?

Page 15: "We envision that a similar and common mechanism is responsible for improving the conformational biogenesis of F508G- and deltaF508-CFTR, and the P67L or L206W TMD1 mutant by the NBD1-6SS-induced allosteric stabilization" This seems to be more of an assumption than evidence-based. Can the authors spell out their arguments in favour of lumping lasso and F508-loop mutations? "The functional redundancy of CFTR inter-domain allosteric networks is in line with the paradigm that misfolding by mutational disruption could be reversed by mono- or combinatorial pharmacochaperones, relying on either orthosteric or allosteric stabilization of the primarily afflicted domain. THIS paradigm will NEED TO BE tested to select possible pharmacotherapies for conformational diseases associated with several other ABC-transporters."

Page 16: These paragraphs are repeated twice: "BHK-21 and CFBE410- human CF bronchial epithelial cell lines were grown in DMEM/F-12 (5% fetal bovine serum; Invitrogen) and in MEM (Invitrogen) supplemented with 10% FBS, 2 mM l glutamine and 10 mM 4-(2-hydroxyethyl)-1-piperazineethanesulfonic acid: HEPES), respectively, at 37°C under 5% CO₂. CFTR, MRP1 and ABCC6 variants were stably expressed in BHK-21 cells as described²⁸. Generation of CFBE410- cell lines (a gift from D. Gruenert, University of California, San Francisco)⁷⁵ that expresses the inducible P67L-6SS-CFTR was done as described⁷⁶. CFTR variants were induced for ≥ 3 days with 250 ng/ml doxycycline."

Page 17: "Ab binding was determined after correcting FOR background fluorescence signal"

Page 18: "The pulse-labeling efficiency WAS determined in parallel samples after 15 min radioactive pulse to allow comparison."

"Quantification of CFTR radioactivity FOLLOWING background SUBTRACTION"

Page 20: "Fractional unfolding OF NBD1 VARIANTS was determined at increasing UREA concentrations"

Page 27: "The INWARD-FACING MRP1 structure (PDBID: 5UJ9 95) and the OUTWARD-FACING, ATP-bound structure"

1. Kleizen, B., et al., Co-Translational Folding of the First Transmembrane Domain of ABC-Transporter CFTR is Supported by Assembly with the First Cytosolic Domain. *Journal of Molecular Biology*, 2021. 433(13): p. 166955.
2. Du, K., M. Sharma, and G.L. Lukacs, The Δ F508 cystic fibrosis mutation impairs domain-domain interactions and arrests post-translational folding of CFTR. *Nat Struct Mol Biol*, 2005. 12(1): p. 17-25.
3. Du, K. and G.L. Lukacs, Cooperative Assembly and Misfolding of CFTR Domains In Vivo. *Molecular Biology of the Cell*, 2009. 20(7): p. 1903-1915.
4. Fiedorczuk, K. and J. Chen, Molecular structures reveal synergistic rescue of Δ 508 CFTR by Trikafta modulators. *Science*, 2022. 378(6617): p. 284-290.
5. Yang, Z., et al., Structural stability of purified human CFTR is systematically improved by mutations in nucleotide binding domain 1. *Biochim Biophys Acta - Biomembranes*, 2018. 1860(5): p. 1193-1204.

Reviewer #4 (Remarks to the Author):

Manuscript NCOMMS-22-47099 by Lukacs and colleagues represents a Herculean effort to demonstrate the mechanisms by which CFTR folding correctors VX-809 and VX-445 -- components of the transformative therapeutic Ivacaftor -- have their biochemical effect. As is always that case for work from this lab, the biochemical studies are extremely well performed, and the results are quite noteworthy. The manuscript is quite a magnum opus, with a huge amount of data presented. The results will be of significance well beyond the CFTR field, and the authors do a pretty nice job of extending their findings to two other ABCC proteins closely related to CFTR.

The methodology is quite sound.

However, there are a number of disconcerting errors in the presentation of the results, and I find the interpretation not very convincingly explained. This really impacts only the writing, not the quality of the work itself.

Major concerns:

1. There are English errors throughout the manuscript. MUCH more importantly, one gets the impression that no one read the manuscript carefully before submission, given the retention of several important errors as listed below.
2. On page 3, line 20, and repeated on page 13, the authors incorrectly state that Trikafta is improving therapy of >90% of CF patients. Only about 90% of CF patients are genetically eligible for Trikafta, and a much smaller fraction are actually prescribed this drug. Furthermore, it does not work on a significant number of those eligible, and others who are eligible cannot tolerate it.
3. Very importantly, you never consider the impact of the R-domain on any of the experimental results in this paper. It is only mentioned at bottom of page 3, but is then included in the schema of Figure 7. Neither MRP1 nor ABCC6 have an R-domain.
4. Page 5, line 7. Here, you refer to data for F508del-CFTR being shown in Figure 1B. However, Fig. 1B has no data for this variant. This is an example of the surprising sloppiness in the presentation.

Surely, someone should have caught this?

5. Top of page 7, sentence ending on line 5. I do not agree with this conclusion, because the effect of 2PT on R107G/F508G is larger than the effect of 2PT on F508G alone (Fig. S1D). This is an example of where the authors need to use words to support their statements, a frequently missed effort.

6. Figure S2. There really is no interpretation of these data, other than a brief mention at the bottom of Page 6. Why include?

7. Figure S3A. There is no interpretation of these data. Why include it? Furthermore, the legend says that the data are means \pm SD for $n=3$, but there are no error bars. Finally, there is no explanation for what the "6SS" means in the "6SS/F508G" label in Fig. S3A, and you have not introduced the term until page 8 in the text.

8. Figure S3. Indeed, the legend does not match the figure, and part D is completely unrelated to the data shown there.

9. Page 8, line 5. The authors state that adding 6SS to F508G restored the CL4 dynamics. I see no justification for this statement in Fig. 3B.

10. Page 9, line 6. How is it shown that there were no "large differences"? What would represent a large difference?

11. Page 9, line 20. This long sentence makes no sense at all.

12. In the allosteric network analysis shown in Fig. S4A, why are there so very many differences between the two variants shown above and below in each pair? These differences are spread along the full 1,480-residue length of the protein. Given this observation, how does one interpret each deviation shown, to know if these are significant? Also, wouldn't it be useful to indicate where along each trace the noted mutation(s) reside? Note again that the "6SS" is not yet defined.

13. Figure 3. Don't we need to see the allosteric coupling figure (as in part C) for WT-6SS, too?

14. Figure 6. This is really confusing, and does not follow the presentation in the text on page 12. Are both "WT" and "P67L" variants bearing the 6SS stabilizations?

Minor concerns:

1. In the abstract, even the first sentence reads oddly. Instead of "are used" I would recommend "can be used".

2. In the abstract, please use "canonical ABC-transporters' core" to leave space for the incredible diversity of ABC proteins and their subunit composition.

3. Page 3, line 28. Not clear what is meant here.

4. Page 3, line 31. You have not yet mentioned the R-domain.

5. Figure S1D shows "R508G" instead of "F508G".

6. Figure S7A and B. What on Earth does "06" mean?

7. The authors probably need to provide a caveat (in Discussion) regarding the interpretation of the effects of either VX-809 or VX-445 on these peptides given the absence of membrane lipids.

8. Figure 7B. The second step is not possible, since the graphic implies that NBD1 is translated and folded before TMD1 is translated.

Response to Reviewers

We would like to thank all four Reviewers for their thorough and insightful comments that helped us to significantly improve our manuscript.

Reviewer #1 (Remarks to the Author):

We thank Reviewer #1 for the providing valuable feedback on our manuscript.

In this study, Soya et al. investigated mainly the effects of the folding correctors VX-405 and VX-809 on CFTR and its variants by the pulse-chase analyses, molecular dynamics (MD) simulations, and hydrogen-deuterium-exchange mass spectrometries (HDX-MS), and so on. The results obtained here and their findings would provide important insights into the folding landscape of ABC C-subfamily transporters. However, some revisions should be required before the publication as follows.

Major points:

(1) The structure-function relationships (cotranslational or posttranslational folding) are the main point of this study. The first section of results (p.4, l.23-) describes the post-translational rescue of CFTR misfolding (L206W and Δ F508 mutants) by VX-809 and VX-445. Therefore, in Fig. 1A, not only the position of F508, but also at least the position of L206 should be depicted on the CFTR whole structure (the readers cannot know it until Fig. 5). As the result for the P67L mutant (not described in this section; a little confusing) is also shown in Fig. 1B, it might be helpful to depict the position of P67 as well. Minorly, in the pulse-chase result on Fig. 1C, the band figures are separated and the columns (–,–,gray) are duplicated, which should be explained in the legend or somewhere.

The location of the L206 and P67 residues are indicated in Fig. 1A as recommended.

The illustration of the Δ F508-CFTR marginal ER maturation was taken from a representative independent experiment. This was indicated now in the legend of Fig.1.

The gray box marks that the radioactive-free chase period of the cells was not included to determine the pulse-labelling efficiency of CFTR. This explanation was included in the Fig.1C-D legends.

(2) Moreover, in p.5, l.5 or p.6, l.24, the efficiencies are described as “~0.4% to ~2% (Fig. 1B-C)” or “~1.8% to ~19% in the presence of the 2PT-suppressor (Fig. 2F)”, and so on. However, it is difficult to see the values only from the graphs. So, it is better to show the values of efficiencies (Figs. 1B, 1C, 1D, 2F, and so on) in the supporting material as tables. (It is difficult to judge whether the values such as efficiency are correct.) Similarly, several representative values should be labeled as the readers can understand the increase/decrease such as “by ~20-fold and ~3-fold, respectively (Fig. 2F)”.

As requested, the conformational maturation efficiencies of all CFTR variants are listed in Table S1 in the revised manuscript.

We also labeled the EC₅₀ values shifts in trypsin-sensitivity of the F508G-, F508-6SS- and WT-CFTR variants. “The trypsin-sensitivity of the mature full-length F508G-CFTR and the tryptic F508G-NBD1 fragment was increased by ~20-fold and ~3-fold, respectively, relative to that of the WT (Fig. 2F)”.

(3) The second section of results (p.5, l.24-) focuses on the NBD1-CL4 interface (particularly F508-related). So, the CL1/CL4(TMD1/TMD2)-NBD1 structure and its close-up (insert) shown in Fig. 1A should be moved to Fig. 2 (where it is desirable to indicate “the cation- π interaction of the F508 side chain with the R1070(CL4)” (p.5, l.30-31)). Several residues in CL4 are discussed in this and the following sections.

As recommended, the CL1/CL4(TMD1/TMD2)-NBD1 structure and its close-up (insert) shown in Fig. 1A has been moved to Fig. 2 as panel A.

(4) The final working model diagram (Fig. 7) will need to be significantly revised. The rescue by VX-809 in the L206W mutant as presented in Fig. 1 by the pulse-chase analysis (and Fig. 5) is ignored there (In Fig. 7B, only the P67L rescue is focused presently). Thus, it is recommended that Fig. 7A (top) focuses mainly on cotranslational and post-translational folding for WT, Δ F508, L206W, and P67L, and Fig. 7B (bottom or middle) focuses on their rescues by VX-445 and VX-809 (and the P67L details on the 7C/bottom?). In the revised Fig. 7B, it is desired to show the drug-rescue relationships such as the TMD1 mutations L206W and P67L are rescued by VX-809 (which binds to TMD1), and the NBD1 mutation Δ F508 (at the NBD1-CL4 interface) is rescued by both VX-809 and VX-445 (which binds between TMD1 and TMD2). The final figure is desired to summarize the research in general, and if P67L in the working model (Fig. 7B) is the main topic, the presentation of the results should be rearranged.

We revised Fig.7 according to the Reviewer #1 suggestions and included all major mutants (Δ F508, L206W, P67L, and F508G) and the WT-CFTR that are discussed in our manuscript. Panel A focuses on the co- and post-translational folding stages, illustrated by the estimated free folding energy gain of the CFTR variants and their partitioning between the ER secretion and ER associated degradation route.

Panel B primarily focuses on the folding rescue of the three CF causing mutations by the available FDA approved corrector drugs that were used in our experiments, including their productive association with CFTR folding intermediates, and their effect on the overall conformational maturation efficiency of the Δ F508, L206W, and P67L variants.

(5) In the structural modeling and simulations, the phosphorylated, ATP-bound (OF) CFTR structure (PDB ID:6MSM, p.27, l.1) was used as an initial conformation. From the description, the treatment of ATP molecules is unclarified. Why did the authors use the OF structure instead of IF structure? (despite showing the IF structure in most of the figures such as Figs. 1A and 6A/B/C). And, were the ATP molecules included or not? (There might be observed the relaxation from OF state (of the starting structure) in the ATP-free state, which would affect the results).

Previous observations indicated that the NBDs of the inward facing (IF) CFTR structure(s) exhibits a high level of rigid body motions (Corradi *et al.* 2018 Biophysical Journal 114: 1751–1754, Tordai *et al.* 2017 BBRC 491: 986-993). To minimize this phenomenon that would have also introduced some bias, we selected the outward facing (OF) structure for our simulations with bound ATP molecules. This approach was also favored by the observation that in the absence of ATP, the NBDs are prone to dissociate on the time scale of our MD simulations, leading to rigid body motions similar to that observed for the IF structure.

Importantly, the non-phosphorylated structure poorly mimics the CFTR physiological conformation, since it was determined in the absence of ATP. In contrast, under physiological conditions, the non-hydrolyzing, degenerate ATP Site-1 is practically constantly occupied by an ATP (Aleksandrov L *et al.* 2002 JBC 277:15419-25), thus at 1-2 mM cytosolic ATP concentration the NBD1/NBD2 remain associated most of the time. These notes were also inserted into our manuscript (p28, l.23).

(6) Of course, it is difficult to simulate the folding of a large molecule like CFTR and their processes after the translation, but it is possible to compare the apo and holo states of a drug. Here, it was a simulation in the apo state, and there are several previous studies on Δ F508 such as McDonald *et al.*, Biomolecules (2022) and Odera *et al.* Biophys Physicobiol (2018). It will be necessary to cite and discuss them.

After the VX-809/VX-661 and VX-445 CFTR binding sites were identified by Fiedorczuk and Chen in 2022 (Cell, Science), we continued our project towards comparing apo and holo states of various drugs. setting up, performing, and analysing these MD simulations requires sufficient time and resources. Thus, we feel that our ongoing simulations with small molecules are beyond the scope of this manuscript.

Although McDonald et al. used a great toolset, the Rosetta in their work, the combination of the applied tools for their specific task (predicting mutant CFTR ensembles and effects of drugs on them) are not expected to provide reliable results. Even the authors themselves mention several crucial issues in their Discussion section: 1) Conformation of residues in the F508 loop are different than expected. 2) Loop regions, including RI, are missing from their models. 3) “R1070W destabilizes the protein structure in our models but stabilized the NBD1/TMD2 interface suggesting that our models captured local energetic changes but failed to capture global changes”. (4) “Given the sequestered R1070W and the occluded VX-809 binding pocket in the inactive conformation both increased the overall scores of our models, we speculate our method is limited at handling perturbations to the inside of the CFTR structure, and perhaps better suited to modeling surface level changes, such as small molecules bound to exposed binding pockets.”

Odera et al. performed MD simulations on a homology model that displayed higher than ideal RMSD values (e.g.: 6-8 Å) (Fig. 1 of their paper).

In light of these weaknesses, we respectfully disagree with Reviewer #1 and do not think that inclusion of these papers would help the reader to better interpret our results.

(7) In the network analysis of MD trajectories, the final “short time period (450-500 ns)” (p.26, l.2) was used for each. Please add the root mean square deviation (RMSD) from the initial structure to the supporting material for each, and justify them to see if the analysis for this period is appropriate.

As requested, we analysed the RMSD throughout the simulation. These results are included in the supplementary material to confirm the selection of frames for analysis from the end of the simulations (450-500 ns) that reach near steady-state. Please see Supplementary Figure S3A.

Minor points:

(8) There are some typos such as retore → restore (in the title), Trikata → Trikafta (p.3, l.17). It is necessary to check throughout the manuscript for those typos. Moreover, the “protein quality control” is abbreviated twice (as PQC) (p.3, l.10 and p.3, l.24). Please also check these points.

We carefully proof-read the revised text and tried to correct all typographical and abbreviation errors.

(9) Figs. 4E and 4H show some energetics, but the whole folding energy is not so clear. Therefore, it would be appropriate to correct the words “folding energy landscape” to “folding landscape” (p.2, l.14, p.4, l.9, p.8, l.12, and p.15, l.1).

We changed the terminology to folding landscape.

(10) In the structural modeling of CFTR, two short missing regions (residues: 410-434 and 890-899) were once modeled by the loop modeling, then the “RI was deleted manually” (p.27, l.2-4). Please specify which parts were left missing and where they were modeled. Specific descriptions are also required for hMRP1 and hMRP6. (i.e., The modeled regions (residues) of CFTR, MRP1, (and/or MPR6/ABCC6) should be provided specifically in the Methods section.)

We inserted the requested details for the structural modelling in the revised Materials and Methods. Furthermore, the input files for homology modelling and the final structures were made available at <https://cftr.hegelab.org>.

(11) In the “structural models” section (p.26, l.30-), MRP6 is used, but it would be better to unify them with ABCC6 after clarifying the relationship.

In the revised manuscript we switched to the ABCC6 terminology.

(12) In the homology modeling, both bottom-open and bottom-closed structures were used as templates for MRP1 and MRP6. In the MD simulations, which was the initial structure?

Because of the rigid body motion of NBDs, we used the ATP-bound outward-facing [OF] (or inward-closed) conformations for our simulations. This was specified in the Methods of the revised manuscript (p.28).

(13) Now, the human MRP1/MRP6 model structures are available from the AlphaFold database. Although the identity and similarity seem to be sufficiently/moderately high, please compare which one is more suitable for use as the initial structure of the simulations.

We showed that AlphaFold performance for membrane protein structural predictions is also high and AF models may be superior to homology models (Hegedus et al. 2022 CMLS, 10.1007/s00018-021-04112-1). Nevertheless, considering that the AI-based structural models likely represent in a kinetically trapped conformation that may impede conformational fluctuations (Heo et al. 2021 Proteins, 10.1002/prot.26161). We did not use the AF-predicted MRP1 structure in our simulations since it represents an inward-facing conformation (<https://alphafold.ebi.ac.uk/entry/P33527>). A limitation of AlphaFold is that its predictions (randomly?) belong to a specific conformation that may not be the one required for the study (in our case it is not the outward-facing conformation). Because of the same issue, only the inward-facing ABCC6 AF-structure could be utilized. However, our direct comparison suggests that the difference between the AF-predicted structure and our homology model is small (Appendix Fig. 1.)

Appendix Fig. 1: Comparison of the AF-predicted and our homology models. The transmembrane regions of the AF model (blue, <https://alphafold.ebi.ac.uk/entry/O95255>) and our homology model (salmon and red) are depicted, since these domains exhibit low sequence conservation thus demonstrate the agreement of the two methods the best. Phe and Tyr residues are shown by sticks, which are easy to follow by eyes. The overall structures are highly similar, and differences are caused only by the slightly distinct openness towards the cytosol (bottom). Even the positioning of some of the side chains are also similar.

(14) It is recommended that the sequence information of CFTR (with TM1, TM2, NBD1, NBD2, RI, 2PT/6SS sites, and mutational sites (L206, F508, R170)) are provided in Fig. 1 or the supporting materials, and that those (TMs, F713, F728, etc.) in MRP1 and/or ABCC6 are in the supporting (which is kind for the readers to understand the results).

We inserted the requested information into the new Fig. S1A.

Reviewer #2 (Remarks to the Author):

We thank Reviewer #2 for the appreciative words and the constructive critique.

In this article, Soya and colleagues investigate the posttranslational folding landscape of ABCC transporters with a focus on CFTR and as a corollary, a rationale for the mode of action of the CFTR correctors VX809 and VX445. The work is thorough and of excellent quality, although very dense. The research spans from biophysical characterization to cellular studies, bringing together molecular level observations with cellular ones.

As my technical expertise was required, I can only approve of the HDX-MS data acquisition and analysis, these are of good quality and clearly support the findings. My only reservation stems from the fact the measurements were not done on biological replicates. Since the findings are cross validated with other techniques, I do not think it is a big issue, but I would refrain from doing any quantitative analysis such as the DG analysis (unfolding activation energy and opening free energy) if it is not reproduced. The results are convincing per se. (fig 4, D, E, G, H) and the DDG analysis can be removed without undermining the findings. If not, at least a word of caution about their usefulness in the context of a lack of biological replicates should be stated.

For HDX-MS analysis of all proteins (WT-CFTR, P67L-CFTR, WT-NBD1, WT-NBD1-6SS, except the WT-CFTR-6SS, were performed at least on two or three independently purified protein preparations. As suggested, we repeated the HDX kinetic analysis on a newly isolated WT-CFTR-6SS protein preparation to increase the number of biological replicate and strengthen the quantitative aspect of the propagated energetic inter-domain stabilization by the 6SS-suppressor in CFTR. The new HDX-MS results were combined with our previous data and all unfolding activation energy and opening free energy values of TMD1 and TMD2 peptides were recalculated and included in the revised manuscript (Fig. 4E). The primary data set for the WT-CFTR-6SS has also been updated (Table S4) and uploaded to the public database (ProteomeXchange Consortium via the PRIDE partner repository. The dataset identifier: PXD042481).

As a minor comment, it seems to me that the results section was re-shuffled without checking for consistency because I was often confused by statements that were missing introductory sentences (ex: mention of deuteration in the results section about MD while having not introduced HDX-MS yet). In general, one has to jump back and forth between figures and that complicates the reading. Besides this comment on the format, the content is of good quality.

We apologize for these oversights. In the revised version, we tried to minimize the back and forth jumping between figures, wherever it was possible. We moved the introduction of the HDX-MS methodology to Fig.2I, preceding the MD simulation studies.

Other minor comments – I found a few typos and inconsistencies

Typo in the title: “Folding correctors can retore CFTR post-translational folding landscape by allosteric domain-domain coupling.

p.3 Line 17: trikata

p4 line 30. First mention of mutant L206W but no explanation regarding the mutation.

P7 line 13 : deuteration? I guess it’s a leftover from previous version because at this stage deuteration has not yet been mentioned.

Fig 1B does not match with the main text description – it shows results for mutant P67L that was not introduced

P8 Line 27 deuteration

These typos and inconsistencies have been corrected.

Reviewer #3 (Remarks to the Author):

We thank Reviewer #3 for appraising our study, the exceptionally thorough edits, as well as providing invaluable scientific critic that helped to improve our manuscript.

General Comments

In this complex paper the authors provide extensive evidence that CFTR folding, like that of other ABC transporters of the C subfamily, relies on many post-translational inter-domain interactions. The mRNA sequentially encodes transmembrane domain 1 (TMD1), nucleotide binding domain 1 (NBD1), regulatory domain, TMD2 and NBD2, and the ribosome translates the domains in sequence. Some co-translational folding and secondary structure formation occurs immediately. But following this initial step, each domain interacts with structural elements of other domains to attain a native conformation.

This is certainly not surprising or novel. The CFTR structure includes complex interfaces with elements from multiple domains, one of which involves structural elements of cytosolic NBD1, of TMD1 (cytosolic loop 1, CL1) but also of TMD2 (the "domain-swapped" CL4). Thus, it has been shown that NBD1 folding depends on the presence of natively folded TMD1 [1]. That NBD2 folding occurs post-translationally has been shown by the Lukacs lab and others decades ago [2, 3]. That correctors can work post-translationally is also already known. For instance, heterologously-expressed, purified F508del-CFTR can be coaxed into a native WT-like structure by incubation with FDA-approved CFTR modulators, VX-809 and VX-445 [4].

However, this paper does provide strong evidence of the role played by inter-domain communication in native folding, in misfolding due to mutations, and in reversion of misfolding by pharmacological correctors. Noteworthy are elegant pulse-chase experiments with correctors added at different phases to different CFTR variants including F508del-CFTR, unambiguously demonstrating correction during post-translational folding. In addition, CFTR variants with the F508G, P67L, L206W, R170G mutations are used in several different assays to shed light on post-translational conformational maturation of WT, F508del-CFTR and other CF-causing CFTR variants. Limited proteolysis protocols, monitored by using domain-specific antibodies, demonstrate the effects of the NBD1 F508G mutation (and the homologous mutation in phylogenetic CFTR relative MRP1), on "allosteric" domain folding (i.e. folding of TMD1, TMD2 and NBD2). Similar experiments investigate how the TMD1 P67L and R170G mutations affect NBD1, TMD2 and NBD2 folding and how some of these effects can be reversed (i) by the 2PT and 6SS "suppressor" mutations (two sets of NBD1 deletions and mutations known to provide structural stability to NBD1 and to full-length CFTR), and (ii) by the FDA-approved small-molecule modulators, VX-809 and VX-445. Further investigations include extensive molecular dynamics simulations and analyses (on F508G-, R170G-, P67L-CFTR, with and without the 6SS suppressor mutations) as well as measurements of amide backbone dynamics, following the kinetics of hydrogen/deuterium exchange, in isolated NBD1 (of CFTR, MRP1 and ABCC6) and in full-length CFTR variants (WT-CFTR, 6SS-CFTR and P67L/6SS-CFTR).

Overall, this work presents a considerable body of evidence supporting the hypothesis that a network of domain interactions plays a crucial role in CFTR folding and biogenesis. I believe the results are informative and will have an impact on the field, solidifying interpretative frameworks on folding of polytopic proteins that are used in basic and translational science alike. Results largely support the conclusions, but the rationale behind some of the inferences made is not clearly presented (see below).

There is scope for manuscript improvement.

The main problem I see lies in the multitude of variants used, and in how they are used to answer important questions of basic biology or of translational importance. Throughout the paper, very general, sweeping conclusions are drawn, but how exactly these relate to the individual results presented and variants investigated is sometimes not clear. Results obtained with many different variants are presented in loosely related subsections. The argument for the use of each variant needs to be made more explicitly and clearly and clarified for each experiment. In particular:

1. Why are the F508G mutation effects reversed by the 2PT suppressor in some experiments (Fig. 2, Fig. S2) and by the 6SS suppressor in the others? What difference do the three extra mutations in 6SS make?

It is stated that 2PT mutations are not located at NBD1 domain interfaces. However, S492P is at the interface with CL4 of TMD2.

The information that S492P is located at the at the NBD1 domain interface was included in the revised manuscript.

The impact of second site suppressor mutations has been extensively studied on the stabilization of the Δ F508-NBD1 as well as the Δ F508-CFTR. These studies concluded that the Δ F508-CFTR conformational maturation and thee rescued form expression can be improved by various combinations of suppressor mutations proportionally with the NBD1 thermal stabilization and largely independently of the specific suppressor mutation (Cell 2012 148(1-2):164-74, Cell 2012 148(1-2):150-63), J Mol Biol 2015 427(1):106-20, Biochim Biophys Acta Biomembr 2018 1860(5):1193-1204.

Initially we employed the 2PT-suppressors as they increased the melting temperature (T_m) of the WT-NBD1 (39°C) and Δ F508-NBD1 (32°C) to \sim 62°C (BBA Biomembr 2018 1860:1193-1204). The 6SS-suppressors, incorporating three additional second site suppressors, further elevated the WT-NBD1 T_m to \sim 74°C (BBA Biomembr 2018 1860:1193-1204). We confirmed that 2PT-suppressor increased the T_m of the WT- and F508G-NBD1 by \sim 18°C (Fig.2B), which was sufficient to partially restore the cellular and molecular consequences of the NBD1-TMD1 or NBD1-TMD2 interface uncoupling in CFTR (Fig.2C-G, Fig. S2D and E). Based on published data and preliminary experiments we expected that the 6SS-suppressors have a more pronounced rescue effect than the 2PT on the mature F508G-CFTR expression in line with augmented thermal stabilization of the NBD1-6SS relative to the NBD1-2PT. This was indeed the case (Appendix Fig.2).

Appendix Fig.2 Misfolding of the F508G-CFTR can be rescued by both 2PT- and 6SS-suppressor mutations. The indicated CFTR-3HA variants were expressed stably in BHK-21 cell and immunoblotted with anti-HA antibody. Na/K-ATPase was used as loading control. The amount of cell lysate was adjusted as indicated to facilitate the visualization of the CFTR variants expression relative to that of the WT-CFTR (left panels). The amount of the complex-glycosylated CFTR variants (C-band) was quantified by densitometry and expressed as the percentage of the WT-CFTR (right panel). Means \pm S.E.M., n=3-5

We choose the 6SS-suppressor in subsequent studies to maximize its biological effect: i) in MD simulations and mapping the allosteric networks that responded to the F508G-NBD1 stabilization in CFTR, ii) on improving the P67L-CFTR protein production, and iii) on augmenting the impact of the NBD1 stabilization in the WT-CFTR conformational dynamics in HDX-MS experiments.

Thus, the *in vivo* results obtained by the 2PT- and 6SS-suppressor in rescuing the cooperative F508G-CFTR misfolding are qualitatively concur and support our conclusion. Expanding our preliminary data to a complete cellular and molecular analysis of the F508G-CFTR-6SS, we think is beyond the scope of this study. This would require significant additional efforts and resources and the results would provide limited novel insights into the allosteric domain (mis)folding mechanism of CFTR.

2. The F508G mutation is introduced in the biochemical and MD investigations but not in the HDX experiments, In the HDX experiments focus is on the stabilization provided by the 6SS suppressor mutations (6 point mutations plus a 30-residue deletion). How is the detailed investigation of backbone amide dynamics and the changes due to the 6SS mutations informing us on the process of folding of WT-CFTR and F508G- (or Δ F508-) CFTR to a native conformation? The authors link the F508G- and 6SS-CFTR datasets during interpretation, but very often the logical steps of the arguments are not clearly made (see details below, Page10-12.

We were unable to perform HDX-MS studies on the F508G-CFTR or the F508G-CFTR-2PT as their cellular expression was insufficient to purify the required amount channel. The expression of these CFTR variants was \sim 18% and \sim 35% of the WT-CFTR, respectively (Fig.2D, right panel).

We concur with Reviewer #3 comments and faced the limitations that individual technique does not permit to run a panel of experiments under identical conditions. To emphasize that the foundation of the folding/misfolding working model of CFTR is built on the complementarity of results obtained by different approaches, we provide an overview of our methods in the Introduction that generated the most salient results for our working model:

“Our working model integrates results that have been obtained on three ABCC-transporters, using structural perturbation techniques by mutations and pharmacophores application at four different levels. i) At the cellular level we monitored the transporters biosynthetic processing, expression, and metabolic stability, as well as the conformational dynamics at domain level. ii) At the isolated NBD1 level we determined the domains' thermal stability and backbone NHs conformational dynamics. iii-iv) In the full-length WT and mutant CFTR variants, changes in the isolated channel conformational dynamics were assessed both at the backbone NHs dynamics and fast atomic motions levels by using HDX-MS (hydrogen-deuterium exchange with mass spectrometry) and molecular dynamics (MD) simulations, respectively.”

We would like to emphasize that the F508G mutation failed to alter the folding energetics and the HDX kinetics of the isolated NBD1 (Fig. 2B and J, Fig. S2A), while it was sufficient to rearrange the interdomain allosteric networks by disrupting the NBD1-TMD2 (F508-CL4) coupling at the fast atomic motion level (e.g. Fig.3B-C). Conversely, the 2PT-suppressor could hyperstabilize the backbone NHs dynamics in the isolated F508G-NBD1 (Fig.2J, Fig.S2A) and, concomitantly, partially rescued the cellular folding defect by promoting the F508G-CFTR domains allosteric stabilization (Fig.2D-F). For the MD simulation we switched to the 6SS-suppressor to amplify the interdomain allosteric coupling effect of suppressors (Fig.3C-D and Fig.S4A-C).

The F508G mutation effects on the isolated NBD1 variants were examined by HDX-MS, CD spectroscopy and differential scanning fluorimetry (DSF) (Fig. 2B and J) as well as with MD simulations in the context of the full-length CFTR (Fig.3A-C). In case of the F508G- and P67L-CFTR mutations, we attempted to restore the core domains conformation dynamics by the 2PT and 6SS suppressor mutations, respectively. As a read out, we used limited proteolysis in concert with domain-specific immunoblotting of isolated microsomes (Fig.2E and 5B, Fig S1C and Fig. S5G), as well as metabolic pulse chase technique to monitor the post-translational domain folding propensity of the mutants (e.g.: Fig. 1B-D, Fig. 2G). Collectively, these techniques were able to uncover the misfolding of the TMD1, TMD2, and NBD2 in the F508G-CFTR, while the F508G-NBD1 backbone NHs conformational dynamics and folding energetics remained very similar to that of the WT-NBD1 (Fig.2B and J), but not at the side chain fast atomic motion level. These observations are consistent with our working model, proposing that conservation of the CL4-NBD1 hydrophobic interface coupling contribute to both energetic and fast inter-domain dynamic coupling of CFTR and, importantly, tuning the channel posttranslational folding landscape. Evidence suggests that a similar mechanism may be involved in the conserved CL7-NBD1 interface coupling in the ABCC6 and MRP1 transporters.

We introduced the isolated NBD1 to interrogate the effect of the F508G on the NBD1 conformational dynamics by HDX (Fig.2H and revised: 2J). We also documented the 2PT-suppressor effect on the NHs deuterium exchange kinetics of the F508G-NBD1 (Fig.2J), demonstrating that the 2PT impeded the backbone conformational dynamics of the F508G-NBD1 domain-domain interfaces. This effect likely contributes to the F508G-CFTR stabilization and rescued ER folding efficiency (Fig.2G). In addition, we showed that the 6SS-suppressors could impose a more pronounced rigidification on the backbone conformational dynamics of the NBD1 than the 2PT-suppressors (Fig.4A and data not shown) that significantly improved the post-translational domain folding.

We predicted that based on the near complete rescue effect of the Δ F508-CFTR by the 6SS (BBA Biomembr 2018 1860:1193-1204), we would experience a complete rescue of the cellular processing defect of the F508G-CFTR-6SS construct. Our preliminary studies indeed validated our prediction (Appendix Fig.2). Based on this prediction, we preferred to pursue our HDX studies on the disease-relevant and corrector-susceptible P67L-CFTR conformational dynamics. Importantly, the cellular phenotype of the P67L-CFTR could only be partially rescued by the 6SS-suppressors (Fig. 5H) and this modification preserved the P67L-CFTR-6SS susceptibility to VX-809 and VX-445 correction (Fig. 5H) which allowed us to confirm the drugs binding pose and uncover, at least in part, their

allosteric stabilizing effect on the backbone dynamics of interfacing domains. Nevertheless, we acknowledge that employing NBD1 hyperstabilization by the 6SS suppressor has some weaknesses, which are described in the Discussion.

To complement the above results and provide support for the stabilizing role of interdomain interface coupling in CFTR, three additional lines of investigations were pursued.

i) The 6SS-suppressors induced the propagated allosteric dynamic stabilization of the WT-CFTR-6SS, uncovered by HDX-MS, T_m , and protease susceptibility measurements (Fig. 4B, C, Fig. S7E), as well as the allosteric energetic stabilization of selected peptides in the TMD1 and TMD2 (Fig.4E), manifesting in improved folding efficiency of the WT-CFTR-6SS (Fig. S7B).

ii) Using a combination of techniques (limited proteolysis, pulse-chase experiment, HDX-MS) we showed that the P67L CF mutation at the N-terminal Lasso motif can allosterically destabilizes all core domains except the NBD1-6SS in the context of the CFTR-6SS in relation to the WT-CFTR-6SS that can be rescued by the folding correctors at multiple level.

iii) Notably, uncoupling of the NBD1-CL4 interface by the F508G mutation, prevented the allosteric rescue of the P67L- and L206W-CFTR-6SS with VX-809, underscoring the critical role of inter-domain coupling in both the propagated domain misfolding and its rescue.

In summary, we are convinced that the complementary of our approach, provide sufficient initial evidence to support our working model for the permissive role of interdomain allostery in the biogenesis of ABCC-transport and corrector-dependent rescue mechanism.

3. Is the P67L/6SS-CFTR a good model system to investigate mechanism of action of correctors on CF-causing variants in patients? The cellular/biochemical experiments (Fig.5) are performed on P67L-CFTR, but the HDX experiments are on P67L/6SS-CFTR (Fig. 6 – the figure legend indicating “WT” is a bit misleading, should be changed to 6SS-CFTR).

We do not think that the mechanism of corrector action can be generalized based on the results of our P67L/6SS-CFTR studies alone. Nevertheless, salient elements of our results obtained on the P67L-6SS- and P67L-CFTR jointly with earlier data (e.g. on the Δ F508-CFTR conformational rescue by combination of preclinical correctors acting synergistically on the core domains structural defects) strongly suggest that restoring inter-domain interface coupling by either in cis or in trans correctors can successfully restore the processing of a subset of class II CFTR folding mutants (JCI Insight 2020 Sep 17;5(18):e139983).

The WT-CFTR labelling was changed to WT-CFTR-6SS in the Fig.6 legend.

Please, note that the biochemical experiments were not only performed on the P67L-CFTR, but on the P67L-CFTR-6SS as well. The P67L-CFTR-6SS preserved its folding corrector susceptibility at a higher expression level (Fig.5C-E and H).

The second, related, aspect that needs improvement is clarity of language. The language is often very obscure, with multiple clauses in long, winding sentences. Specific details I found confusing/inaccurate are presented below. It is possible that my interpretation is incorrect. This strengthens the point that more clarity is needed.

Numerous changes were implemented to improve the accessibility of the text. All these changes in the manuscript have been tracked.

Finally, some figures appear to be:

i) mislabelled (Figure S3-Panels A, B, C and D do not correspond to what is described in the figure legend; mention is made of Δ F508-CFTR in the legend but not in the panels),

The mislabelling has been corrected.

(ii) not enough information is given to readers to understand what is presented in each panel (Fig. S4A)

We expanded the Fig.S4A legend, while respecting the word limitation.

(iii) text descriptions and figures do not appear to match (Fig. 4c, see Page 10, below; Figs S4C-E and S3A, see Page 12, below). This also contributes to making reading very arduous.

We apologize for these oversights. They have been corrected in the revised manuscript.

Minor comments and specific language suggestions – alterations from original shown in CAPITALS

We are grateful to Reviewer #3 for providing in depth recommendation for improving the readability of the manuscript and correcting the grammatical mistakes.

All suggested grammatical/languages corrections and suggestions were introduced in the revised version. Here, we address only the remaining minor questions.

Page 4: “Here, we posit THAT the folding energy landscape during ABC-transporter biogenesis requires multiple inter-domain (TMD1/NBD1/TMD2/NBD2) dynamics- and enthalpic-driven interactions, a process that may be facilitated, IN CFTR, by the predominant co-translational folding of NBD1 AND more delayed FOLDING OF other domains.”

Could it be explained here, what is meant by “dynamics-driven interactions”?

Electrostatic, van-der-Waals and polar interactions of amino acids confined to domain-domain interface can mutually modulate their fast atomic motions as well as the backbone dynamics. However, these dynamic changes are note restricted to the first shell of interacting amino acid but according to previous studies are prone to propagate beyond the second shell allosterically in most proteins (Curr Opin Struct Biol 2019 Feb;54:1-9).

The sentence has been replaced by:

We posit that the folding landscape biogenesis of ABC-transporters requires the progressive development of array of inter-domain (TMD1/NBD1/TMD2/NBD2) dynamics- and enthalpy-driven interface interactions in space and time. This process is conceivably facilitated by the NBD1 predominant co-translational folding^{35,36} in CFTR and more delayed folding of other domains.

Page 5:

“Furthermore, TMD1/2 stabilization BY CORRECTORS allosterically suppresses Δ F508-CFTR cooperative domain-misfolding, initiated by Δ F508-NBD1 misfolding in trans, A MECHANISM THAT MIGHT UNDERLIE THE SYNERGISTIC EFFICACY of preclinical small-molecules that target distinct conformational defects of Δ F508-CFTR.” This is unclear, and I am unsure I understood correctly.

These sentences were deleted from the Result and a revised version was included in the Discussion:

Page 16:Thus VX-809/VX-661 and VX-445 binding to the TMD1/2^{15,17,19} can rewire the interdomain allosteric networks and tune the post-translational folding landscape of CFTR domains (Figs. 3C, 4C, and 6A), a mechanism likely constitutes the basis for the considerable rescue of Δ F508-CFTR and several rare misfolding mutations^{11,12}.

Page 6:

“Importantly, the F508G-CFTR global misfolding was partially reversed by F508G-NBD1 hyperstabilization with the 2PT-suppressor mutations, LOCATED outside the domain interfaces (Δ RI+S492P+A534P+I539T, Table S1), which have been used to stabilize NBD1.” But note that S492P is at the interface with ICL4.

The statement has replaced by:

“To assess the possible role of dynamic coupling of F508G-NBD1 interfaces in F508G-CFTR folding, we hyperstabilized the F508G-NBD1 with the 2PT-suppressor mutations (Δ RI+S492P+A534P+I539T, Table S2). The 2PT-suppressors have been established to stabilize NBD1 variants and located outside the domain interfaces except the S492P.”

Page 7:

“The R170G mutation increased the centrality of a single residue, S168, while decreasing the centrality of residues in neighboring regions, REFLECTING/DUE TO the redundancy of allosteric signal propagation (Fig. S4A).”

“DESPITE the centrality of CL4, RESULTING IN AN INCREASED dynamic coupling via the F508 residue in the R170G-CFTR, this could only marginally suppress the channel global misfolding, marked by the allosteric conformational destabilization of CFTR domains and ~80% loss of the complex-glycosylated and PM-resident R170G-CFTR pools, BOTH reversed by the 2PT-suppressor (Fig. 2C and S1F-G).”

We changed these sentences as follows.

“The R170G mutation decreased the centrality of residues in its neighborhood, except of a single residue (S168), reflecting a decrease in the redundancy of allosteric signals propagation (Fig. S4A). Although the centrality of the CL4 ensured an augmented dynamic coupling via the F508 residue, this could marginally suppress the R170G-CFTR channel global misfolding. This was marked by the allosteric conformational destabilization of CFTR domains and ~80% loss of the complex-glycosylated and PM-resident R170G-CFTR pools, both reversed by the 2PT-suppressors (Fig. 2D and S1G-H).”

Page 8: “Introducing the 6SS-suppressor (2PT+M470V, S495P, R555K, Table S1) mutation in F508G-CFTR restored the CL4 dynamics, the native contacts at NBD1-NBD2, and NBD1-CL1, but not at the NBD1-CL4 interface, as expected (Fig. 3A-B, Fig. S3A).” – What is the significance of the very high RMSF at G1069 (Fig. S3A)?

We do not attribute particular significance to the high RMSF at G1069, considering that it is located in a flexible loop. The G1069 residue can be in different conformation causing high RMSF with little impact on neighboring structural elements. To demonstrate this, three very similar conformations of CL4 are shown below, in which the C α of G1069 (blue sphere) is located differently. We added a note on this issue to Fig. S3D.

Page 9: “At the cellular level, the 6SS-suppressor significantly improved the conformational maturation and steady-STATE expression of the complex-glycosylated CFTR (Figs. S7A-B), while it did not alter the basal and cAMP-dependent protein kinase (PKA)-stimulated CFTR chloride transport activity in bronchial epithelia, based ON short circuit current measurements (Fig. S7C).” This result is puzzling to me: why was no increase in I_{sc} found, when presumably, the improved biogenesis would have delivered more channels to the membrane? 6SS mutations have been shown to slow down the channel gating kinetics, but without altering open probability or single-channel conductance [5]. Do the authors believe the improved biogenesis was not occurring in the bronchial epithelia?

Why were changes in biogenesis and in ion channel function associated with the 6SS mutations measured in different systems (Fig. S7A,B vs. S7C)?

We do not have a simple explanation regarding the lack of increased short circuit current (I_{sc}) in CFBE14o-polarized epithelia, although the following mechanism may contribute to the lack of increased I_{sc} for the CFTR-6SS.

i) The comparison of the WT- and 6SS-CFTR activity was performed under different conditions in the quoted as compared to our studies. In the quoted reference, the channel gating was determined on constructs containing three SUMO- and EGFP-tags after reconstitution of isolated microsomes in phospholipid bilayer, in the presence of 0.3 mM ATP at 33°C temperature. The lipid composition of the apical membrane in CFBE14o- cells, the elevated cytosolic ATP concentration (1 mM), and the lack of EGFP and SUMO-tags in our experiments may all influence the single channel gating characteristic of our construct. Specifically, in the bilayer studies the P_o of CFTR was determined upon partial activation, while at near maximal activation in CFBE14o- cells, where the cytosolic ATP concentration is ~1mM. According to previous results the WT-CFTR channel open probability is nearly two-fold higher at 1 mM than at 0.3 mM ATP concentration (PMID: 7520292). Therefore, it possible that that 6SS-suppressors inhibitory effect on the channel gating remained unmasked in the bilayer studies and only manifested in the I_{sc} measurements at 37°C in vivo.

ii) We used CFTR encoding lentivirus transduction to generate the stably transduced CFBE14o- cells, expressing CFTR variant under the control of Tet-ON promoter. We cannot preclude the possibility that the 6SS-CFTR expressing cell lines had lower copy number transcripts than its WT-CFTR counterpart.

iii) We would like to note that the primary goal of our functional studies was to assess whether the 6SS-suppressors may interfere with the relative magnitude of the constitutive and the PKA-stimulated channel activity of the CFTR. We choose to perform short circuit current measurements in CFBE14o- cells, as model that allows monitoring of the macroscopic channel function in its cellular environment (as opposed to in BHK-21 cells). Our results suggest that the relative level maximum PKA-induced activation of CFTR-6SS and WT-CFTR as compared to their constitutive channel activities are similar in CFBE14o- cells.

As all CFTR the folding studies have been performed in BHK-21 cells, we think that repeating the folding studies in CFBE14o- cells is beyond the scope of the present work.

Page 10: Why are the peptides shown in Fig. 4C not the ones mentioned in the text? True, you provide data tables that would allow a reader to check, but text and main figures should provide an overview that most readers would be satisfied with.

As recommended, in the revised manuscript the results description focuses first on peptides that are visible on Fig.4B-C, followed by the discussion of the entire set of results, supported by the supplementary Figures (Figs.4B-C, S7I, S8 and Tables S3-4). We also prepared a new panel (Fig.4C) that illustrates the pervasive dynamic stabilization of the CL1 and CL4 by the 6SS-suppressor, highlighting the central role of coupling helices 1 and 4 backbone rigidification by NBD1 interaction.

“The attenuated side chain and backbone dynamics, imposed by the NBD1-6SS, at least partly accounts for the long-range thermodynamic and kinetic stabilization of the TMDs in the 6SS-CFTR (Figs. 3B and 4C).” I do not understand how the text relates to Figure 3B.

We apologize for this mistake. The Fig. S3B was mistakenly defined as Figure 3B. We corrected the reference.

“The prominent inter-domain dynamic and energetic coupling between the NBD1 F508-loop and the CL4 interface by HDX provides a plausible explanation for the F508G deleterious effect on cooperative domain assembly during the post-translational conformational maturation of CFTR (Fig. 2D-F).” I do not see how the full-length HDX data is informing us on the F508 loop-CL4 interface in particular. Most of the data compares WT to the 6SS mutant. Only

Fig. 4B, on the isolated NBD1 refers to peptides at the CL4 interface. But is that enough to support the sweeping statement? I would appreciate a reference to the specific supporting evidence.

We agree with the reviewer that the allosteric energetic stabilization of the TMD1/2 alone is not sufficient to support our conclusion in the Result section. Therefore, our interim conclusion was deleted, and it is discussed, jointly with other supportive data in the Discussion.

In regard to Reviewer's #3 concern that "I do not see how the full-length HDX data is informing us on the F508 loop-CL4 interface in particular" we have a new illustration that depicts the pervasive dynamic stabilization of the of coupling helices 1 and 4 (CH1 and CH4) backbone, initiated by the NBD1-6SS interaction, and its progressively attenuated dynamic propagation via the CL1 and CL4 to TMD1/TMD2. (Fig.4C).

Page 12: "The P67L-INDUCED structural destabilization is indicated by the significantly reduced thermal stability of the CFTR-6SS as compared to the WT-CFTR, based on their Tm (Fig. S7F)." This sentence does not appear right. Do you mean "reduced thermal stability of the P67L-CFTR-6SS as compared to the CFTR-6SS, based on their Tm (Fig. S7F)."?

Thank you for noting this error, which has been corrected in the revised version.

"The accelerated atomic motions at the F508-loop, CL4, and NBD2 elicited by the P67L MUTATION were also suppressed by the 6SS MUTATIONS based ON MD simulations (Fig. S4C-E), amplifying the dynamic coupling of intracellular parts of central TM helices AS WAS observed for the F508G-6SS mutant." This is all very unclear. If one looks at the CL4 panel of figure S4C, P67L appears to reduce fluctuations at most residues compared to WT, especially at the C-terminal end, with the 6SS-suppressor mutations restoring the high amplitude fluctuations when added to the P67L background. The opposite happens for the F508 loop: P67L increases fluctuations, and the 6SS-suppressor mutations restore low-amplitude fluctuations. On the contrary, in Fig S3A, the F508G mutation increases the amplitude of fluctuations in the F508 loop, but adding the 6SS-suppressor mutations in the F508G background further increases fluctuation amplitude. The sweeping statement above seems to be overly simplistic and/or misleading.

Thank you for highlighting this issue. We replaced these sentences to make our statements clearer.

"While we did not observe a consistent 6SS mutation elicited decrease in atomic motions of back-bone dynamics at the time scale of MD simulations (Figure S4C-E), the 6SS mutation amplified the dynamic coupling of intracellular parts of central TM helices in P67L-CFTR-6SS (Figure S4B), similar to that in F508G-CFTR-6SS."

"Jointly, our HDX data provide structural evidence for the dynamical basis of VX-809 and VX-445-dependent allosteric rescue mechanism of CF folding mutants in the TMD1, and by extrapolation in other CFTR domains" This sentence is quite vague. What exactly, in your data is evidence for a "dynamical basis" of the correctors' mechanism of action?

This sentence has been deleted from the Results section.

The propagated interdomain backbone stabilization, initiated by the VX-809 or VX-445 binding to P67L-CFTR-6SS, as well as the substantially restored protease susceptibilities of all core domain in P67L-CFTR provides support, at least partly, for the "dynamical basis" of the correctors' mechanism of action.

Page 13:

Very unclear – sentence rewritten: "As demonstrated by the NBD1 energetic stabilization (~5 kcal/mol) in the full-length CFTR (Fig. 4G-H), inter-domain allosteric communication is required to avoid kinetic traps and to achieve energetic stabilization of individual native-like domains. As is the case for other ABC transporters^{37,39,40}, correct folding of individual domains of CFTR requires the completion of translation, as well as the progression of post-translational folding."

We replaced this section in the Discussion as recommended.

“As demonstrated by the isolated NBD1 energetic stabilization (~5 kcal/mol) by domain packing in CFTR (Fig. 4F-H), inter-domain allosteric communication is required to avoid kinetic traps and to achieve energetic stabilization of individual native-like domains. This inference is also consistent with the ER retention of the CD4T-NBD1 chimera, which processing significantly improved by second-site suppressor mutations. As is the case for other ABC-transporters^{37,39,40}, correct folding of individual domains of CFTR requires the completion of translation, as well as the progression of post-translational folding.”

Page 15: “We envision that a similar and common mechanism is responsible for improving the conformational biogenesis of F508G- and deltaF508-CFTR, and the P67L or L206W TMD1 mutant by the NBD1-6SS-induced allosteric stabilization” This seems to be more of an assumption than evidence-based. Can the authors spell out their arguments in favour of lumping lasso and F508-loop mutations?

This paragraph has been revised (p.17).

Likewise, shifting the mutants global conformational ensembles towards the native ensembles upon VX-809 or VX-445 binding to the TMD1¹⁵ or TMD1/TMD2¹⁷, respectively, was able to suppress the P67L-CFTR cooperative domain misfolding by allosterically restricting the backbone NHs dynamics in multiple domains (Figs. 6B-C and 7B), and by inference of the L206W-CFTR⁶². Thus VX-809/VX-661 and VX-445 binding to the TMD1/2^{15,17,19} can also rewire the interdomain allosteric networks and tune the post-translational folding landscape of CFTR domains (Figs. 3C, 4C, and 6A). A similar mechanism likely constitutes the basis for the considerable folding rescue of ΔF508-CFTR and other rare misfolding mutations^{11,12}.

Page 16:

CFTR, MRP1 and ABCC6 variants were stably expressed in BHK-21 cells as described 28. Generation of CFBE41o-cell lines (a gift from D. Gruenert, University of California, San Francisco)⁷⁵ that expresses the inducible P67L-6SS-CFTR was done as described⁷⁶. CFTR variants were induced for ≥ 3 days with 250 ng/ml doxycycline.”

Page 17: “Ab binding was determined after correcting FOR background fluorescence signal”

Page 18: “The pulse-labeling efficiency WAS determined in parallel samples after 15 min radioactive pulse to allow comparison.”

“Quantification of CFTR radioactivity FOLLOWING background SUBTRACTION”

Page 20: “Fractional unfolding OF NBD1 VARIANTS was determined at increasing UREA concentrations”

Page 27: “The INWARD-FACING MRP1 structure (PDBID: 5UJ9 95) and the OUTWARD-FACING, ATP-bound structure”

The indicated corrections were implemented in the revised text.

Reviewer #4 (Remarks to the Author):

We would like to thank to Reviewer #4 for the appreciative words and the helpful criticism that improved our manuscript.

Manuscript NCOMMS-22-47099 by Lukacs and colleagues represents a Herculean effort to demonstrate the mechanisms by which CFTR folding correctors VX-809 and VX-445 -- components of the transformative therapeutic Ivacaftor -- have their biochemical effect. As is always that case for work from this lab, the biochemical studies are extremely well performed, and the results are quite noteworthy. The manuscript is quite a magnum opus, with a huge amount of data presented. The results will be of significance well beyond the CFTR field, and the authors do a pretty nice job of extending their findings to two other ABCC proteins closely related to CFTR.

The methodology is quite sound.

However, there are a number of disconcerting errors in the presentation of the results, and I find the interpretation not very convincingly explained. This really impacts only the writing, not the quality of the work itself.

Major concerns:

1. There are English errors throughout the manuscript. MUCH more importantly, one gets the impression that no one read the manuscript carefully before submission, given the retention of several important errors as listed below.

The revised manuscript has been carefully proofread to eliminate all errors.

2. On page 3, line 20, and repeated on page 13, the authors incorrectly state that Trikafta is improving therapy of >90% of CF patients. Only about 90% of CF patients are genetically eligible for Trikafta, and a much smaller fraction are actually prescribed this drug. Furthermore, it does not work on a significant number of those eligible, and others who are eligible cannot tolerate it.

These mistakes have been corrected.

3. Very importantly, you never consider the impact of the R-domain on any of the experimental results in this paper. It is only mentioned at bottom of page 3, but is then included in the schema of Figure 7. Neither MRP1 nor ABCC6 have an R-domain.

The R domain contribution to the conformational maturation of CFTR has not been discussed in the first submission. Although, our knowledge of the R domain in the channel biogenesis is limited, previous publications indicate that the intrinsically unstructured R domain is dispensable regarding the functional and biochemical expression of the Δ R-CFTR (1712985). Some of these results are referenced in the revised manuscript (p.4).

4. Page 5, line 7. Here, you refer to data for F508del-CFTR being shown in Figure 1B. However, Fig. 1B has no data for this variant. This is an example of the surprising sloppiness in the presentation. Surely, someone should have caught this?

The mistakenly referenced Fig.1B has been replaced by Fig.1C.

5. Top of page 7, sentence ending on line 5. I do not agree with this conclusion, because the effect of 2PT on R107G/F508G is larger than the effect of 2PT on F508G alone (Fig. S1D). This is an example of where the authors need to use words to support their statements, a frequently missed effort.

Thank you for pointing out the ambiguous wording in the following sentences on the top of p7.:

“Although the R170G has no discernable effect on fast dynamical motions of the TMD1 (Fig. S3A), it elicited severe cellular/PM expression defect and the TMD1/2 and NBD2 allosteric misfolding that could be rescued by the 2PT-suppressor (Figure 2C and Figure S1F-G). This rescue, however, was severely compromised by the F508G-induced uncoupling of the NBD1-CL4(TMD2) interface (Figs 2C, 2F, Figures S1D, F)”.

While the Fig.2C (revised Fig.2D) and Fig.1SF illustrates the PM density changes in the presence and absence of the 2PT mutations, our conclusions that the 2PT-suppressor significantly increases the PM-density and the complex-glycosylated form abundance of the F508G-CFTR are supported by the presented data. This rescue effect, however, was abrogated in the presence of F508G+R170G mutations, as also indicated by the statistical analysis. On the other hand, as pointed out by the Reviewer #4, both Fig.2F and Fig.S1D require additional explanation, which was omitted in the original submission.

Fig.2F (revised Fig. 2G) depicts the 2PT-induced robust conformational maturation rescue of the F508G-CFTR, measure by metabolic pulse chase technique. The rescue, however, was compromised by destabilizing the CL1-NBD1 interface with the R170G mutation (Fig.2F, revised Fig. 2G).

We do not agree with Reviewer’s #4 statement that “the effect of 2PT on R107G/F508G is larger than the effect of 2PT on F508G alone (Fig. S1D)”. The steady-state expression level of the complex-glycosylated CFTR variants expression level is determined by their ER folding efficiency and their degradation rate from post-Golgi compartments by the peripheral protein QC. Fig. S1D-E (revised) illustrate the 2PT-suppressor effect on mature form turnover rate, using cycloheximide chase and quantitative immunoblotting. The 2PT-suppressor fully restored the F508G-CFTR turnover to the WT-CFTR level (Fig.S1D), a phenomenon that was profoundly attenuated in the presence of both F508G+R170G interface mutations, underlining the inference that CL1-NBD1 interface coupling plays a critical role in the native fold restoration of the F508G-CFTR-2PT upon the NBD1 stabilization (Fig.S1D).

6. Figure S2. There really is no interpretation of these data, other than a brief mention at the bottom of Page 6. Why include?

Fig. S2 provides a comprehensive representation for the HDX kinetics of the entire isolated NBD1s of CFTR, ABCC6 and MRP1, including the NBD1 interdomain interfaces (altogether nine variants). We apologize for inadvertently not referencing these panels. In the revised version the references for all Fig.S2 panels are included (p.8, l10, p10, l9, p11, l18).

7. Figure S3A. There is no interpretation of these data. Why include it? Furthermore, the legend says that the data are means +/- SD for n=3, but there are no error bars. Finally, there is no explanation for what the "6SS" means in the "6SS/F508G" label in Fig. S3A, and you have not introduced the term until page 8 in the text.

In the original manuscript the panel A and panel D were mixed (as you noted in your next comment). Therefore, the interpretation of the HDX data +/- SD for n=3 for the current Fig. S3C. We are sorry for the confusion.

The 6SS-suppressor definition now introduced earlier on p.7 (l29) and discussed more on p9. (l16).

8. Figure S3. Indeed, the legend does not match the figure, and part D is completely unrelated to the data shown there.

These mistakes have been corrected.

9. Page 8, line 5. The authors state that adding 6SS to F508G restored the CL4 dynamics. I see no justification for this statement in Fig. 3B.

The 6SS-suppressors (Table S2) that impart a greater thermal^{51,57} and backbone stabilization of the NBD1 than the 2PT-suppressors (Figs. 4A and 2J), restored the CL4 dynamics (Fig. 3A and S3D), as well as the native contacts of NBD1-NBD2 and NBD1-CL1. Neither the dynamics nor the native contacts of the NBD1-CL4 interface was restored in the F508G-CFTR, as we expected (Figs. 3B and S3D).

10. Page 9, line 6. How is it shown that there were no "large differences"? What would represent a large difference?

The sentences were replaced as follows:

The F728G-MRP1 NBD1-TMD2 interface uncoupling by the F728G mutation was documented by MD simulations (Fig. S6A-C). In addition, a reduction in NBD1/NBD2 native contacts occurred in two out of three trajectories when compared to that of the WT-MRP1 (Fig. S6B).

11. Page 9, line 20. This long sentence makes no sense at all.

We replaced the long sentence with the following statements.

Intramolecular allostery has been established to regulate the CFTR and MRP1 function^{6,25,28,48,49,57-59}. However, allosteric pathways that contribute to CFTR folding and stability, and are likely exploited by folding correctors, have yet to be mapped.

12. In the allosteric network analysis shown in Fig. S4A, why are there so very many differences between the two variants shown above and below in each pair? These differences are spread along the full 1,480-residue length of the protein. Given this observation, how does one interpret each deviation shown, to know if these are significant? Also, wouldn't it be useful to indicate where along each trace the noted mutation(s) reside? Note again that the "6SS" is not yet defined.

Several differences, observed in these types of network analysis, highlight that a single (and potentially small) perturbation can substantially rewire the allosteric network determined based on protein dynamics (therefore we also do not find helpful to indicate the positions of mutations). We did not select significant differences here, but we highlighted those important regions for which we have experimental data. Then we used the corresponding betweenness centrality measure to explain the importance of a residue in allosteric signal propagation that cannot be interrogated using exclusively experimental methods. Since the network analysis at this level (per residue betweenness centrality) explains our experimental results well, it also supports the validity of the results of the community-level network analysis.

First description of 6SS-suppressor is included in p.7 in the revised manuscript.

13. Figure 3. Don't we need to see the allosteric coupling figure (as in part C) for WT-6SS, too?

The MD simulations were primarily aimed to understand the negative effect of the mutational perturbation on CFTR allosteric networks, and its reversal by the 6SS-suppressor towards that of the WT-CFTR.

14. Figure 6. This is really confusing and does not follow the presentation in the text on page 12. Are both "WT" and "P67L" variants bearing the 6SS stabilizations?

The presentation of Fig.6 has been rewritten.

In the revised description we also included the justification and the weaknesses of using the P67L-CFTR-6SS variant.

Minor concerns:

1. In the abstract, even the first sentence reads oddly. Instead of "are used" I would recommend "can be used".
2. In the abstract, please use "canonical ABC-transporters' core" to leave space for the incredible diversity of ABC proteins and their subunit composition.

The abstract has been corrected and partially rewritten.

3. Page 3, line 28. Not clear what is meant here.
4. Page 3, line 31. You have not yet mentioned the R-domain.
5. Figure S1D shows "R508G" instead of "F508G".
6. Figure S7A and B. What on Earth does "06" mean?

These minor concerns have been addressed in the revised version.

7. The authors probably need to provide a caveat (in Discussion) regarding the interpretation of the effects of either VX-809 or VX-445 on these peptides given the absence of membrane lipids.

The following caveat has been provided in the Discussion:

The HDX-MS studies on purified CFTR variants have the limitation that the channel was reconstituted in GDN micelles. This may alter the channel microenvironment and influence the inter-domain coupling dynamics/energetics in response to mutations and pharmaco-chaperone binding.

8. Figure 7B. The second step is not possible since the graphic implies that NBD1 is translated and folded before TMD1 is translated.

The coloring of the revised figure now indicates that correctors can interact with full-length CFTR folding intermediates, consistent with the results of our biochemical studies.

REVIEWER COMMENTS

Reviewer #1 (Remarks to the Author):

The authors addressed most comments. However, there are still some concerns below.

Major or minor (the numbers below are inheriting those from the previous comments):

(1) (minor) In Fig.1c, the left panel is separated. There is no band at about 500 kDa in (-,-,gray) in the left/left 5-column diagram. On the other hand, there is a distinct band at that position of (-,-,gray) in the left/right two-column diagram. Are these the same ones from independent experiments?

(2) (minor?) Suppl. Data 1 is very hard to find data of interest. There are three sheets, Sheet1 has L206W: N=4, DF508: N="5", whereas Sheet2 has L206W: N=4, ΔF508: N="3" (please check the numbers, and delete unnecessary sheets (single: recommended)).

Now, regarding I.123 in MS: "they reduce the WT-CFTR folding efficiency from '~32%' to '~1-2%' ", Sheet2 says WT: '31%', L206W: "0.67%", ΔF508: "0.4% (different from 1.2 (Sheet1))". The authors might choose 3 (Sheet3) out of 5 (Sheet1). Then, "~2%"(marginal increase) in MS I.126, and "~31% and ~37%" in MS I.140 were not found in these sheets. Moreover, the value of F508G of "~1.8 ± 0.1%" in I.168 might be "2.1 ± 0.3"(from Sheet2), and for F508G-2PT, "~1.8%" in 178 might be "2.3%" (very minorly, the red bar (F508G+2PT) looks like it's over 20% in Fig.2g). Please verify these (or all) data.

(The panel name "I" (uppercase) in Fig.2. would be "i" (lowercase))

(3) (very minor) MS says "L1065/F1068/Y1073/F1074" in I.151, but L1065 is missing in Fig. 2a (also in Fig.3a). Please depict it.

(5) First, please answer the following simple question. Were ATP molecules included in the simulations?

If YES, please describe the force field parameters etc. in the METHODS section. There would be almost no particular problem in descriptions.

If NO, then the current descriptions within MS are very misleading. Much of the "ATP-bound" and "bound ATP" in MS and SI should be removed. In addition, MS I.818-821 and SI "Under physiological ..., most of the time" regarding ATP binding should also be deleted.

Moreover, in this case, the reply seems to ignore the previous comment: "There might be observed the relaxation from OF state (of the starting structure) in the ATP-free state, which would affect the results".

In general, the distance between Walker A and signature motifs is often used as an index to measure the stability of NBD dimers and pocket formation. In any case (YES or NO), please show plots of these distances for each pocket in the SI. It will be important information to understand what happened to NBDs in the simulations. (For dimerized NBDs, the value would be approximately 1.1 nm.)

(6) The author's ongoing simulations with small molecules are promising.

In general, there are some issues using modeled structures (Rosetta, homology modeling etc.) as the authors replied. However, they are similar in this study with loop modeling (for CFTR). To validate the data obtained here (in comparison with no loop modeling), it is necessary to cap the ends and examine the effect of the modeled loop. We do not require such verification.

McDonald et al. appropriately cite the previous studies (Abreu et al., 2019 & Odera et al., 2018).

McDonald et al. describes F1068, Y1073, F1074, etc. and Odera et al. does F1068, F1074, W496, etc., which would be highly relevant to this study. Therefore, it seems appropriate to cite these relevant previous studies. The author seems to respectfully disagree with the reviewer, but a large RMSD value is not a good indicator. This is because comparisons between OF-start and IF-start simulations are not valid. The RMSD value between the OF and IF cryo-EM structures is about 7.5 Å. Therefore, the RMSD value of 6-8 Å is not necessarily large when conformational changes occur.

Moreover, recently, MRP1 simulation study has been published by Tóth et al (2023). In it, the RMSD

value seems to exceed 12 Å (from the supplement).

Then, the authors may deny homology modeling, but if so, the results of MRP1 and ABCC6 here are also unnecessary.

(7) (minor) In relation to the above comment, please show RMSD not only in CFTR (in Fig. S3) but also in MRP1 (and/or ABCC6) (in Fig. S6).

(13) We understand that the authors used the structure modeled here rather than the AlphaFold model structure because they assumed an OF state. As such, the use of IF structures in many figures is misleading to the readers. In the MD simulation section, it should be stated more explicitly that the authors analyzed OF structures of CFTR, MRP1, and ABCC6.

Moreover, in the Supplementary, AlphaFold structures are used in Figs. S2 and S5. Fig. S2 is good because it is a comparison between OF structures. However, why are AlphaFold's MRP1 and ABCC6 displayed in Fig. S5 even though the authors have done their own modeling? In science, it is necessary to understand what is close to the truth. If the AlphaFold model is more appropriate, the authors may add a few notes to the Fig legend or Methods (such as showing the AlphaFold structure in some parts of the Suppl.).

Below are some additional minor comments (sorry).

(15) (additional) In l.93 of MS, MD results (Fig. S3a) are suddenly cited, but this sentence seems unnecessary (nothing could be said from the present simulations).

(16) (additional) Please check the difference between F1140 in Fig. S5b and F1141 in l.243 (for ABCC6), and R1173 in the fig. and R1172 in the same line (for MRP1).

(17) (additional) While reading the MS for peer review, a little interesting thing was found as following. The RMSF values around the residue 1070 of the mutant is larger than those of WT in Fig.S3c and Fig.S6a. On the other hand, in Fig. S4, those RMSFs in R170G and P67L mutants are slightly smaller than those of WT. If this point reinforces anything, please describe a little in MS.

This study, which deals with folding collectors in CFTR, is very interesting and would deserve publication.

Reviewer #2 (Remarks to the Author):

My concerns have been properly addressed, I have no reservation regarding the publication of this research work. Thanks to the thorough work of the authors and other reviewers, the quality of this manuscript is very much improved.

Reviewer #3 (Remarks to the Author):

I had two main concerns regarding the original manuscript: (A) clarity of language and presentation and (B) the use of many different experimental systems, without a clearly articulated experimental plan, leading to vague, general conclusions.

(A) I do not believe the clarity of the revised manuscript is sufficiently improved for publication yet. The paper is still extremely hard to read, with grammatical and typographical errors in the text, and unclear layout and inaccurate labelling in figures. Following are my detailed suggestions for improvement. However, I think a potential final version would require further careful proofreading and

a special effort at making ideas more accessible.

Abstract

What is meant by "combinatorial approaches"? I do not think the methods relate to combinations of different parts. Suggestion: "Different approaches..."

"Allosteric or orthosteric binding of VX-809 or VX-445 folding correctors can rescue kinetically trapped CFTR post-translational folding by cystic fibrosis (CF) mutations in NBD1 or TMD1, which requires NBD1-TMD2 coupling." Suggestion: "Allosteric or orthosteric binding of VX-809 or VX-445 folding correctors can rescue post-translational folding of CFTR that is kinetically trapped by cystic fibrosis (CF) mutations in NBD1 or TMD1. This rescue requires NBD1-TMD2 coupling."

"offering a framework for mechanistic dissection of genetic diseases caused of ABC transporters." Suggestion: "offering a framework for mechanistic dissection of genetic diseases caused by mutations in ABC transporters."

Page 3

"The CLs drive the TMD1/2 conformational transitions upon the NBDs association-dissociation cycle, a requirement for substrate translocation that is facilitated by domain-swapped structural elements of P-glycoprotein (ABCB1)- like exporters, enabling communications of each domain with all the others." Suggestion ""The CLs drive TMD1/2 conformational transitions upon NBD association-dissociation cycles. The NBD/TMD coupling is a requirement for substrate translocation and is facilitated by domain-swapped structural elements in Type IV exporters (Thomas et al., 2020 <https://doi.org/10.1002/1873-3468.13935>), which enable communication of each domain with all the others."

"the empirical development of the FDA-approved Trikafta for ~90% of CF-patients." Suggestion: "the empirical development of the FDA-approved Trikafta for which ~90% of CF patients are eligible" [see comment 2. of Reviewer #4]

Page 5

"First, at the cellular level, the transporters biosynthetic processing, expression, and metabolic stability, as well as their domains' conformational dynamics, were monitored" What is meant by monitoring of "their domain's conformational dynamics" at a cellular level? I think no measurements directly monitoring "domain conformational dynamics" were obtained at a cellular level. Inferences based on core vs. complex glycosylation were made, but that is not equivalent.

Page 6

"regardless their exit from ER and N-linked complex-glycosylation permitted or not." Suggestion: "regardless OF WHETHER their exit from THE ER and N-linked complex-glycosylation ARE permitted or not"

Supplementary Data 1 needs tidying up: remove worksheets 1 and 3; correct spelling mistake (efficiency); define what is meant by "corrected"; is the error shown standard deviation (as in worksheet 1), or Standard Error, of the mean (as in worksheet 2)? Check consistency of text and table: Is modulator exposure during (depletion+pulse) or (pulse+chase) only rescuing 6% and 7% of F508del-CFTR? "if drug(s) were present only during the chase periods [...] the folding efficiency of the ΔF508- and L206W-CFTR was restored to ~31% and ~37%, respectively" - but I don't find any 31% and 37% values in the table.

Page 8

"reduced the maturation efficiency of the core-glycosylated F508G-CFTR from $31.0 \pm 2.2\%$ to $\sim 1.8 \pm 0.1\%$ " Why does Supplementary data 1 have two values slightly above 2% and no 1.8%?

"The 2PT-suppressor mutations have been established to stabilize NBD1 variants and are located outside of the domain interfaces with exception for the S492P." Suggestion: "The 2PT-suppressor mutations have been established to stabilize NBD1 variants and are mainly located outside the domain interfaces (S492P however, is positioned at the NBD1/CL4 interface)."

Page 9

"by the 2PT-suppressors at peptides confined to all three domain interfaces" What is meant by "confined"? Many of these peptides extend from the domain core to the interfaces.

Page 10

"Allosteric coupling between dynamic communities (e.g.: CL3 and CL4, TMH1+TMH2 and CL1) was attenuated (Fig. 3b-c and S4a-b)" This all still seems unclear. Are we comparing WT with F508G- (the only mutation mentioned in this section yet) or WT- with F508G-6SS-CFTR? Would a direct WT- to F508G-CFTR comparison in Fig. S4a be useful?

"Although the centrality of the CL4 ensured an augmented dynamic coupling via the F508 residue, it could marginally suppress the R170G-CFTR channel global misfolding. This was marked by the allosteric conformational destabilization of CFTR domains and ~80% loss of the complex glycosylated and PM-resident R170G-CFTR pools, both reversed by the 2PT-suppressors (Fig. 2d and S1g-h). These observations with the F508G-CFTR-2PT phenotype support the notion that stabilization of fast atomic motions at both CL1- and CL4-NBD1 interfaces contributes to the allosteric conformational rescue of the F508G-CFTR by the 2PT-suppressors, because both interface disruptions thwarted the 2PT dependent folding rescue (Fig. 2d and S1h)." Suggestion (although I am not sure I understood correctly): "Although the INCREASED centrality of the CL4 ensured an augmented dynamic coupling via the F508 residue, THIS could ONLY marginally suppress the R170G-CFTR channel global misfolding. The R170G-CFTR DEFECT was marked by the allosteric conformational destabilization of CFTR domains and ~80% loss of the complex glycosylated and PM-resident R170G-CFTR pools, both reversed by the 2PT-suppressors (Fig. 2d and S1g-h). These observations, IN CONJUNCTION with OBSERVATIONS ON the F508G-CFTR-2PT phenotype (FIG. XX) support the notion that stabilization of fast atomic motions at both CL1- and CL4-NBD1 interfaces CONTRIBUTE to the allosteric conformational rescue of the F508G-CFTR by the 2PT-suppressors, because SIMULTANEOUS DISRUPTION OF BOTH INTERFACES thwarted the 2PT-dependent folding rescue (Fig. 2d and S1h)."

"The 6SS-suppressors (Supplementary Data 2) that impart a greater thermal and backbone stabilization of the NBD1 than the 2PT-suppressors (Fig. 3a and 2j), restored the CL4 dynamics, as well as the native contacts of NBD1-NBD2 and NBD1-CL1, but not the NBD1-CL4 interface in the F508G-CFTR as expected (Fig. 3b and S3d)." Suggestion "The 6SS-suppressors (Supplementary Data 2) that impart a greater thermal and backbone stabilization of the NBD1 than the 2PT-suppressors (Fig. 3a and 2j), restored CL4 dynamics, as well as the native contacts of NBD1-NBD2 and NBD1-CL1, but not OF the NBD1-CL4 interface in the F508G-CFTR, as expected (Fig. 3b and S3d)."

Page 11

"The conserved F713(ABCC6) and F728(MRP1) of NBD1 H3-H4 loops (Fig. S5a) are engaged in a hydrophobic patch formation with F1141/F1146(CL7/TMD2) and a cation- π interaction with the R1172 (CL7/TMD2) in ABCC6 and bovine MRP1-transporters, respectively (Fig. S5b)" R1172 is present in bovine MRP1, but alignments (even yours in Fig. S5a) show it is not homologous to R1070 in CFTR. Is there a basic residue interacting with F713 in human ABCC6? Residue 1172 is an alanine, not an arginine in ABCC6.

Check numbering inconsistencies R1172/R1173 (text/Figure S5a legend) possibly due to MRP1 bovine/human differences.

Page 12

"Our assumption is supported by the bidirectional rescue mechanism of NBD1 CF-mutations (e.g. Δ F508) with correctors targeting the TMDs and NBD2 or genetic suppressors of the NBD1." This

sentence is unclear. What assumption? What exactly are the two "directions" implied? Does it relate to the synergistic rescue by correctors and genetic revertants? Or to the mutual stabilization of NBD1 by other domains and other domains by NBD1? This needs to be clarified.

Page 13

"Intuitively, the 6SS-suppressors not only stabilized the conformational dynamics of the isolated NBD1 interfaces (Fig. 4a and S2d), but their coupled interfaces in trans and beyond in the isolated CFTR. This was visualized by projecting individual peptides differential deuteration in 6SS- and WT-CFTR [% Δ D(6SS-WT)]" Suggestion: "Intuitively, the 6SS-suppressors not only stabilized the conformational dynamics of peptides positioned at interfaces in isolated NBD1 (Fig. 4a and S2d), but also of peptides positioned in other domains, beyond the NBD1 interface in the full length CFTR. This was visualized by projecting the differential deuteration [% Δ D(6SS-WT)] of individual peptides"

Page 14

"The reduced deuteration of the TMH1, TMH2, TMH5, TMH8, and TMH11 segments became apparent only after 20 min incubation (Fig. S8)." All TM11 peptides I can spot (from residue 1069 to 1120) are less deuterated throughout the time course.

"132-1336" to "1325-1336"

"Thus, allosteric dynamic rigidification contributes to the stabilization of multiple domains in the 6SS-CFTR (Fig. S4c and S7h), albeit the channel global conformation remains similar to that of the WT-CFTR, suggested by their comparable sensitivity to PKA-mediated activation (Fig. S7c)"

What is meant by "allosteric dynamic rigidification"? Does it mean that some loops in NBD1 positioned at domain interfaces become less flexible, affecting conformational flexibility of other domains? (I don't think Fig. S4c is relevant, as it does not include WT-6SS-CFTR). Wouldn't it be expected that reducing mobility of some loops at domain interfaces would not alter the global conformation of a multidomain protein, just "rigidify" it?

"The impact of inter-domain packing on the NBD1 conformational stability remained unknown and was investigated next by comparing the HDX kinetics of the isolated NBD1 and in the context of WT-CFTR." Suggestion: "The impact of inter-domain packing on NBD1 conformational stability was investigated by comparing HDX kinetics of NBD1 in isolation versus in the context of full-length WT-CFTR."

"These results revealed that the multidirectional dynamic/energetic interdomain allosteric networks improve the thermal stability of the newly translated NBD1 in the context of CFTR and can, in principle, relay the pharmacological stabilization TMD1/2 to rescue the primary or coupled folding defects of NBD1, which was assessed next." Suggestion: "These results revealed that the multidirectional dynamic/energetic interdomain allosteric networks improve the thermal stability of the newly translated NBD1 in the context of FULL-LENGTH CFTR and can, in principle, relay the pharmacological stabilization OF TMD1/2 to rescue the primary or coupled folding defects of NBD1, which was assessed next."

Page 15

"Although VX-809/VX-661 correctors were proposed to stabilize the TMD1 native fold by binding to the hydrophobic pocket formed by the TMH1, 2, 3, and 6 of the WT-CFTR (Fig. 5a), the molecular mechanism action of VX-809/VX-661 or VX-445 on CF mutations yet to be elucidated." Suggestion: "Although VX-809/VX-661 correctors were proposed to stabilize the TMD1 native fold by binding to the hydrophobic pocket formed by the TMH1, 2, 3, and 6 of WT-CFTR (Fig. 5a), the molecular mechanism OF action of VX-809/VX-661 or VX-445 on VARIANTS CARRYING CF mutations HAS yet to be elucidated."

"we employed the P67L and L206W CF mutations (Fig.5a) that are confined to the L0 region and the TMH3(TMD1) in CFTR, respectively, and are susceptible to correction" Suggestion: "we employed the

P67L and L206W CF mutations (Fig.5a) that are IN the L0 and the TMH3 (TMD1) REGIONS in CFTR, respectively, and are susceptible to PHARMACOLOGICAL correction”

Page 16

“To directly demonstrate the P67L-induced inter-domain perturbations at the backbone dynamics level and its reversal by folding correctors, we used the HDX-MS technique on isolated CFTR variants. As the P67L mutation reduced the WT-CFTR expression by >90% (Fig. 5d-e), we selected the P67L-CFTR-6SS variant, which partially restored the P67L-CFTR misprocessing, while ensured acceptable protein yield (Figs. S7d). Importantly, the P67L mutation reduced the thermal stability of the CFTR-6SS (Fig.S7e) and preserved the P67L-CFTR-6SS rescue ability by VX-809 and VX-445 (Figs. 5d right panel and 5i).” Suggestion: “To directly demonstrate the P67L-induced inter-domain perturbations at the level of backbone dynamics, and their reversal by folding correctors, we used the HDX-MS technique on isolated CFTR variants. As the P67L mutation reduced the WT-CFTR expression by >90% (Fig. 5d-e), we selected the P67L-CFTR-6SS variant, which partially mended P67L-CFTR misprocessing, while ensuring acceptable protein yield (Figs. S7d). Importantly, the P67L mutation reduced the thermal stability of the CFTR-6SS (Fig.S7e) and preserved the susceptibility to VX-809 and VX-445 rescue (Figs. 5d right panel and 5i).”

“The P67L mutation provoked accelerated fast atomic motions at the F508-loop, CL4, and NBD2 that were mitigated by the 6SS-suppressors based on MD simulations (Fig. S4b-e), amplifying the dynamic coupling of intracellular parts of TM helices, as was observed for the F508G-6SS mutant (Fig. S3d-f).” This sentence has not been replaced, despite what is stated in the rebuttal. To me it is still unclear. I repeat what I stated earlier: in Fig S4c, P67L appears to reduce fluctuations at most residues in CL4 compared to WT, especially at the C-terminal end, with the 6SS-suppressor mutations (in P67L background) restoring the high amplitude fluctuations. The opposite happens for the F508 loop: P67L increases fluctuations, and the 6SS-suppressor mutations restore low-amplitude fluctuations. On the contrary, in Fig S3d, the F508G mutation increases the amplitude of fluctuations in the F508 loop, but adding the 6SS-suppressor mutations in the F508G background further increases fluctuation amplitude. Perhaps I am not understanding correctly, but the sentence above seems to oversimplify, and does not describe the data accurately.

Page 18

“While the first phase consists of the obligatory co-translational formation of secondary structural elements of the core domains, the native tertiary fold completion is delayed by the combination of several processes. Ribosome-nascent chain interactions, and the large size and complex subdomains’ topology of NBDs, as well as the protracted posttranslational development of inter-domain coupling, inherent to the transporters’ domain-swapped structure in analogy to some soluble multi-domain proteins (Fig. 1a and 7a). A main verb is missing from the second sentence. Was a colon intended, instead of a full stop, after “processes”? If so, I’m not sure ribosome-nascent chain interactions and complexity of NBD subdomain topology can be considered “processes”. The final clause “in analogy to some soluble multi-domain proteins” is repeated later (more clearly) and could be omitted here.

“The post-translational domain-folding phase of CFTR is reinforced by the reciprocal dynamic/energetic stabilization of NBD1 in the full-length CFTR (Fig. 4f-h), which is dependent on the conformational maturation of the TMD1/TMD2/NBD2 that consequently requires the reciprocal influence of the NBD1 inter-domain coupling to attain the ER-exit competent conformation (Fig. 7a).” Could this sentence be re-written? I think it refers to the mutual stabilization of (i) NBD1 by other CFTR domains, and (ii) of TMD1/TMD2/NBD2 by a stably folded NBD1. However, as written, the subjects of “is dependent”, and “requires” are unclear.

Page 20

“The NBD1-6SS was able to globally stabilize both the fast atomic motions and backbone NHs dynamics, which translated into favorably folding energy and unfolding activation energy changes of TMD1/2 peptides of WT-CFTR (Fig. 4b-c).” Suggestion: “The NBD1-6SS was able to globally stabilize

both the fast atomic motions and backbone NHs dynamics, which translated into favorable CHANGES in folding energy and unfolding activation energy of TMD1/2 peptides of WT-CFTR (Fig. 4b-c).” However, 6SS mutations do not always “stabilize [...] fast atomic motions” (see points referring to statement on P67L on page 16, referring to Figs S4c and S3d, repeated from first round of Reviewer #3 comments).

Page 21

“This paradigm requires further validation to select possible pharmacotherapy several other ABC-transporters-mediated conformational diseases.”

“This paradigm requires further validation to select possible pharmacotherapies for conformational diseases associated with several other ABC-transporters.”

Figures:

Clarity of the figures in general might be aided by using harmonised formats and labels.

- how is “% maturation efficiency” (fig. 1b, 1c) different from “% of matured WT-CFTR” (fig. 2d) or “% CFTR maturation efficiency” (fig 2g) or “% of complex-glycosylated WT-CFTR expression” (fig. 5 e,h)? If they do not differ, they should have identical labels on figures. This should be defined clearly in the Methods section.

- Could time course of deuteration for selected peptides be added to Figure 6, to make it more similar to Figs. 4a-c, S1, and S6h? How differential deuteration is abbreviated (in text, legends and figure labels) should also be harmonised. Several abbreviations are used, including $\% \Delta D(6SS-WT)$; $\Delta \% D$; $\% HDX6SS-WT$; $\Delta HDX P67L-WT$. It might also be helpful to use the same colour scale to indicate differential deuteration in Figs 4b, 6a-c and S6h.

- RMSD and RMSF are used interchangeably. Better use only one.

Figures 3c and S4b are difficult to read. Perhaps more extensive labelling could help. For instance, does the purple sphere that gains importance in F508G include TM11-TM10? The labelling and colour coding don’t seem to correspond between 3c and 4b: is light blue CL1, CL3 or CL4, or all of them? It might help if a unique scheme could be used, perhaps with addition of a legend, if labels become too crowded. Figure S4b could be reorganised, with F508G - WT - P67L on the top row, and F508G-6SS - R170G - P67L-6SS below. This way WT could be easily compared with each single mutant, and F508G and P67L could be compared vertically with their corresponding 6SS versions.

Figure 3f, left: why are the lanes repeated?

Figure 4f: For peptide 490-494, should an “S6” label be added, in red (Q loop), to link to Fig. 4g?

Figure Legends:

Fig. 1a: “The SAME color coding of CFTR domains is used for all relevant illustrations.”

Fig. 1b: Label “VX3” is not explained. Suggestion: insert “For the last lane (“labelled 3-VX”),” before “Cells were exposed...”

Fig. 1c: “Cells that were exposed to VX-445 (2 μ M) and VX-809 (+3 μ M) are indicated by + during the Met/Cys depletion, pulse-labelling and/or chase. Gray box depicts that only pulse-labelling was included without the chase.” Suggestion: “Exposure to VX-445 (2 μ M) and VX-809 (+3 μ M) during the Met/Cys depletion, pulse-labelling and/or chase is indicated by “+” in the corresponding rows. A gray box in “Chase” row indicates that only pulse-labelling was included without the chase, while “-” indicates that the chase was performed but modulators were not present.” A note could be added on why two lane are included in which correctors were absent from depletion and pulse, and no chase was performed - presumably to have controls for the effect of the chase with modulators (first lane) and without modulators (lane6) [see Reviewer #1, point (1)].

Fig. 4b – “deuteration kinetics OF selected” and remove duplication “Means \pm S.D., n=3-4.”

Fig. 4c In the legend for Fig. S7i it is stated “Data are means \pm S.E.M., n=3. ” In the legend for Fig. 4c “Means \pm S.D., n=3-4. Means \pm S.D., n=3-4.” (twice). Some of the peptides are shown in both figures (e.g. 164-170 and 1063-1067) and look identical. Is the error bar SD or SEM?

Fig. 5c “Phosphorimage visualization (left panel) and quantification of the maturation efficiency (right panel).” Suggestion: “Phosphorimage visualization (TOP panel) and quantification of the maturation

efficiency (BOTTOM panel)."

Fig. 6a What is SDM?

Fig. 6b-c. On page 33 (Methods) you state "was pre-incubated at 25°C for 5 min in the absence or presence of 30 μ M VX-809 or VX-445 correctors. [...] The corrector concentration was kept at 30 μ M during HDX incubation for 10, 240 and 600 s" In the figure legend you mention "3 μ M VX-809 (B) or 2 μ M VX-445 (C)". What was the corrector concentration? In addition, in the methods the longest incubation indicated is 10 minutes (600s). In the figure legend you mention 20 minutes. However, I don't see the 20 minutes data in the Supplementary Data 5 tables - just 10 s and 4 minutes. Can the measurements for the longer incubations be added?

Fig. 6b: Add "VX-809 is shown as magenta spheres"

Fig. 6c: The structure shown is not 5UAK. Probably it is 8EIQ, with VX445 in yellow spheres. Please clarify.

Supplementary Information

S1a: what does the asterisk close to 505-511 indicate?

S3a: axis mis-labelled R1070G instead of R170G

S3 legend "RMSD (root mean square fluctuation) values were also calculated from simulations with complete CFTR structures but computed without taking into account to demonstrate that the high RMSD and some of their fluctuations were caused by the dynamic RI loop." Something is missing.

(B) Overall, I think the rationale for combining experiments on a multitude of variants and systems is now more openly discussed. The many different methods employed have different technical requirements; the different mutations all give broadly consistent results (conformational maturation of CFTR requires interactions that span domain interfaces). However, the fact that experimental results were obtained on different systems (different mutants, different cell lines, different conditions) inevitably means that conclusions are weaker and less mechanistically informative, than if the study had focused on a smaller number of mutant variants and systems.

To illustrate this point, to my question "Why were changes in biogenesis and in ion channel function associated with the 6SS mutations measured in different [cellular] systems (Fig. S7A,B vs. S7C)?" the authors answer "We choose to perform short circuit current measurements in CFBE14o- cells, as model that allows monitoring of the macroscopic channel function in its cellular environment (as opposed to in BHK-21 cells [where the biogenesis studies were performed])." These 6SS suppressor mutations (6 point mutations and a 30-residue deletion) cause extensive modifications, including at the crucial NBDI/CL1/CL4 domain interfaces. Clearly, folding and stability are improved. But function might be negatively affected. However, the experiments presented do not inform on the effect of these mutations on the function of individual channels: the overall transepithelial current measured depends both on the number of channels at the membrane and on the single-channel activity. Because biogenesis (maturation and stability) and function were measured in different systems, it is unclear whether the unchanged overall current measured is the result of (i) a lack of folding/stability improvement in this system (unlike in all other experimental systems presented in the paper) or (ii) a functional inhibition. There might be technical reasons underlying the unusual experimental design, but this is clearly a limitation. (Note that two of the differences the authors cite in the rebuttal - higher ATP, lack of N-terminal SUMO* tag - will result in an increased single-channel open probability, making it even more surprising that they do not see an increased short-circuit current, given the presumably increased plasma membrane density of 6SS-CFTR compared to WT-CFTR).

I am not fully convinced that the extensive investigation on systems including the 6SS suppressor mutations, are truly informative with respect to folding of WT-CFTR and of variants bearing CF mutations. While 6SS suppressors might facilitate purification of mutant protein, the assumption that they restore interdomain coupling (e.g. in F508G- and P67L-CFTR mutants) that is similar to that in WT-CFTR is not necessarily valid. The betweenness centrality graphs (Fig. S4a), for instance, highlight

how the 6SS suppressor mutations appear to shift the allosteric pathways: e.g. centrality of residues in the latter part of ICL4 and in TM11 is low in WT and in F508G, but high in the F508G-6SS-CFTR; centrality of residues 350-375 is low in WT and in F508G, but high in F508G-6SS. A "rigidification", expressed as reduced loss of native contacts at the NBD1-NBD2 interface by 6SS suppressor mutations (Figs. 3b), also suggests the rescue of conformational maturation is not achieved by restoring WT molecular dynamics.

Best wishes,
Paola Vergani

Reviewer #4 (Remarks to the Author):

The authors are commended for doing a great job in responding to Reviewer input. However, there remain a number of small errors in the writing, that would surely be caught if the three senior authors would read the manuscript carefully.

Reviewer #1 (Remarks to the Author):

We thank Reviewer #1 for the constructive critique that helped to further improve our data presentation.

Major or minor (the numbers below are inheriting those from the previous comments):

(1) (minor) In Fig.1c, the left panel is separated. There is no band at about 500 kDa in (-,-,gray) in the left/left 5-column diagram. On the other hand, there is a distinct band at that position of (-,-,gray) in the left/right two-column diagram. Are these the same ones from independent experiments?

The composite phosphorimage panel of Fig.1c (left) depicts a representative result of n=4-6 independent experiments (pharmacological rescue of the F508del-CFTR). The right phosphorimage panel is a representative independent experiment of four. The quantification of the F508del-CFTR maturation efficiency is depicted in the bar plot Fig.1c.

In the revised Fig.1c we included the entire phosphorimage, demonstrating the presence of the high molecular mass F508del-CFTR, as detectable for the L206W-CFTR, as well on Fig.1b. All uncropped phosphorimages are included in the Source Data files.

(2) (minor?) Suppl. Data 1 is very hard to find data of interest. There are three sheets, Sheet1 has L206W: N=4, ΔF508: N="5", whereas Sheet2 has L206W: N=4, ΔF508: N="3" (please check the numbers, and delete unnecessary sheets (single: recommended)).

Now, regarding I.123 in MS: "they reduce the WT-CFTR folding efficiency from '~32%' to '~1-2%' ", Sheet2 says WT: '31%', L206W: "0.67%", ΔF508: "0.4% (different from 1.2 (Sheet1))". The authors might choose 3 (Sheet3) out of 5 (Sheet1). Then, "~2%" (marginal increase) in MS I.126, and "~31% and ~37%" in MS I.140 were not found in these sheets. Moreover, the value of F508G of "~1.8 ± 0.1%" in I.168 might be "2.1 ± 0.3" (from Sheet2), and for F508G-2PT, "~1.8%" in 178 might be "2.3%" (very minorly, the red bar (F508G+2PT) looks like it's over 20% in Fig.2g). Please verify these (or all) data.

(The panel name "I" (uppercase) in Fig.2. would be "i" (lowercase).

We apologize for our oversight and including interim data in the Suppl. Data 1. The revised data sheet now only depicts the relevant pulse chase experiments as numerical results.

(3) (very minor) MS says "L1065/F1068/Y1073/F1074" in I.151, but L1065 is missing in Fig. 2a (also in Fig.3a). Please depict it.

The L1065 residues has been included in the revised Fig.2a and Fig.3a.

(5) First, please answer the following simple question. Were ATP molecules included in the simulations? If YES, please describe the force field parameters etc. in the METHODS section. There would be almost no particular problem in descriptions.

If NO, then the current descriptions within MS are very misleading. Much of the "ATP-bound" and "bound ATP" in MS and SI should be removed. In addition, MS I.818-821 and SI "Under physiological ,,, most of the time" regarding ATP binding should also be deleted.

Moreover, in this case, the reply seems to ignore the previous comment: "There might be observed the relaxation from OF state (of the starting structure) in the ATP-free state, which would affect the results".

In general, the distance between Walker A and signature motifs is often used as an index to measure the stability of NBD dimers and pocket formation. In any case (YES or NO), please show plots of these distances for each pocket in the SI. It will be important information to understand what happened to NBDs in the simulations. (For dimerized NBDs, the value would be approximately 1.1 nm.)

We are sorry for missing these questions from the previous review.

The Mg²⁺ ions and ATP molecules present in the 6MSM structure were also present in the simulations. The parameters for these molecules were included in the CHARMM36m force field we used for the simulations and are specified in the Methods section of the revised manuscript.

In our recent study investigating available ABC protein structures (Tordai et al. IJMS 2022, <https://pubmed.ncbi.nlm.nih.gov/36012140>), we defined the NBD distance as the distance between C α atoms of a Walker A and an opposite signature residue (for CFTR Site-1: 464/1348 and Site-2: 1250/550). We found 14.5 Å as a reasonable cutoff to differentiate between closed and open NBDs. These distances are 10.9 Å and 11.1 Å for site I and site II, respectively, in the 6MSM structure. Although these values were fluctuating in our simulations, they are small when compared to NBD movements in simulations started with inward-facing conformation that also result in non-native NBD dimer conformations^{112,113}. Please find the requested data and plots in Supplementary Table S1.

(6) The author's ongoing simulations with small molecules are promising.

In general, there are some issues using modeled structures (Rosetta, homology modeling etc.) as the authors replied. However, they are similar in this study with loop modeling (for CFTR). To validate the data obtained here (in comparison with no loop modeling), it is necessary to cap the ends and examine the effect of the modeled loop. We do not require such verification.

McDonald et al. appropriately cite the previous studies (Abreu et al., 2019 & Odera et al., 2018). McDonald et al. describes F1068, Y1073, F1074, etc. and Odera et al. does F1068, F1074, W496, etc., which would be highly relevant to this study. Therefore, it seems appropriate to cite these relevant previous studies. The author seems to respectfully disagree with the reviewer, but a large RMSD value is not a good indicator. This is because comparisons between OF-start and IF-start simulations are not valid. The RMSD value between the OF and IF cryo-EM structures is about 7.5 Å. Therefore, the RMSD value of 6-8 Å is not necessarily large when conformational changes occur.

We fully agree with Reviewer#1 that we should not reason with the high RMSD value of Odera's simulations. However, mistakenly we aimed to provide quantitative reasoning and avoid a statement that their homology modeling likely failed based on visual inspection of their Fig. 2. E.g. unusual conformation of TM11 and TM12 developed that should not happen within 100 ns of simulations independently from the presence or absence of F508. Importantly, the 6-8 Å RMSD for the full structure developed was mostly because of NBD dimerization, since the TM helices did not close.

Nevertheless, we inserted citations for these papers: "Previous studies have aimed to learn Δ F508 effects on the NBD1 structure, but few studies have attempted to understand F508del effects on multi-domain CFTR (Ref 33-34, p.5).

Moreover, recently, MRP1 simulation study has been published by Tóth et al (2023). In it, the RMSD value seems to exceed 12 Å (from the supplement).

These large deviations are derived from rigid body movements, which we aimed to avoid as discussed in our supplementary material (legend of Fig. S6A).

Then, the authors may deny homology modeling, but if so, the results of MRP1 and ABCC6 here are also unnecessary.

We do not have reservations against homology modeling and we find it justified even after AlphaFold (Hegedus et al. CMLS 2022, <https://pubmed.ncbi.nlm.nih.gov/35034173>, “one should be careful with simulations using AI-based structural models, since their conformation may be kinetically trapped into a specific state, inhibiting the study of conformational changes [https://pubmed.ncbi.nlm.nih.gov/34156124/]”).

We rather try to avoid using models in case they do not reach sufficient reliable MD simulations (e.g. indicated by the RMSD values or the unfolding of helices during short MD simulations with the models) regardless whether they derived either from homology model or AlphaFold prediction.

(7) (minor) In relation to the above comment, please show RMSD not only in CFTR (in Fig. S3) but also in MRP1 (and/or ABCC6) (in Fig. S6).

We inserted the RMSD plot calculated from MRP1 simulations (Fig.S6a) and the RMSD values of ABCC6 and MRP1 structures as compared to the template structure in the text (p.37).

(13) We understand that the authors used the structure modeled here rather than the AlphaFold model structure because they assumed an OF state. As such, the use of IF structures in many figures is misleading to the readers. In the MD simulation section, it should be stated more explicitly that the authors analyzed OF structures of CFTR, MRP1, and ABCC6.

We stated now explicitly that OF structures were used in all MD simulations and the corresponding analysis is with these OF structures (Methods). Moreover, we replaced some IF structures with OF structures. We use IF structures for visualization purposes, since important regions are buried and are not visible in the OF conformation.

Moreover, in the Supplementary, AlphaFold structures are used in Figs. S2 and S5. Fig. S2 is good because it is a comparison between OF structures. However, why are AlphaFold's MRP1 and ABCC6 displayed in Fig. S5 even though the authors have done their own modeling? In science, it is necessary to understand what is close to the truth. If the AlphaFold model is more appropriate, the authors may add a few notes to the Fig legend or Methods (such as showing the AlphaFold structure in some parts of the Suppl.).

We fully agree with the reviewer and replaced the AlphaFold structures of ABCC6 and MRP1 with our homology models. We inserted OF structures into Fig.S2 and Fig.S5B.

Below are some additional minor comments (sorry).

(15) (additional) In l.93 of MS, MD results (Fig. S3a) are suddenly cited, but this sentence seems

unnecessary (nothing could be said from the present simulations).

This section has been replaced with:

The contribution of NBD1-TMD1(CL1) coupling to the WT-CFTR and the F508G-CFTR-2PT domain-folding was assessed by disrupting the E403(NBD1)-R170(CL1) electrostatic interaction with the R170G CF-mutation (Fig. S1f). Although the R170G mutation severely diminished the CFTR cellular/PM expression and caused TMD1/2 and NBD2 allosteric misfolding, R170G has a limited effect on fast dynamical motions of TMD1 (Fig. S3a) consistent with the inference that R170-E403 interaction has permissive effect on the CFTR conformational maturation by strengthening the NBD1-CL1(TMD1) interface coupling. Both the expression and folding defects of the R170G-CFTR were rescued by the 2PT-suppressor (Fig. 2d and S1g-h).

(16) (additional) Please check the difference between F1140 in Fig. S5b and F1141 in l.243 (for ABCC6), and R1173 in the fig. and R1172 in the same line (for MRP1).

We corrected these inconsistencies.

(17) (additional) While reading the MS for peer review, a little interesting thing was found as following. The RMSF values around the residue 1070 of the mutant is larger than those of WT in Fig.S3c and Fig.S6a. On the other hand, in Fig. S4, those RMSFs in R170G and P67L mutants are slightly smaller than those of WT. If this point reinforces anything, please describe a little in MS.

In the Fig. S3c (now Fig. S3d) the 1070 CFTR residue does not correspond to the 1170 MRP1 residue in Fig S6a (now S6b), which is located in the intracellular helix towards the membrane bilayer (not at the interface). Nevertheless, the F508G directly affects the interface with 1070 residue, while mutations of the 170 and 67 residues are distantly located from R1070 in 3D, thus the difference is not unexpected. We do not think that the slightly smaller RMSF value of R1070 in these mutants reinforce an important phenomenon.

Reviewer #2 (Remarks to the Author):

My concerns have been properly addressed, I have no reservation regarding the publication of this research work. Thanks to the thorough work of the authors and other reviewers, the quality of this manuscript is very much improved.

We thank Reviewer #2 for constructively participating in the review process and appreciating of our work.

Reviewer #3 (Remarks to the Author):

We thank Reviewer #3 for the suggestions to further improve the presentation (a) and the encouraging words to improve the clarity the scientific message, as well as to better “articulate the experimental plan” (b).

I had two main concerns regarding the original manuscript: (A) clarity of language and presentation and .

Section A: I do not believe the clarity of the revised manuscript is sufficiently improved for publication yet. The paper is still extremely hard to read, with grammatical and typographical errors in the text, and unclear layout and inaccurate labelling in figures. Following are my detailed suggestions for improvement. However, I think a potential final version would require further careful proofreading and a special effort at making ideas more accessible.

Abstract

What is meant by "combinatorial approaches"? I do not think the methods relate to combinations of different parts. Suggestion: "Different approaches..."

The wording was changed as recommended.

“Allosteric or orthosteric binding of VX-809 or VX-445 folding correctors can rescue kinetically trapped CFTR post-translational folding by cystic fibrosis (CF) mutations in NBD1 or TMD1, which requires NBD1-TMD2 coupling.” Suggestion: “Allosteric or orthosteric binding of VX-809 or VX-445 folding correctors can rescue post-translational folding of CFTR that is kinetically trapped by cystic fibrosis (CF) mutations in NBD1 or TMD1. This rescue requires NBD1-TMD2 coupling.”

“offering a framework for mechanistic dissection of genetic diseases caused of ABC transporters.” Suggestion: “offering a framework for mechanistic dissection of genetic diseases caused by mutations in ABC transporters.”

The wording was changed as recommended, while trying to comply with the word limit.

Page 3

“The CLs drive the TMD1/2 conformational transitions upon the NBDs association-dissociation cycle, a requirement for substrate translocation that is facilitated by domain-swapped structural elements of P-glycoprotein (ABCB1)- like exporters, enabling communications of each domain with all the others.” Suggestion ““The CLs drive TMD1/2 conformational transitions upon NBD association-dissociation cycles. The NBD/TMD coupling is a requirement for substrate translocation and is facilitated by domain-swapped structural elements in Type IV exporters (Thomas et al., 2020 <https://doi.org/10.1002/1873-3468.13935>), which enable communication of each domain with all the others.”

“the empirical development of the FDA-approved Trikafta for ~90% of CF-patients.” Suggestion: “the empirical development of the FDA-approved Trikafta for which ~90% of CF patients are eligible” [see comment 2. of Reviewer #4]

The wording was changed as suggested.

Page 5

“First, at the cellular level, the transporters biosynthetic processing, expression, and metabolic stability, as well as their domains’ conformational dynamics, were monitored” What is meant by monitoring of "their domain's conformational dynamics" at a cellular level? I think no measurements directly monitoring “domain conformational dynamics” were obtained at a cellular level. Inferences based on core vs. complex glycosylation were made, but that is not equivalent.

To improve clarity, we replaced “conformational dynamics” with “conformational stability” in the first sentence, although these terminologies are exchangeable used in the literature. The tertiary structural stability of CFTR variants were determined by measuring the transporter protease resistance with limited proteolysis in combination with immunoblotting in isolated microsomes (native-like environment). This approach requires the isolation of microsomes from CFTR expressing cells and uses monoclonal antibodies that can recognize the epitope (specific for individual domains) in the proteolytic fragmentation pattern four core domains. The description of the method and its capacity to probe CFTR domain misfolding or stabilization are documented in our previous publications (e.g. Nature Struct Biol 1997, Nature Struct Mol Biol 2005, Mol Biol Cell 2009, Science 2010, Cell 2012, Nature Med 2018) and also described on p. 26 (Methods: “*In situ* conformational stability determination of CFTR, MRP1 and their domains using limited proteolysis and domain-specific immunoblotting”).

We did included results demonstrating the impact of interface and CF-causing mutations on the global conformational stabilities of NBD1/2 and TMD1/2 by using the limited proteolysis and immunoblotting technique on the following CFTR variants; NBD1 interface mutants (F508G-, F508G-2PT-, R170G-, and R170G-2PT-CFTR), and CF causing mutations (P67L-, P67L-6SS-, L206W- and L206W-6SS-CFTR).

Page 6

“regardless their exit from ER and N-linked complex-glycosylation permitted or not.” Suggestion: "regardless OF WHETHER their exit from THE ER and N-linked complex-glycosylation ARE permitted or not"

The wording was corrected.

Supplementary Data 1 needs tidying up: remove worksheets 1 and 3; correct spelling mistake (efficiency); define what is meant by "corrected"; is the error shown standard deviation (as in worksheet 1), or Standard Error, of the mean (as in worksheet 2)? Check consistency of text and table: Is modulator exposure during (depletion+pulse) or (pulse+chase) only rescuing 6% and 7% of F508del-CFTR? "if drug(s) were present only during the chase periods [...] the folding efficiency of the Δ F508- and L206W-CFTR was restored to ~31% and ~37%, respectively" - but I don't find any 31% and 37% values in the table.

We apologize for the accidental submission of our work sheet in progress and thank the Reviewer#3 for noting this mistake, which was corrected. The consistency of the text and the primary data has also been revised and corrected.

Page 8

“reduced the maturation efficiency of the core-glycosylated F508G-CFTR from $31.0 \pm 2.2\%$ to $\sim 1.8 \pm 0.1\%$ ” Why does Supplementary data 1 have two values slightly above 2% and no 1.8%?

The F508G-CFTR measured and corrected maturation efficiency is $2.06 \pm 0.3\%$ and $2.3 \pm 0.7\%$ ($n=5$), respectively. The corrected maturation efficiency was obtained by taking into consideration the accelerated removal of the complex-glycosylated forms of the indicated mutants (F508G, F508G+R170G, and F508G+R170G+2PT) during the chase period in the Supplementary Data 1.

We apologized for the inadvertent inclusion of an interim excel data sheet in the original manuscript submission, which was deleted from the revised version.

“The 2PT-suppressor mutations have been established to stabilize NBD1 variants and are located outside of the domain interfaces with exception for the S492P.” Suggestion: “The 2PT-suppressor mutations have been established to stabilize NBD1 variants and are mainly located outside the domain interfaces (S492P however, is positioned at the NBD1/CL4 interface).”

The wording was corrected.

Page 9

“by the 2PT-suppressors at peptides confined to all three domain interfaces” What is meant by “confined”? Many of these peptides extend from the domain core to the interfaces.

The sentence has been replaced by: “The WT-like the HDX kinetics of the F508G-NBD1 was globally attenuated by the 2PT-suppressors (except residues 579-594 and 631-650), including peptides in contact with all three domain interfaces (Fig. 2j and S2a), suggesting that structural and dynamic stabilization of F508G-NBD1-2PT interfaces may contribute to the F508G-CFTR-2PT improved folding”

Page 10

“Allosteric coupling between dynamic communities (e.g.: CL3 and CL4, TMH1+TMH2 and CL1) was attenuated (Fig. 3b-c and S4a-b)” This all still seems unclear. Are we comparing WT with F508G- (the only mutation mentioned in this section yet) or WT- with F508G-6SS-CFTR? Would a direct WT- to F508G-CFTR comparison in Fig. S4a be useful.

We made this sentence clearer as: “Allosteric coupling was attenuated indicating dynamic uncoupling between structural segments (Fig. 3b-c and S4a-b) and associated perturbations likely contributed to impeding the post-translational conformational maturation of F508G-CFTR and the destabilization of its final complex-glycosylated form (Fig. S1c-e).”

We understand the difficulty of reading the Fig. S4a plots. To facilitate comparison, black vertical lines indicate the betweenness centrality values in WT-CFTR on each graph. Importantly, we refined our message that the 6SS-mutations partially restore WT-like dynamics and conformational stability.

“Although the centrality of the CL4 ensured an augmented dynamic coupling via the F508 residue, it could marginally suppress the R170G-CFTR channel global misfolding. This was marked by the allosteric conformational destabilization of CFTR domains and ~80% loss of the complex glycosylated and PM-resident R170G-CFTR pools, both reversed by the 2PT-suppressors (Fig. 2d and S1g-h). These observations with the F508G-CFTR-2PT phenotype support the notion that stabilization of fast atomic motions at both CL1- and CL4-NBD1 interfaces contributes to the allosteric conformational rescue of the F508G-CFTR by the 2PT-suppressors, because both interface disruptions thwarted the 2PT dependent folding rescue (Fig. 2d and S1h).” Suggestion (although I am not sure I understood correctly): “Although the INCREASED centrality of the CL4 ensured an augmented dynamic coupling via the F508 residue, THIS

could ONLY marginally suppress the R170G-CFTR channel global misfolding. The R170G-CFTR DEFECT was marked by the allosteric conformational destabilization of CFTR domains and ~80% loss of the complex glycosylated and PM-resident R170G-CFTR pools, both reversed by the 2PT-suppressors (Fig. 2d and S1g-h). These observations, IN CONJUNCTION with OBSERVATIONS ON the F508G-CFTR-2PT phenotype (FIG. XX) support the notion that stabilization of fast atomic motions at both CL1- and CL4-NBD1 interfaces CONTRIBUTE to the allosteric conformational rescue of the F508G-CFTR by the 2PT-suppressors, because SIMULTANEOUS DISRUPTION OF BOTH INTERFACES thwarted the 2PT-dependent folding rescue (Fig. 2d and S1h)."

The wording has been changed as suggested.

"The 6SS-suppressors (Supplementary Data 2) that impart a greater thermal and backbone stabilization of the NBD1 than the 2PT-suppressors (Fig. 3a and 2j), restored the CL4 dynamics, as well as the native contacts of NBD1-NBD2 and NBD1-CL1, but not the NBD1-CL4 interface in the F508G-CFTR as expected (Fig. 3b and S3d)." Suggestion "The 6SS-suppressors (Supplementary Data 2) that impart a greater thermal and backbone stabilization of the NBD1 than the 2PT-suppressors (Fig. 3a and 2j), restored CL4 dynamics, as well as the native contacts of NBD1-NBD2 and NBD1-CL1, but not OF the NBD1-CL4 interface in the F508G-CFTR, as expected (Fig. 3b and S3d)."

The wording has been changed as suggested.

Page 11

"The conserved F713(ABCC6) and F728(MRP1) of NBD1 H3-H4 loops (Fig. S5a) are engaged in a hydrophobic patch formation with F1141/F1146(CL7/TMD2) and a cation- π interaction with the R1172 (CL7/TMD2) in ABCC6 and bovine MRP1-transporters, respectively (Fig. S5b)" R1172 is present in bovine MRP1, but alignments (even yours in Fig. S5a) show it is not homologous to R1070 in CFTR. Is there a basic residue interacting with F713 in human ABCC6? Residue 1172 is an alanine, not an arginine in ABCC6.

Check numbering inconsistencies R1172/R1173 (text/Figure S5a legend) possibly due to MRP1 bovine/human differences.

R765 may interact with F713 in ABCC6. However, this Arg is in the NBD1 and not in the coupling helix, thus this interaction will not directly contribute to interface stabilization. We proposed that the conserved side chain of F713(ABCC6) in the NBD1 H2-H3 loop interacts with hydrophobic patch, formation by F1141/F1146 of the CL7/TMD2 in ABCC6 Fig.S5b (right panel).

In the homology model of human MRP1, the cation- π interaction is predicted to be formed between the side chains of the F728(NBD1) and the R1173(CL7) residues. We apologise for misnumbering the R1172 in the Result section. This has been corrected, consistent with the labelling on the homology model of the hMRP1. The cation- π interaction is predicted to be formed between the side chains of the F729(NBD1) and the R1173(CL7) residues, as was correctly illustrated in the previous Fig.S5b. We agree with Reviewer #3's relevant comments. The conservative scores for individual residues in the CL7 indicate that there is only a modest conservation for the R1173 position. The explanation for conservative scoring scale labelling was included in the revised Fig. S5a legend.

Page 12

"Our assumption is supported by the bidirectional rescue mechanism of NBD1 CF-mutations (e.g. Δ F508)

with correctors targeting the TMDs and NBD2 or genetic suppressors of the NBD1." This sentence is unclear. What assumption? What exactly are the two "directions" implied? Does it relate to the synergistic rescue by correctors and genetic revertant? Or to the mutual stabilization of NBD1 by other domains and other domains by NBD1? This needs to be clarified.

The sentence has been rephrased as follows:

The notion that the NBD1 primary folding defect, caused by CF-mutations (e.g. $\Delta F508$), can be rescued both in cis and in trans perturbations is supported by the observations that targeting the TMDs or NBD2^{12,15,63,64} with corrector molecules or suppressor mutations can alleviate the $\Delta F508$ -CFTR TMD1/TMD2 and NBD2 conformational defects^{51,57}.

Page 13

"Intuitively, the 6SS-suppressors not only stabilized the conformational dynamics of the isolated NBD1 interfaces (Fig. 4a and S2d), but their coupled interfaces in trans and beyond in the isolated CFTR. This was visualized by projecting individual peptides differential deuteration in 6SS- and WT-CFTR [% $\Delta D(6SS-WT)$]" Suggestion: "Intuitively, the 6SS-suppressors not only stabilized the conformational dynamics of peptides positioned at interfaces in isolated NBD1 (Fig. 4a and S2d), but also of peptides positioned in other domains, beyond the NBD1 interface in the full length CFTR. This was visualized by projecting the differential deuteration [% $\Delta D(6SS-WT)$] of individual peptides"

The sentences have been changed as recommended.

Page 14

"The reduced deuteration of the TMH1, TMH2, TMH5, TMH8, and TMH11 segments became apparent only after 20 min incubation (Fig. S8)." All TM11 peptides I can spot (from residue 1069 to 1120) are less deuterated throughout the time course.

The sentence was replaced by:

The TMH1, TMH2, TMH5, and TMH8 segments reduced deuteration became apparent after 4 min incubation, while that of the TMH11 was suppressed from 10 sec onwards (Fig. S8).

"132-1336" to "1325-1336"

The typo has been corrected.

"Thus, allosteric dynamic rigidification contributes to the stabilization of multiple domains in the 6SS-CFTR (Fig. S4c and S7h), albeit the channel global conformation remains similar to that of the WT-CFTR, suggested by their comparable sensitivity to PKA-mediated activation (Fig. S7c)"

What is meant by "allosteric dynamic rigidification"? Does it mean that some loops in NBD1 positioned at domain interfaces become less flexible, affecting conformational flexibility of other domains? (I don't think Fig. S4c is relevant, as it does not include WT-6SS-CFTR). Wouldn't it be expected that reducing mobility of some loops at domain interfaces would not alter the global conformation of a multidomain protein, just "rigidify" it?

Thank you for noting the incorrectly references Fig. S4c. The correct panel (Fig. 4c) was used in the revised manuscript.

Yes, as you stated, the 6SS-induced NBD1 interfaces dynamic rigidification leads to the altered TMDs peptide folding (opening) energetics, based on calculations using the differential HDX kinetics of the WT- and WT-6SS-CFTR. Whether this process is associated with considerable structural and function changes during the CFTR maturation, cannot be predicted (e.g. the unexpected effect of R170G and F508G mutations on the global conformation of CFTR without influencing the respective domain dynamics and energetics).

Although the cryo-EM structure of the G551D-CFTR-6SS is comparable to the WT-CFTR and G551D-CFTR (Wang et al, <https://doi.org/10.1101/2022.10.10.510913>, Fideorczuk and Chen, Science 2022, 378:284-90), and WT-like channel function of the 6SS-CFTR was detectable in phospholipid bilayer (Yang et al, BBA 1860:5 1193), we felt that there is a need to examine whether the 6SS has a global impact on the 6SS-CFTR *functional responsiveness to phosphorylation under in vivo condition*. Therefore, we examined the channel PKA-induced fold activation in relation to the channel constitutive activity in human bronchial epithelial cells (CFBE14o-) by using short circuit current (Isc) measurements (which cannot be performed on the non-polarized BHK-21 cell and we do not have access to patch clamp set up. The forskolin-titration induced fold-activation of the WT- and 6SS-CFTR relative to their constitutive activity were similar in CFBE14o- cells (Fig. S7c). The constitutive channel activity indicates the largely dephosphorylated state transport capacity according to the CFTR phosphoproteomics study (Sci Rep. 2019:9 12706-).

The 6SS-CFTR maximal activation by and sensitivity to forskolin in relation to the channel constitutive activity, were comparable to those of the WT-CFTR, indicating that the R-domain phosphorylation and the associated ATP-induced NBD1-NBD2 dimerization, as well as the subsequent ATP hydrolysis remained largely intact in the 6SS-CFTR. Consistent with this inference, the PKA-stimulated 6SS-CFTR open probability was similar to that of the WT-CFTR, determined in reconstituted phospholipid bilayer (Yang et al, BBA 1860:5 1193).

We agree with Reviewer #3 that the comparable magnitude of the activated Isc of the 6SS- and WT-CFFR in CFBE14o- is somewhat unexpected in light of the ~50% augmented maturation efficiency of the 6SS- over the WT-CFTR in BHK-21 cells. However, several factors can explain the different level of expression CFTR variants in the lentivirus transduced polarized human bronchial epithelial (CFBE14o-) cells. Our future experiments will aim to address whether the lower transcript level due to the viral titer variation and/or altered PM residence time may influence the CFTR-6SS macroscopic function in CFBE14o- cells. We feel that elucidating the detailed mechanism behind the comparable channel function of the 6SS and WT-CFTR in CFBE14o- will not influence the validity of our main conclusions and is beyond the scope of this manuscript and will be pursued in a separate study.

“The impact of inter-domain packing on the NBD1 conformational stability remained unknown and was investigated next by comparing the HDX kinetics of the isolated NBD1 and in the context of WT-CFTR.”
Suggestion: “The impact of inter-domain packing on NBD1 conformational stability was investigated by comparing HDX kinetics of NBD1 in isolation versus in the context of full-length WT-CFTR.”

The sentence has been changed as suggested.

"These results revealed that the multidirectional dynamic/energetic interdomain allosteric networks improve the thermal stability of the newly translated NBD1 in the context of CFTR and can, in principle, relay the pharmacological stabilization TMD1/2 to rescue the primary or coupled folding defects of NBD1, which was assessed next." Suggestion: "These results revealed that the multidirectional dynamic/energetic interdomain allosteric networks improve the thermal stability of the newly translated NBD1 in the context of FULL-LENGTH CFTR and can, in principle, relay the pharmacological stabilization OF TMD1/2 to rescue the primary or coupled folding defects of NBD1, which was assessed next."

The suggested changes have been inserted.

Page 15

"Although VX-809/VX-661 correctors were proposed to stabilize the TMD1 native fold by binding to the hydrophobic pocket formed by the TMH1, 2, 3, and 6 of the WT-CFTR (Fig. 5a), the molecular mechanism action of VX-809/VX-661 or VX-445 on CF mutations yet to be elucidated." Suggestion: "Although VX-809/VX-661 correctors were proposed to stabilize the TMD1 native fold by binding to the hydrophobic pocket formed by the TMH1, 2, 3, and 6 of WT-CFTR (Fig. 5a), the molecular mechanism OF action of VX-809/VX-661 or VX-445 on VARIANTS CARRYING CF mutations HAS yet to be elucidated."

The suggested changes have been inserted.

"we employed the P67L and L206W CF mutations (Fig.5a) that are confined to the L0 region and the TMH3(TMD1) in CFTR, respectively, and are susceptible to correction" Suggestion: "we employed the P67L and L206W CF mutations (Fig.5a) that are IN the L0 and the TMH3 (TMD1) REGIONS in CFTR, respectively, and are susceptible to PHARMACOLOGICAL correction"

The suggested changes have been implemented.

Page 16

"To directly demonstrate the P67L-induced inter-domain perturbations at the backbone dynamics level and its reversal by folding correctors, we used the HDX-MS technique on isolated CFTR variants. As the P67L mutation reduced the WT-CFTR expression by >90% (Fig. 5d-e), we selected the P67L-CFTR-6SS variant, which partially restored the P67L-CFTR misprocessing, while ensured acceptable protein yield (Figs. S7d). Importantly, the P67L mutation reduced the thermal stability of the CFTR-6SS (Fig.S7e) and preserved the P67L-CFTR-6SS rescue ability by VX-809 and VX-445 (Figs. 5d right panel and 5i)."

Suggestion: "To directly demonstrate the P67L-induced inter-domain perturbations at the level of backbone dynamics, and their reversal by folding correctors, we used the HDX-MS technique on isolated CFTR variants. As the P67L mutation reduced the WT-CFTR expression by >90% (Fig. 5d-e), we selected the P67L-CFTR-6SS variant, which partially mended P67L-CFTR misprocessing, while ensuring acceptable protein yield (Figs. S7d). Importantly, the P67L mutation reduced the thermal stability of the CFTR-6SS (Fig.S7e) and preserved the susceptibility to VX-809 and VX-445 rescue (Figs. 5d right panel and 5i)."

The suggested changes have been implemented.

"The P67L mutation provoked accelerated fast atomic motions at the F508-loop, CL4, and NBD2 that

were mitigated by the 6SS-suppressors based on MD simulations (Fig. S4b-e), amplifying the dynamic coupling of intracellular parts of TM helices, as was observed for the F508G-6SS mutant (Fig. S3d-f).” This sentence has not been replaced, despite what is stated in the rebuttal. To me it is still unclear. I repeat what I stated earlier: in Fig S4c, P67L appears to reduce fluctuations at most residues in CL4 compared to WT, especially at the C-terminal end, with the 6SS-suppressor mutations (in P67L background) restoring the high amplitude fluctuations. The opposite happens for the F508 loop: P67L increases fluctuations, and the 6SS-suppressor mutations restore low-amplitude fluctuations. On the contrary, in Fig S3d, the F508G mutation increases the amplitude of fluctuations in the F508 loop, but adding the 6SS-suppressor mutations in the F508G background further increases fluctuation amplitude. Perhaps I am not understanding correctly, but the sentence above seems to oversimplify, and does not describe the data accurately.

We fully agree with the reviewer and intended to change that sentence, but unfortunately it was missed. We rewrote this sentence as follows: “The effects of P67L mutation on fast atomic motions were assessed in MD simulations at the F508-loop, CL4, and NBD2. Similar motion acceleration was observed in F508-loop and NBD2 compared to F508G, while CL4 showed reduced dynamics in P67L (Fig. S3d-f and Fig. S4b-e). Conversely, in the 6SS constructs, CL4 dynamics decreased in F508G and restored in P67L, but a common effect of 6SS in these backgrounds was the increased dynamic coupling of the intracellular parts of central helices (cyan community in Fig. S4b).”

Page 18

“While the first phase consists of the obligatory co-translational formation of secondary structural elements of the core domains, the native tertiary fold completion is delayed by the combination of several processes. Ribosome-nascent chain interactions, and the large size and complex subdomains’ topology of NBDs, as well as the protracted posttranslational development of inter-domain coupling, inherent to the transporters’ domain-swapped structure in analogy to some soluble multi-domain proteins (Fig. 1a and 7a). A main verb is missing from the second sentence. Was a colon intended, instead of a full stop, after “processes”? If so, I’m not sure ribosome-nascent chain interactions and complexity of NBD subdomain topology can be considered “processes”. The final clause “in analogy to some soluble multi-domain proteins” is repeated later (more clearly) and could be omitted here.

To rectify this ambiguity, we made the following changes:

While the first folding phase consists of the obligatory co-translational formation of secondary structural elements of domains, the native tertiary fold completion is likely delayed by the combination of i) ribosome interactions with CFTR nascent chains, ii) the large size and complex subdomains’ topology of NBDs, and iii) the slow post-translational development of inter-domain coupling due to the proteins’ domain-swapped structure (Fig. 1a and 7a)^{35,41,66,67}.

Page 19

“The post-translational domain-folding phase of CFTR is reinforced by the reciprocal dynamic/energetic stabilization of NBD1 in the full-length CFTR (Fig. 4f-h), which is dependent on the conformational maturation of the TMD1/TMD2/NBD2 that consequently requires the reciprocal influence of the NBD1 inter-domain coupling to attain the ER-exit competent conformation (Fig. 7a).” Could this sentence be re-written? I think it refers to the mutual stabilization of (i) NBD1 by other CFTR domains, and (ii) of TMD1/TMD2/NBD2 by a stably folded NBD1. However, as written, the subjects of “is dependent”, and “requires” are unclear.

We rephrased the sentence and the corresponding paragraph in the Discussion (p18.):

“We propose a simplified two-step folding model for the *in vivo* multi-layered conformational biogenesis of ABCC transporters (Fig. 7). This model incorporates critical inter-domain conformational coupling events that, by diminishing the folding intermediates’ propensities to fall into kinetic traps, ensure the development of native conformers at the right time scale (Fig. 7a). While the first folding phase consists of the obligatory co-translational formation of secondary structural elements of domains, the native tertiary fold completion is likely delayed by the combination of i) ribosome interactions with CFTR nascent chains, ii) the large size and complex subdomains’ topology of NBDs, and iii) the slow post-translational development of inter-domain coupling due to the proteins’ domain-swapped structure (Fig. 1a and 7a)^{35,41,66,67}. Allosteric communication is required at the fast atomic motions and backbone dynamics level to minimize kinetic traps along the folding pathway, as well as to achieve stabilization of individual native-like domain. This inference is supported by the domain packing-induced NBD1 energetic stabilization in the full-length CFTR (Fig. 4f-h) and the global consequences of the localized NBD1-CL1/CL4 interface perturbations.”

Page 20

“The NBD1-6SS was able to globally stabilize both the fast atomic motions and backbone NHs dynamics, which translated into favorably folding energy and unfolding activation energy changes of TMD1/2 peptides of WT-CFTR (Fig. 4b-c).” Suggestion: “The NBD1-6SS was able to globally stabilize both the fast atomic motions and backbone NHs dynamics, which translated into favorable CHANGES in folding energy and unfolding activation energy of TMD1/2 peptides of WT-CFTR (Fig. 4b-c).” However, 6SS mutations do not always “stabilize [...] fast atomic motions” (see points referring to statement on P67L on page 16, referring to Figs S4c and S3d, repeated from first round of Reviewer #3 comments).

The sentence has been rephrased in the following paragraph (p.21)

“Although 6SS mutations lack a universal stabilizing effect on the fast atomic motions (see Figs S4c and S3d), the NBD1-6SS stabilized allosterically the fast atomic motions and backbone NHs dynamics in CFTR, which translated into favorably changes in folding energy and unfolding activation energy of TMD1/2 peptides in CFTR-6SS (Fig. 4b-c). Conversely, CFTR was globally destabilized upon uncoupling the CL1- or CL4-NBD1 interfaces. Thus, uncoupling of CL4-NBD1 by the F508G mutation not only misfolded CFTR, but also attenuated the TMD1-dependent VX-809 rescue of P67L-CFTR misfolding without destabilizing the isolated NBD1 (Fig. 5b-d). Concordantly, VX-445, the TMD2 stabilizer, more effectively rescued the F508G-induced NBD1-TMD2 uncoupling than VX-809, while they additively restored the F508G-CFTR expression defect (Fig. 5h).”

Page 21

“This paradigm requires further validation to select possible pharmacotherapy several other ABC-transporters-mediated conformational diseases.”

“This paradigm requires further validation to select possible pharmacotherapies for conformational diseases associated with several other ABC-transporters.”

The sentence was replaced as suggested.

Figures:

Clarity of the figures in general might be aided by using harmonised formats and labels.

- how is "% maturation efficiency" (fig. 1b, 1c) different from "% of matured WT-CFTR" (fig. 2d) or "% CFTR maturation efficiency" (fig 2g) or "% of complex-glycosylated WT-CFTR expression" (fig. 5 e, h)? If they do not differ, they should have identical labels on figures. This should be defined clearly in the Methods section.

The y axis labeling of the metabolic pulse-labelling experiments quantification in Fig.1b, 1c, 1d (% maturation efficiency) and in Fig. 2g (% CFTR maturation efficiency) was harmonized. In the revised manuscript we use % maturation efficiency.

The "% of matured CFTR and % of complex-glycosylated CFTR" (Fig.2d) are synonymous terminology and signify the mature or complex-glycosylated CFTR steady-state expression level, determined by immunoblotting as shown in Fig.2c.

- Could time course of deuteration for selected peptides be added to Figure 6, to make it more similar to Figs. 4a-c, S1, and S6h?

We prepared figure and included as Fig.S9 as requested.

How differential deuteration is abbreviated (in text, legends and figure labels) should also be harmonised. Several abbreviations are used, including $\Delta D(6SS-WT)$; $\Delta\%D$; $\%HDX6SS-WT$; $\Delta HDXP67L-WT$. It might also be helpful to use the same colour scale to indicate differential deuteration in Figs 4b, 6a-c and S6h.

The deuteration abbreviations were harmonized [$\Delta\%D(6SS-WT)$] in the Figures and the text.

We prefer that the color coding is tailored for individual experiments, as the $\Delta\%D$ scale spans highly variable ranges, including both inhibition and acceleration of HDX, while in others it displays a much smaller range.

- RMSD and RMSF are used interchangeably. Better use only one.

RMSF characterizes the mean fluctuation of each residue (RMSF versus plots). Reviewer #1 requested to use RMSD (deviation from the starting structure along the time course of the simulation; RMSD versus time plots) to demonstrate the overall stability of our simulations.

We mistakenly mixed up the explanation of the RMSD abbreviation as root mean square fluctuation (RMSF). This was corrected.

Figures 3c and S4b are difficult to read. Perhaps more extensive labelling could help. For instance, does the purple sphere that gains importance in F508G include TM11-TM10? The labelling and colour coding don't seem to correspond between 3c and S4b: is light blue CL1, CL3 or CL4, or all of them? It might help if a unique scheme could be used, perhaps with addition of a legend, if labels become too crowded.

The color coding across different panels does not represent structural regions; instead, it signifies dynamic communities. As a result, identical regions could be associated with distinct communities, each having a different color. Responding to the reviewer's suggestion to improve the clarity of the figures, we have included additional labeling. Furthermore, we have made tables (in MS Word format) available, along with simulation and parameter files, which can be accessed at <http://www.hegelab.org/resources.html>

Figure S4b could be reorganised, with F508G - WT - P67L on the top row, and F508G-6SS - R170G - P67L-6SS below. This way WT could be easily compared with each single mutant, and F508G and P67L could be compared vertically with their corresponding 6SS versions.

This panel was reorganised for easier comparison.

Figure 3f, left: why are the lanes repeated?

We wanted to demonstrate the technical reproducibility of the method.

Figure 4f: For peptide 490-494, should an "S6" label be added, in red (Q loop), to link to Fig. 4g?

The S6 labeling was included as suggested.

Figure Legends:

Fig. 1a: "The SAME color coding of CFTR domains is used for all relevant illustrations."

Fig. 1b: Label "VX3" is not explained. Suggestion: insert "For the last lane ("labelled 3-VX")," before "Cells were exposed..."

The labeling has been changed and corrected.

Fig. 1c: "Cells that were exposed to VX-445 (2 μ M) and VX-809 (+3 μ M) are indicated by + during the Met/Cys depletion, pulse-labelling and/or chase. Gray box depicts that only pulse-labelling was included without the chase." Suggestion: "Exposure to VX-445 (2 μ M) and VX-809 (+3 μ M) during the Met/Cys depletion, pulse-labelling and/or chase is indicated by "+" in the corresponding rows. A gray box in "Chase" row indicates that only pulse-labelling was included without the chase, while "-" indicates that the chase was performed but modulators were not present." A note could be added on why two lane are included in which correctors were absent from depletion and pulse, and no chase was performed - presumably to have controls for the effect of the chase with modulators (first lane) and without modulators (lane6) [see Reviewer #1, point (1)].

The following explanation was included in the legend:

Exposure to VX-445 (2 μ M) and VX-809 (+3 μ M) during the Met/Cys depletion, pulse-labelling and/or chase is indicated by "+" in the corresponding rows. Pulse-labelling was also included without the chase, indicated by a gray box in "chase" row, to determine the total radioactivity incorporated into the newly synthesized core-glycosylated CFTR in the absence or presence of correctors. This was indicated by a gray box in "chase" row.

Fig. 4b – "deuteration kinetics OF selected" and remove duplication "Means \pm S.D., n=3-4."

Fig. 4c in the legend for Fig. S7i it is stated "Data are means \pm S.E.M., n=3. " In the legend for Fig. 4c "Means \pm S.D., n=3-4. Means \pm S.D., n=3-4." (twice).

The recommended changes have been introduced.

N=3-4 repeat was done for Fig. S7i. We used means \pm S.D. for the evaluation of all HDX-MS data.

Some of the peptides are shown in both figures (e.g. 164-170 and 1063-1067) and look identical. Is the error bar SD or SEM?

We apologize for the duplication of two peptides, which were removed from Fig. S7i.

Fig. 5c "Phosphorimage visualization (left panel) and quantification of the maturation efficiency (right panel)." Suggestion: "Phosphorimage visualization (TOP panel) and quantification of the maturation efficiency (BOTTOM panel)."

The mislabeling was corrected.

Fig. 6a What is SDM?

We corrected S.D.M. to S.D. (standard deviation).

Fig. 6b-c. On page 33 (Methods) you state "was pre-incubated at 25°C for 5 min in the absence or presence of 30 μ M VX-809 or VX-445 correctors. [...] The corrector concentration was kept at 30 μ M during HDX incubation for 10, 240 and 600 s" In the figure legend you mention "3 μ M VX-809 (B) or 2 μ M VX-445 (C)". What was the corrector concentration? In addition, in the methods the longest incubation indicated is 10 minutes (600s). In the figure legend you mention 20 minutes. However, I don't see the 20 minutes data in the Supplementary Data 5 tables - just 10 s and 4 minutes. Can the measurements for the longer incubations be added?

We used 30 μ M VX-809 or VX-445 concentration during the preincubation and the deuteration experiments as indicated in the methods. The deuteration kinetics were measured after 10 sec and 4 min, as the raw data files indicate. We apologize for the incorrectly stated incubation time for 10 min.

Fig. 6b: Add "VX-809 is shown as magenta spheres"

Fig. 6c: The structure shown is not 5UAK. Probably it is 8EIQ, with VX445 in yellow spheres. Please clarify.

The drugs labelling was added and the PDB file name was corrected.

Supplementary Information

S2a: what does the asterisk close to 505-511 indicate?

The asterisks were deleted.

S3a: axis mis-labelled R1070G instead of R170G

This mistake was corrected.

S3 legend “RMSD (root mean square fluctuation) values were also calculated from simulations with complete CFTR structures but computed without taking into account to demonstrate that the high RMSD and some of their fluctuations were caused by the dynamic RI loop.” Something is missing.

A wrong deletion was mistakenly introduced because of the character limitations. The correct sentence: RMSD (root mean square deviation) values were also calculated from simulations with complete CFTR structures but computed without taking the regulatory insertion (RI) into account to demonstrate that the high RMSD and some of their fluctuations were caused by the dynamic RI loop.

Section (B): Overall, I think the rationale for combining experiments on a multitude of variants and systems is now more openly discussed. The many different methods employed have different technical requirements; the different mutations all give broadly consistent results (conformational maturation of CFTR requires interactions that span domain interfaces). However, the fact that experimental results were obtained on different systems (different mutants, different cell lines, different conditions) inevitably means that conclusions are weaker and less mechanistically informative, than if the study had focused on a smaller number of mutant variants and systems.

We respectfully disagree with the statement of Reviewer #3: “However, the fact that experimental results were obtained on different systems (different mutants, different cell lines, different conditions) inevitably means that conclusions are weaker and less mechanistically informative, than if the study had focused on a smaller number of mutant variants and systems.”

We believe that orthogonal methods are expected to be used; in fact, their application is expected by high profile journals to ensure solid support of unexpected biological findings. We think that the following reasons well justify the use of multiple mutations in combination with a variety of complementary techniques to unravel a previously unrecognized molecular aspects of ABCC transporters posttranslational conformational maturation in vivo and highlight the critical role of fast and slow interdomain dynamic coupling during the CFTR, ABCC6 and MRP1 post-translational folding.

i) We used different mutations to rigorously demonstrate the cooperative post-translational folding and misfolding mechanism of CFTR and to address whether selective uncoupling of domain-domain interface(s) (NBD1-CL4 or NBD1-CL1) or destabilization of a single domain (e.g. TMD1) can elicit a global cooperative conformational defect during the posttranslational domain folding. Our thorough approach was also necessary in light of the previous models of cotranslational domain folding mechanism on the ribosome (e.g. *Nature* 1997, *Mol Cell* 2005, 20:277-87, 388:343-9, *Science* 2015, 350:1104-7, and *Nature Struct Mol Biol* 2016:23:278-85).

ii) The consequences of the NBD1-CL1 and NBD1-CL4 interface destabilization were probed by two (R170G and F508G) mutations, alone and in combination. To reverse the consequence of these mutations, we also interrogated two second site suppressor mutations (2PT and 6SS) that were previously characterized. The CF-mutations (P67L and L206W) had to be introduced to probe the allosteric domain misfolding initiated by the CFTR TMD1 conformational defect.

iii) Furthermore, the impact of various mutations on the uncoupling or stabilization of NBD1 interfaces was investigated at four different, but complementary, levels in ABCC transporters: i) at the steady-state

cellular level by determining the mature and immature form expression, ii) at the protein trafficking level (ER conformational determination and post-ER stability), iii) at individual domain level by limited proteolysis and immunoblotting, and iii-iv) at the backbone conformational dynamic level and at the fast atomic motion level by HDX-MS and MD simulations, respectively.

In summary, we believe that applying complementary techniques represents one of the strengths, rather than weaknesses of our work, which appears to be shared by Reviewer #1 ("The results obtained here and their findings would provide important insights into the folding landscape of ABC C-subfamily transporters"), Reviewer #2 ("The work is thorough and of excellent quality, although very dense. The research spans from biophysical characterization to cellular studies, bringing together molecular level observations with cellular ones."), and Reviewer #4 ("...As is always that case for work from this lab, the biochemical studies are extremely well performed, and the results are quite noteworthy. The manuscript is quite a magnum opus, with a huge amount of data presented. The results will be of significance well beyond the CFTR field, and the authors do a pretty nice job of extending their findings to two other ABCC proteins closely related to CFTR").

....To illustrate this point, to my question "Why were changes in biogenesis and in ion channel function associated with the 6SS mutations measured in different [cellular] systems (Fig. S7A,B vs. S7C)?" the authors answer "We choose to perform short circuit current measurements in CFBE14o- cells, as model that allows monitoring of the macroscopic channel function in its cellular environment (as opposed to in BHK-21 cells [where the biogenesis studies were performed])."

These 6SS suppressor mutations (6 point mutations and a 30-residue deletion) cause extensive modifications, including at the crucial NBD1/CL1/CL4 domain interfaces. Clearly, folding and stability are improved. But function might be negatively affected. However, the experiments presented do not inform on the effect of these mutations on the function of individual channels: the overall transepithelial current measured depends both on the number of channels at the membrane and on the single-channel activity.

We did not intend to validate the 6SS impact on WT-CFTR folding efficiency and expression in CFBE14o-cells, or to characterize the 6SS-CFTR biophysical properties at single channel level, but to demonstrate that the 6SS-CFTR channel sensitivity to cAMP-dependent protein kinase (PKA) activation was preserved. This was important because the 6SS-CFTR was used to measure the effect of P67L mutation and corrector molecules on the purified channel conformation dynamics by HDX-MS. As the fractional activation of the 6SS-CFTR and WT-CFTR at increasing forskolin concentrations were similar, we concluded that the 6SS-CFTR channel PKA-dependent complex activation mechanism remained largely unaffected despite the extensive modifications of the NBD1.

Because biogenesis (maturation and stability) and function were measured in different systems, it is unclear whether the unchanged overall current measured is the result of (i) a lack of folding/stability improvement in this system (unlike in all other experimental systems presented in the paper) or (ii) a functional inhibition. There might be technical reasons underlying the unusual experimental design, but this is clearly a limitation. (Note that two of the differences the authors cite in the rebuttal - higher ATP, lack of N-terminal SUMO* tag - will result in an increased single-channel open probability, making it even more surprising that they do not see an increased short-circuit current, given the presumably increased plasma membrane density of 6SS-CFTR compared to WT-CFTR).

This concern was partially addressed in the previous section. We would like to emphasize, that the purpose of these experiments was to assess whether the 6SS mutations does alter the channel susceptibility to PKA-induced activation, and not to complement our CFTR processing studies performed in BHK-21 cells.

Regarding the technical reason: we lack the instrumentation for patch clamp studies and short-circuit current measurements cannot be performed on BHK-21 cells. Thus, the inclusion of short-circuit current measurements on the CFBE14o- model was well justified.

The statement “... *that experimental results were obtained on different cell lines....*” requires some specifications. Our cell-based results (35 panels) were obtained, predominantly, on one cell type, BHK-21 cells. We used the CFBE14o- epithelia (2 panels) cells to rule out that 6SS mutations impose a functional perturbation of CFTR. The results of these studies suggested that the 6SS do not elicit significant conformational perturbations that can interfere with the PKA-dependent WT-CFTR channel function.

I am not fully convinced that the extensive investigation on systems including the 6SS suppressor mutations, are truly informative with respect to folding of WT-CFTR and of variants bearing CF mutations. While 6SS suppressors might facilitate purification of mutant protein, the assumption that they restore interdomain coupling (e.g. in F508G- and P67L-CFTR mutants) that is similar to that in WT-CFTR is not necessarily valid. The betweenness centrality graphs (Fig. S4a), for instance, highlight how the 6SS suppressor mutations appear to shift the allosteric pathways: e.g. centrality of residues in the latter part of ICL4 and in TM11 is low in WT and in F508G, but high in the F508G-6SS-CFTR; centrality of residues 350-375 is low in WT and in F508G, but high in F508G-6SS. A “rigidification”, expressed as reduced loss of native contacts at the NBD1-NBD2 interface by 6SS suppressor mutations (Figs. 3b), also suggests the rescue of conformational maturation is not achieved by restoring WT molecular dynamics.

The 6SS- (and 2PT-) suppressor mutations represent one of the genetic tools, besides CF-causing and interface mutations as well as pharmacological chaperones, to interrogate the consequences of various perturbations of CFTR on the allosteric communication networks both in the WT and mutant background. As we lack a corrector drug that can efficiently stabilize the NBD1 (contrary to the available TMD1 and TMD2 specific stabilizers), the 6SS and 2PT may also be considered as a substitution for NBD1 stabilizing small molecules.

Jointly, our results we provide novel mechanistic insight into the 6SS- (or 2PT-) induced CFTR long-range conformational dynamic stabilization at the full-length channel, domain, peptide (backbone NHs dynamics), and fast atomic motions at amino acid levels that can not only reshape the posttranslational conformational folding landscape of the F508G-, R170G-, and P67L-CFTR variants but the WT-CFTR as well. These observations with the relevant HDX-MS and MD simulation results, including that of the P67L-CFTR-6SS, illustrate that suppressor mutations and pharmacological chaperones exert their allosteric effect via modulating the coupled domain folding landscaper of CFTR variants. The allosteric domain misfolding P67L and L206W mutants could also be rescued by the TMD1 or the TMD2 folding corrector (VX-809 and VX-445), displaying a long-range effect that was propagated by backbone dynamic stabilization to other domains in the P67L-CFTR, based on HDX-MS and limited-proteolysis results. While we agree with Reviewer #3 that our explanation may provide a partial account for the CFTR allosteric (mis)folding and rescue (e.g. reinstate the allosteric communication between two protein segments occurring along distinct and/or partially overlapping allosteric pathways), nevertheless it provides the

first mechanistic model for interpreting the global structural defects of point mutations and their rescue by allosterically acting correctors and suppressor mutations in CFTR.

Reviewer #4 (Remarks to the Author):

The authors are commended for doing a great job in responding to Reviewer input. However, there remain a number of small errors in the writing, that would surely be caught if the three senior authors would read the manuscript carefully.

The remaining mistakes will be corrected. Thank you for your time and constructive criticism.

REVIEWERS' COMMENTS

Reviewer #1 (Remarks to the Author):

The authors addressed almost all of my concerns. It's worth publishing. We thank the authors for their contributions to this field, as well as the authors and reviewers for their efforts in revising the manuscript.

As a very minor point, comparing the previous and revised versions of Fig. 1d, the chase-only bars (-, -, +) have been reduced from ~37% to ~32%, according to the revised table. So "~32% and ~32%" (or "both ~32%") seems appropriate for line 141 in MS.

(very very minor: In the abstract (for general readers), it seems better to simply use "NBD1-TMD1/2" instead of the complicated "NBD1-TMD2/-TMD1"; also it would be "but 'is' indispensable". We'll leave them up to the authors.)

The authors are advised to carefully double-check the data and descriptions for discrepancies in the next revision before publication (to avoid author corrections).

Reviewer #3 (Remarks to the Author):

I thank the authors for taking into consideration my suggestions. In my opinion the clarity of the paper has increased. Some corrections might be still required (below I note a few).

I fully agree with the authors that different, complementary techniques provide added insight, but I still believe a more consistent use of mutations (e.g. studying the effects of the F508G mutation always in the 6SS background, if 6PT did not provide enough expression for HDX-MS studies) and cell lines (same cell line for estimating biogenesis and function of 6SS-CFTR) would have increased mechanistic insight. Regardless, the data presented is certainly informative and undoubtedly supports the authors' "post-translational cooperative domain folding model" for CFTR.

One final suggestion: as a reader, I would appreciate if the internal inconsistency regarding the functional characterization of 6SS-CFTR ("We agree with Reviewer #3 that the comparable magnitude of the activated Isc of the 6SS- and WT-CFFR in CFBE14o- is somewhat unexpected in light of the ~50% augmented maturation efficiency of the 6SS- over the WT-CFTR in BHK-21 cells") could be briefly mentioned somewhere in the paper, not only in the rebuttal.

Minor Corrections

Line 30

"We propose that interdomain dynamic allosteric communications not only regulate ABCC-transporters function but indispensable to tune the folding landscape of their post-translational intermediates." ->

"We propose that interdomain dynamic allosteric communications not only regulate ABCC-transporter function but ARE indispensable to tune the folding landscape of their post-translational intermediates."

Line 116

"We have demonstrated that the conformational maturation of pulse-labelled WT-CFTR and CFTR-ΔNBD2 folding intermediates takes approximately an hour, regardless of their exit from ER and N-linked complex-glycosylation are permitted or not" ->

"We have demonstrated that the conformational maturation of pulse-labelled WT-CFTR and CFTR-ΔNBD2 folding intermediates takes approximately an hour, regardless of WHETHER their exit from ER and N-linked complex-glycosylation are permitted or not"

Line 151

"While the F508 side-chain truncation preserved the F508G-NBD1 WT-like energetic and kinetic

stability of the NBD1..." ->

"While the F508 side-chain truncation preserved the WT-like energetic and kinetic stability of the NBD1..."

Line 202

"Collectively, these results suggest that modulation of NBD1 domain-domain coupling efficiency at least two domain interfaces can allosterically stabilize and destabilize the post-translational conformational landscape of CFTR domains." ->

"Collectively, these results suggest that modulation of NBD1 domain-domain coupling efficiency AT at least two domain interfaces can allosterically stabilize and destabilize the post-translational conformational landscape of CFTR domains."

Thanks for your patience!

Point by point response to Reviewers #1 and #3

We thank both Reviewer #1 and #3 for their unwavering dedication and helpful insights in the extended review process of our manuscript, as well as providing valuable feedbacks.

Reviewer #1 (Remarks to the Author):

As a very minor point, comparing the previous and revised versions of Fig. 1d, the chase-only bars (-, -, +) have been reduced from ~37% to ~32%, according to the revised table. So "~32% and ~32%" (or "both ~32%") seems appropriate for line 141 in MS.

According to the final analysis the revised text reads as follows:

...if drug(s) were present only during the chase periods, when most post-translational folding occurs, the folding efficiency of the Δ F508- and L206W-CFTR was restored to ~32% and 35.4%, respectively (Fig. 1c-d and Supplementary Table 1).

(very very minor: In the abstract (for general readers), it seems better to simply use "NBD1-TMD1/2" instead of the complicated "NBD1-TMD2/-TMD1"; also it would be "but 'is' indispensable". We'll leave them up to the authors.)

This has been changed as suggested.

Reviewer #3 (Remarks to the Author):

I thank the authors for taking into consideration my suggestions. In my opinion the clarity of the paper has increased. Some corrections might be still required (below I note a few).

I fully agree with the authors that different, complementary techniques provide added insight, but I still believe a more consistent use of mutations (e.g. studying the effects of the F508G mutation always in the 6SS background, if 6PT did not provide enough expression for HDX-MS studies) and cell lines (same cell line for estimating biogenesis and function of 6SS-CFTR) would have increased mechanistic insight. Regardless, the data presented is certainly informative and undoubtedly supports the authors' "post-translational cooperative domain folding model" for CFTR.

One final suggestion: as a reader, I would appreciate if the internal inconsistency regarding the functional characterization of 6SS-CFTR ("We agree with Reviewer #3 that the comparable magnitude of the activated Isc of the 6SS- and WT-CFTR in CFBE14o- is somewhat unexpected in light of the ~50% augmented maturation efficiency of the 6SS- over the WT-CFTR in BHK-21 cells") could be briefly mentioned somewhere in the paper, not only in the rebuttal.

As requested, the following sentence was included in the revised manuscript (p.13).

The underlying mechanism of the comparable I_{sc} of the WT- and 6SS-CFTR in CFBE14o-cells remains to be investigated.

Minor Corrections

Line 30

“We propose that interdomain dynamic allosteric communications not only regulate ABCC-transporters function but indispensable to tune the folding landscape of their post-translational intermediates.” ->

“We propose that interdomain dynamic allosteric communications not only regulate ABCC-transporter function but ARE indispensable to tune the folding landscape of their post-translational intermediates.”

The sentence was changed as suggested.

Line 116

“We have demonstrated that the conformational maturation of pulse-labelled WT-CFTR and CFTR- Δ NBD2 folding intermediates takes approximately an hour, regardless of their exit from ER and N-linked complex-glycosylation are permitted or not” ->

“We have demonstrated that the conformational maturation of pulse-labelled WT-CFTR and CFTR- Δ NBD2 folding intermediates takes approximately an hour, regardless of WHETHER their exit from ER and N-linked complex-glycosylation are permitted or not”

Line 151

“While the F508 side-chain truncation preserved the F508G-NBD1 WT-like energetic and kinetic stability of the NBD1...” ->

“While the F508 side-chain truncation preserved the WT-like energetic and kinetic stability of the NBD1...”

The sentence was changed as suggested.

Line 202

“Collectively, these results suggest that modulation of NBD1 domain-domain coupling efficiency at least two domain interfaces can allosterically stabilize and destabilize the post-translational conformational landscape of CFTR domains.” ->

“Collectively, these results suggest that modulation of NBD1 domain-domain coupling efficiency AT least two domain interfaces can allosterically stabilize and destabilize the post-translational conformational landscape of CFTR domains.”

The sentence was changed as suggested.